# Pioneer biocrust communities prevent soil erosion in temperate forests after disturbances

Corinna Gall[1], Martin Nebel[2], Dietmar Quandt[2], Thomas Scholten[1], Steffen Seitz[1]

5    [1]Soil Science and Geomorphology, Department of Geosciences, University of Tübingen, Rümelinstr. 19-23, 72070 Tübingen, Germany
[2]Nees-Institute for Biodiversity of Plants, University of Bonn, Meckenheimer Allee 170, 53115 Bonn, Germany

*Correspondence to*: Corinna Gall (corinna.gall@uni-tuebingen.de)

**Abstract**

Soil erosion continues to be one of the most serious environmental problems of our time and is exacerbated by progressive climate change. Until now, forests have been considered an ideal erosion control. However, even minor disturbances of the forest floor, for example, from heavy vehicles used for timber harvesting, can cause substantial sediment transport. An important countermeasure is the quick restoration of the uncovered soil surface by vegetation. To date, very little attention has been paid to the development of nonvascular plants, such as bryophytes, in disturbed areas of temperate forests and their impact on soil erosion. This study examined the natural succession of pioneer vegetation in skid trails on four soil substrates in a central European temperate forest and investigated their influence on soil erosion. For this purpose, rainfall simulations were conducted on small-scale runoff plots, and vegetation was continuously surveyed during the same period, primarily to map the development of bryophytes and the occurrence of biological soil crusts (biocrusts).

Biocrusts appeared immediately after disturbance, consisting primarily of bryophyte protonemata and cyanobacteria as well as coccoid and filamentous algae that lost their biocrust characteristics as succession progressed. They were present from April to July 2019, with a particular expression in the skid trail that was on shale clay (Psilonotenton Formation) and silty clay loam substrate. In general, skid trails on clayey substrates showed considerably higher bryophyte cover and species richness. Although bryophytes were subsequently overtopped by vascular plants, they managed to coexist until their growth was restricted due to leaf litter fall. *Brachythecium rutabulum* and *Oxyrrhynchium hians* were the most important and persistent pioneer bryophyte species, while *Dicranella schreberiana* and *Pohlia lutescens* were volatile and quickly disappeared after spreading in the summer. Sediment discharge was 22 times higher on disturbed bare soil compared with undisturbed forest soil and showed the largest sediment removal in the wheel tracks. Counteracting this, soil erosion decreased with the recovery of surface vegetation and was particularly reduced with growing pioneer biocrusts in summer, but it again increased in winter, when vascular vegetation became dominant. This leads to the conclusion that the role of bryophyte-dominated biocrusts in forests has been underestimated so far, and they can contribute more to soil conservation at specific times of succession than vascular plants.

# 1 Introduction

For decades, soil erosion has been a major environmental problem, as it degrades the most productive soil layers, which threatens, among other things, food production worldwide. Although these effects have long been known, there are still a variety of challenges to mitigating soil erosion in different ecosystems. As climate change progresses, the risk of soil loss increases, particularly due to increased rainfall intensities, making the preparation of effective solutions an urgent matter (Olsson, 2019; Scholten and Seitz, 2019). The most prominent soil loss occurs in agricultural environments, and thus, considerable relevant research has been conducted in these habitats (Morgan, 2005; Maetens et al., 2012). Soil erosion in forests has received comparably less attention, as undisturbed forest ecosystems generally exhibit the lowest soil erosion among all land-use types (Blanco and Lal, 2008; Maetens et al., 2012; Panagos et al., 2015b) and are seen as a successful countermeasure to prevent the soil from being eroded (Panagos et al., 2015a; Wiśniewski and Märker, 2019).

However, soil erosion in forestlands can be locally severe, due in part to management intensity and tree species composition, for example, in subtropical forest ecosystems (Goebes et al., 2015; Seitz et al., 2016). Even forest disturbances on smaller scales, such as human-induced felling and skidding of individual trees or the construction of forest trail systems on sloped terrain, have the potential to drastically increase soil loss (Blanco and Lal, 2008). Sheridan and Noske (2007) showed that unsealed forest roads accounted for 4.4% of the total annual sediment load from a forest, even though they represented only 0.023% of the catchment. The most important reason for this is soil compaction and reduced infiltration rates caused by heavy machines used for timber harvesting (Foltz et al., 2009; Jordán-López et al., 2009; Wemple et al., 2018; Kastridis, 2020). For instance, results from Demir et al. (2007) revealed a significantly higher soil bulk density on skid trails, where soil compaction is caused by direct overpassing with forestry equipment. In this context, Zemke (2016) measured 58 times higher erosion rates on unfortified forest roads ($272.2 \ \text{g m}^{-2}$) compared with undisturbed forest floor ($4.7 \ \text{g m}^{-2}$) in a temperate forest in western Germany. Also, already vegetated wheel tracks of skid trails showed a five-fold higher soil erosion, up to $21.4 \ \text{g m}^{-2}$. Similarly, Safari et al. (2016) reported an increase in erosion rates of a factor of 14 for bare wheel tracks of skid trails relative to the undisturbed forest floor.

The findings of Li et al. (2019), Seitz et al. (2016), and Shinohara et al. (2019) suggest that it is not primarily the forest canopy that protects the soil against erosion, but an intact forest floor. Several studies have also confirmed that soil erosion on skid

60 trails was highest in the first year after logging and decreased significantly thereafter, mainly due to revegetation (Baharuddin et al., 1995; Jourgholami et al., 2017). Thus, the most important measure to counteract negative effects of soil erosion on the upper soil layer after skidding is a quick restoration of the soil surface by vegetation (Zemke, 2016; McEachran et al., 2018). These protective soil covers consist either of leaf and conifer litter from surrounding trees (Li et al., 2014; Seitz et al., 2015) or understory vascular vegetation on the forest soil (Miyata et al., 2009; Liu et al., 2018). They also include a cryptogamic

65 cover of bryophytes, lichens, fungi, algae, cyanobacteria and various other bacteria lineages within or on top of the first millimetres of the soil, referred to as biological soil crust (biocrust; Weber et al. (2016); Weber et al. (2022)). Especially when vascular plant growth is limited by soil conditions such as high acidity or low nutrient and water availability, biocrusts play a vital role as pioneer soil colonizers and stabilizers (Corbin and Thiet, 2020) and can persist even in temperate climates due to these harsh environmental conditions (Szyja et al., 2018).

70 In mesic environments not necessarily constrained by harsh soil conditions, biocrusts occur primarily as an intermediate state of succession following disturbances such as deforestation (Seppelt et al., 2016), although they may redevelop seasonally if disturbances continue (Szyja et al., 2018; Kurth et al., 2021; Weber et al., 2022). The definition of biocrusts first provided by Belnap et al. (2003) referred to organisms that are in close contact with the soil surface and form a coherent hardening layer. In this context, all organisms with a substantial part of their biomass above the ground are excluded, e.g. large cryptogamic

75 mats consisting of bryophytes or lichens, which are common in temperate coniferous forests. However, especially in temperate climates, the boundaries are fluid, so the distinction between biocrusts and cryptogamic covers is not always easy to make. Consequently, evidence of the occurrence of biocrusts in temperate forests is rare (Glaser et al., 2018; Corbin and Thiet, 2020). Biocrusts in general, and especially bryophyte-dominated biocrusts, are known for their influence on hydrological processes (Eldridge et al., 2020) such as reducing surface runoff (Bu et al., 2015; Xiao et al., 2015) and, thus, decreasing sediment

80 discharge (Silva et al., 2019). Such mitigation of soil erosion is also reported by cryptogamic covers consisting of bryophytes (Pan et al., 2006; Parsakhoo et al., 2012), which is inevitably related to their impressive water storage capacity, since bryophytes are able to absorb up to 20 times their dry weight (Proctor et al., 1998), with some *Sphagnum* species even reaching more than 50 times their dry weight (Wang and Bader, 2018). These mechanisms of water storage capacity are influenced by the complex 3D structure of bryophytes, the composition of a variety of individual functional traits, e.g. leaf area, leaf

frequency, leaf area per shoot length, leaf area index, total surface area, shoot length, and shoot density, and their ability to form dense colony-level cushions (Elumeeva et al., 2011; Glime, 2021; Thielen et al., 2021). As most studies investigating the impact of biocrusts on soil erosion have been conducted in arid and semiarid regions, their influence in humid and temperate climates is largely unknown (Weber et al., 2016; Eldridge et al., 2020). Previous studies in subtropical China proved an important erosion-reducing effect of bryophyte-dominated biocrusts within early-stage forest plantations after clear-cutting (Seitz et al., 2017). It can be assumed that similar effects also occur in humid and temperate forest conditions; however, evidence for these effects is missing.

Pioneer biocrust communities could be particularly important as erosion-controlling agents in recently disturbed forest areas, such as along skid trails, where vascular plants are presumed to grow slowly due to harsh soil conditions. To date, few studies have addressed natural plant succession and its influencing factors in skid trail recovery (DeArmond et al., 2021) and of these, the majority relate exclusively to vascular plants (Buckley et al., 2003; Wei et al., 2015). For example, Mercier et al. (2019) observed on skid trails of different forest types in southern Germany that the species composition of vascular plants and bryophytes differed markedly from the forest interior. Furthermore, these vegetation surveys showed that vascular plant species richness benefited from soil compaction in the skid trails, while bryophyte species richness was unaffected. Overall, there are still a variety of unresolved questions regarding the temporal development of species composition, species richness, and coverage of bryophytes in temperate forest disturbance zones and how they are affected by soil properties such as soil texture, bulk density, pH, and carbon and nitrogen content. With respect to these research gaps, it is of great interest to determine at what time and under what conditions biocrust communities naturally develop after the passing over by forestry machinery and when they transition to a more developed bryophyte cover. It is also important to investigate the functional role of these temperate successional stages of bryophyte cover in soil erosion. The knowledge gained from this study can be used to implement more targeted good forestry practice measures to prevent soil erosion, for example, by enhancing the recovery of cryptogamic vegetation in skid trails.

This study examined the natural succession of pioneer vegetation with a focus on bryophytes and the occurrence of biocrusts in skid trails at four different sites with varying substrates and soil properties in a central European temperate forest. Moreover, it investigated the influence of bryophytes and biocrusts on soil erosion processes measured in small-scale runoff plots (ROPs)

with rainfall simulations, while also considering the position of the tracks within the skid trails. We tested the following hypotheses:

1. Species composition of bryophytes varies depending on individual skid trails

2. Bryophyte cover and species richness are highest in wheel tracks, and total vegetation cover and vascular plant species richness are highest in centre tracks, but each differs depending on the individual skid trail

3. Soil erosion is reduced with increasing vegetation cover and is higher in wheel tracks than in centre tracks

4. Bryophytes and early successional bryophyte-dominated biocrusts are a major factor in mitigating soil losses following disturbances in temperate forests

## 2 Material and methods

### 2.1 Study site

This study took place in Schönbuch Nature Park in southwestern Germany (Figure A1), which is situated in Triassic hills consisting of sandstones, marlstones, and claystones with abundant limestones, and a few Lower Jurassic shales, sandstones, and limestones on the hilltops. The Lower Jurassic plateaus are often covered with a loess layer (Einsele and Agster, 1986). Schönbuch Nature Park represents a low altitude (the highest peak, Bromberg, is 583 m above sea level), hilly (69% with slopes ≤3° and 14% with slopes >15°), and almost completely forested (86%) area in the sub-Atlantic temperate climate zone (Einsele and Agster, 1986; Arnold, 1986). While the mean annual temperature is 8.3°C, the average precipitation is 740 mm (mean annual values from 1979 to 1984 at the climate station in Herrenberg; DWD Climate Data Center (2021a)), which is comparable to the long-term average for Germany (DWD Climate Data Center, 2021c, d).

For this research, four newly established (winter 2018/19) and unfortified skid trails in Schönbuch Nature Park with different parent materials, soil properties, and vegetation characteristics were selected (Table A1). All four skid trails consisted of two wheel tracks and a centre track in between. They were created during logging operations conducted by the state forestry service of Baden-Württemberg (ForstBW) and represented an initial point of vegetation development when this study commenced. The four skid trails were differentiated by their parent material and named according to the geological formation of the parent material: Angulatensandstein (AS), Psilonotenton (PT), Löwenstein (LS), and Trossingen (TS). AS consists of thin, platy, fine-grained sandstones containing limestone in an unweathered state; PT is composed of pyrite-bearing shale clay interstratified

by beds of limestone; TS consists of firm, fractured, unstratified claystones with lime nodules; and LS forms medium- to coarse-grained banked sandstones interrupted by reddish marls (Einsele and Agster, 1986). The AS skid trail  was located next to a loess deposition, which also determines soil properties. Since Schönbuch Nature Park was formed by extensive periglacial processes, the geological formation does not represent the parent rock of soil formation in every case (Bibus, 1986).

In the surroundings of LS, a reforested conifer stand was determined with approximately 70-year-old *Pinus sylvestris* and 50-year-old *Picea abies*, where the former occurred with 50% cover and the latter with 40% cover in the highest tree layer. Furthermore, in a second tree layer, about 20-year-old *Fagus sylvatica* and *Carpinus betulus* had colonized, covering the forest floor with leaf litter over the entire area, such that a herb layer of about 10%−20% was formed, which was mainly restricted to sparse areas and dominated by grasses such as *Carex sylvatica* and *Brachypodium sylvaticum*. Additionally, a soil survey was carried out based on the classification system of the German soil mapping guideline (KA5; Ad-hoc-AG Boden (2005)), and subsequently, the soil types according to the World Reference Base for Soil Resources (WRB; IUSS Working Group WRB (2015)) were derived using the WRB Tool for German Soil Data (Eberhardt et al., 2019). For LS, a Eutric Cambisol (Ochric) with typical moder was identified, and the soil surface was covered with a moss layer up to 5% in total.

In comparison, the natural habitat of TS was dominated by young *Picea abies* (approx. 30-year-old), with 90%−100% of the soil surface covered with moss, and in 5%−10% of the area, a herb layer was formed. The soil survey revealed a Eutric Cambisol (Geoabruptic, Clayic, Ochric, Raptic, Protovertic), which was much deeper than the wheel track in the skid trail and covered with a mull-like moder humus layer.

The other two sites were characterized by deciduous tree species: While PT was formed primarily by beech trees (*Fagus sylvatica*) at different ages, developing a sparse tree layer and a very dense shrub layer, in AS, a sparse tree layer of approximately 100-year-old *Quercus petraea* and a second level of younger *Fagus sylvatica* and *Carpinus betulus* were found. In PT, a soil survey revealed a Eutric Calcaric Amphistagnic Cambisol (Loamic, Ochric) with a mull-like moder humus layer, and in the vegetation survey, a herb layer with a cover rate of less than 5% was determined. In contrast, AS had a 20% herb layer formed almost exclusively by *Quercus petraea* and *Carpinus betulus* seedlings, and the soil type was identified as Dystric Stagnic Regosol (Ochric) with L-mull.

## 2.2 Field and laboratory methods

To test for particular impacts of early successional post-disturbance forest floor vegetation on sediment discharge, rainfall simulations with micro-scale ROPs ($0.4 \times 0.4$ m; cf. Seitz (2015)) were performed at four different times (March 2019, July 2019, October 2019, and February 2020). ROPs are stainless steel metal frames connected with a triangular surface runoff gutter and are used to measure interrill erosion processes (Seitz, 2015; Zemke, 2016; Seitz et al., 2019), which is the discharge of sediment in thin sheets between rills due to shallow surface runoff (Blanco and Lal, 2008). Four ROPs were placed in each

right wheel track (n = 4) and four in the centre track (n = 4) at each of the four skid trails, a total of 32 ROPs. Two ROPs were placed in the undisturbed forest soil adjacent to every skid trail (n = 8). While rainfall simulations in the skid trails were conducted for each of the four measurement times (n = 128), in the undisturbed forest soil, they were reduced to measurements in October 2019 and February 2020 (n = 16), yielding a total of 144 measurements.

Rainfall simulations were conducted with the Tübingen rainfall simulator (Iserloh et al., 2013; Seitz, 2015) that was equipped

with a Lechler 460.788.30 nozzle and adjusted to a falling height of 3.5 m. Mean rainfall intensity was set at 60 mm h$^{-1}$, applied over a duration of 30 minutes. This rainfall intensity refers to a regional rainfall event with a recurrence interval of 20 years (DWD Climate Data Center, 2021b). In each run, two ROPs (wheel and centre track) were irrigated simultaneously, with surface runoff and sediment collected in sample bottles (1 L). An overview of the experimental setup is available in Figure A2. Prior to each rainfall simulation, soil moisture was determined next to every ROP using a Thetaprobe ML2 in combination

with an HH2 Moisture Meter (Delta-T Devices, Cambridge, UK).

After soil erosion measurements, the total surface runoff for each ROP was gathered from the associated sample bottles marked with a millilitre measuring scale. To ascertain sediment discharge, the sample bottles were dried at 40°C in a compartment drier and weighed in a dry state. To determine basic soil properties, bulk soil samples of the topsoil (0−5 cm) were collected in the surroundings of every ROP. While aggregate size was obtained by wet sieving, which served as a basis for the calculation

of the mean weight diameter (MWD) of soil aggregates (Tiulin 1933, Van Bavel 1950), grain size distribution was determined with an x-ray particle size analyser (Sedigraph III, Micromeritics, Norcross, GA, US). Soil pH was measured with a pH meter and Sentix 81 electrodes (WTW, Weilheim, Germany) in 0.01 M CaCl$_2$ solution. Additionally, soil organic carbon (SOC) and total nitrogen (N$_t$) were determined with an elemental analyser (Vario EL III, Elementar Analysesysteme GmbH, Hanau,

Germany). Core samples (100 cm³) were taken to determine soil bulk density in the topsoil using the mass-per-volume method

(Blake and Hartge, 1986). Slope was measured on both sides of every ROP using an inclinometer, while aspect for the entire

sites was derived from a digital elevation model (DEM, Geobasisdaten © Landesamt für Geoinformation und Landentwicklung

Baden-Württemberg) using a geographical information system (QGIS-Version 3.16.13-Hannover; QGIS Development Team

(2020)). Furthermore, skid trails were examined for water repellency by applying the water drop penetration time (WDPT) test

(Dekker et al., 2009).

To investigate the development of vegetation cover on the forest floor surface in every ROP, sampling campaigns took place

at five measurement times (April 2019, June 2019, July 2019, October 2019, and February 2020) synchronized with in situ

soil erosion measurements. Vascular plants and bryophytes were classified by eye and identified by morphological

characteristics using a stereomicroscope (SteREO Discovery.V8, Carl Zeiss Microscopy Deutschland GmbH, Oberkochen,

Germany) and a microscope (Leitz SM-Lux, Ernst Leitz GmbH, Wetzlar, Germany). Classification was carried out to the

species level (Table 1 and Table 2), wherever possible, using the following plant identification literature: Rothmaler (2005),

Nebel et al. (2000), Nebel et al. (2001), Nebel et al. (2005), and Moser (1963). In addition, total vegetation and bryophyte

cover were surveyed for each ROP, while the Braun-Blanquet cover-abundance scale was used to determine coverages at the

species level (Braun-Blanquet, 1964). Due to further use of the TS skid trail after the rainfall simulation in March 2019, it was

not possible to survey the vegetation in the centre track in April 2019. Vascular plant cover was calculated as the difference

between total vegetation cover and bryophyte cover. Furthermore, perpendicular photographs were taken of each ROP with a

digital compact camera (Panasonic DC-TZ91, Osaka, Japan) to additionally assess total vegetation cover with a

photogrammetric survey, and were processed with the grid quadrat method and using a digital grid overlay with 100

subdivisions (Belnap et al., 2001). Bare soil and vegetation covers were separated by hue distinction.

### 2.3 Statistics

All analyses were conducted with R 4.0.4 (R Core Team, 2021) on the level of individual samples. To screen for significant

differences, Kruskal−Wallis tests were used in combination with post hoc Wilcoxon rank-sum tests for independent

measurements and Wilcoxon signed-rank tests for related measurements (using the R package "stats"). To test for significant

differences between cover types, we classified ROPs as bare, bryophyte, and vascular plant ROPs. In bare ROPs, there was

neither bryophyte nor vascular plant cover; bryophyte ROPs were mainly covered by bryophytes; and vascular plant ROPs were mainly covered by vascular plants, at the same time, bryophyte cover was lower than or equal to 10%. A nonparametric analysis of covariance comparing nonparametric regression curves was performed to determine if there was a significant difference between vascular plant ROPs and bryophyte ROPs in terms of sediment discharge (R package "sm"; Bowman and Azzalini (2021)). To determine whether bryophyte species composition differed significantly in the individual skid trails, an analysis of similarity (ANOSIM) with 999 permutations from the R package "vegan" was used (Oksanen et al., 2020). Additionally, generalized additive models (GAM) with restricted maximum likelihood and smoothing parameters selected by an unbiased risk estimator (UBRE) criterion were performed to assess the effect of environmental parameters on soil erosion, total vegetation coverage, bryophyte coverage, and bryophyte species richness (R package "mgcv"; Wood (2020)). Prior to all statistical tests, normality was proved with the Shapiro−Wilk test, while homoscedasticity was verified using Levene's test. Significance was assessed as $p < 0.05$ in all cases. For all mean values described, the standard error was also given (mean ± standard error). The selected colours for Figure 1 are from the R package "wesanderson" (Karthik et al., 2018) and for Figures 3, 4, 5, and 6 from the R package "RColorBrewer" (Neuwirth, 2022).

## 3 Results and discussion

### 3.1 Bryophyte species composition

#### 3.1.1 General succession of bryophyte species composition

Within the vegetation survey at five measurement times, a total of 24 moss, two liverwort and two fungi species were found in the skid trails (Table 1), while 13 moss species occurred in the undisturbed forest soil (Table 2). The first bryophyte species to recolonize the skid trails in April 2019 after skidding were *Brachythecium rutabulum* (53.1% of ROPs) and *Oxyrrhynchium hians* (37.5% of ROPs). Protonemata of various species, the earliest stage of bryophyte development consisting of green cell filaments, were observed in 25% of the ROPs. In June 2019, the percentage of ROPs occupied by *Brachythecium rutabulum* and *Oxyrrhynchium hians* increased to 75% and 40.6%, respectively, while protonemata were found in 31.3% of the ROPs. Furthermore, *Plagiomnium undulatum* occurred in 25% of the ROPs and *Thuidium tamariscinum* occurred in 18.8%. When the first bryophyte shoots developed from protonemata in July 2019, many occurrences could be assigned to the species *Pohlia lutescens*, *Dicranella schreberiana,* and *Trichodon cylindricus*. From July 2019 to February 2020, *Oxyrrhynchium hians*,

*Brachythecium rutabulum*, and *Plagiomnium undulatum* remained the most abundant bryophyte species, and the quantity of

different species increased. In comparison, 13 moss species occurred in the undisturbed forest soil (Table 2), eight of which

were also present in the skid trails.

**Table 1: Percentage occurrence of bryophyte and fungi species for a total of 32 runoff plots distributed in four skid trails in Schönbuch Nature Park in southwestern Germany, based on five vegetation surveys from April 2019 to February 2020**

| SPECIES | PERCENTAGE OCCURRENCE OF SPECIES IN RUNOFF PLOTS | | | | | |
| --- | --- | --- | --- | --- | --- | --- |
| | APR 2019 | JUN 2019 | JUL 2019 | OCT 2019 | FEB 2020 | TOTAL |
| Liverworts | | | | | | |
| *Lophocolea bidentata* (L.) Dum. | – | – | – | – | 12.50 | 12.50 |
| *Apopellia endiviifolia* (Dicks.) Nebel & D.Quandt | – | – | 9.38 | 34.38 | 18.75 | 40.63 |
| Mosses | | | | | | |
| *Atrichum undulatum* (Hedw.) P. Beauv. | – | 3.13 | 6.25 | 15.63 | – | 15.63 |
| *Barbula unguiculata* Hedw. | – | – | 3.13 | 12.50 | 3.13 | 12.50 |
| *Brachythecium rutabulum* (Hedw.) Schimp. | 53.13 | 75.00 | 59.38 | 62.50 | 71.88 | 93.75 |
| *Bryum pseudotriquetrum* (Hedw.) P.Gaertn., E.Mey. & Scherb. | – | – | – | 3.13 | – | 3.13 |
| *Bryum tenuisetum* Limpr. | – | – | 3.13 | 3.13 | – | 3.13 |
| *Calliergonella cuspidata* (Hedw.) Loeske | – | – | – | – | 15.63 | 15.63 |
| *Cirriphyllum piliferum* (Hedw.) Grout | 3.13 | – | – | – | 3.13 | 6.25 |
| *Dicranella schreberiana* (Hedw.) Dixon | – | – | 12.50 | 18.75 | 6.25 | 18.75 |
| *Dicranella varia* (Hedw.) Schimp. | – | – | 3.13 | 15.63 | 6.25 | 18.75 |
| *Didymodon fallax* (Hedw.) R.H.Zander | – | – | – | – | 3.13 | 3.13 |
| *Eurhynchium striatum* (Hedw.) Schimp. | 3.13 | 6.25 | 6.25 | 3.13 | 9.38 | 12.50 |
| *Fissidens taxifolius* Hedw. | – | 3.13 | 31.25 | 40.63 | 34.38 | 46.88 |
| *Hypnum cupressiforme* Hedw. s. str. | – | – | – | 3.13 | 3.13 | 6.25 |
| *Oxyrrhynchium hians* (Hedw.) Loeske | 37.50 | 40.63 | 50.00 | 78.13 | 81.25 | 93.75 |
| *Plagiomnium affine* (Blandow ex Funck) T.J.Kop. | 3.13 | 3.13 | – | 0.00 | – | 3.13 |
| *Plagiomnium undulatum* (Hedw.) T.J.Kop. | 9.38 | 25.00 | 40.63 | 68.75 | 56.25 | 71.88 |
| *Pohlia lutescens* (Limpr.) H.Lindb. | – | 9.38 | 18.75 | 6.25 | – | 18.75 |
| *Pohlia melanodon* (Brid.) A.J.Shaw | – | – | 3.13 | 12.50 | 9.38 | 15.63 |
| *Pohlia wahlenbergii* (F.Weber & D.Mohr) A.L.Andrews | – | – | 3.13 | 12.50 | 3.13 | 12.50 |
| *Pseudoscleropodium purum* (Hedw.) M.Fleisch. | – | – | 3.13 | 9.38 | 9.38 | 15.63 |
| *Rhytidiadelphus squarrosus* (Hedw.) Warnst. | – | – | – | 3.13 | – | 3.13 |
| *Rhytidiadelphus triquetrus* (Hedw.) Warnst. | – | – | – | – | 3.13 | 3.13 |
| *Thuidium tamariscinum* (Hedw.) Schimp. | – | 18.75 | 25.00 | 40.63 | 37.50 | 46.88 |
| *Trichodon cylindricus* (Hedw.) Schimp. | – | – | 15.63 | 25.00 | 6.25 | 31.25 |
| Fungi | | | | | | |
| *Scutellinia kerguelensis* (Berk.) Kuntze | – | – | – | 3.13 | – | 3.13 |

| *Scutellinia umbrarum* (Fr.) Lambotte | – | 3.13 | 3.13 | – | – | 3.13 |


**Table 2: Percentage occurrence of bryophyte species for a total of eight runoff plots in undisturbed forest soil in Schönbuch Nature Park in southwestern Germany, based on one vegetation survey in February 2020**

| SPECIES | PERCENTAGE OCCURRENCE OF SPECIES IN RUNOFF PLOTS IN FEBRUARY 2020 |
| --- | --- |
| *Brachythecium rutabulum* (Hedw.) Schimp. | 25.00 |
| *Brachythecium salebrosum* (F. Weber & D. Mohr) Schimp. | 12.50 |
| *Bryum rubens* Mitt. | 12.50 |
| *Dicranella heteromalla* (Hedw.) Schimp. | 25.00 |
| *Eurhynchium angustirete* (Broth.) T.J.Kop. | 25.00 |
| *Euryhnchium striatum* (Hedw.) Schimp. | 12.50 |
| *Fissidens taxifolius* Hedw. | 12.50 |
| *Hylocomium splendens* (Hedw.) Schimp. | 25.00 |
| *Hypnum cupressiforme* Hedw. | 25.00 |
| *Pohlia melanodon* (Brid.) A.J.Shaw | 12.50 |
| *Polytrichastrum formosum* (Hedw.) G.L.Sm. | 25.00 |
| *Rhytidiadelphus triquetrus* (Hedw.) Warnst. | 25.00 |
| *Thuidium tamariscinum* (Hedw.) B.S.G. | 25.00 |

In our study area, the occurrence of cyanobacteria as well as coccoid and filamentous algae (e.g. Chlorphyceae and Xanthophyceae) plus bryophyte protonemata and the subsequent very early developmental stage of bryophyte shoots fulfilled the definition of biocrusts by Belnap et al. (2003) and Weber et al. (2022), and occurred from April to July 2019. Since the species *Pohlia lutescens*, *Dicranella schreberiana,* and *Trichodon cylindricus* have predominantly evolved from protonemata and formed only a minor part of their biomass above the soil surface in their early developmental stages, we include these species here among the temperate biocrust species. According to the biocrust definition of Belnap et al. (2003), we can also include the thallose liverwort *Apopellia endiviifolia* among the temperate biocrust species in our study area. Furthermore, *Brachythecium rutabulum* and *Oxyrrhynchium hians* have emerged as the most important pioneer species. Both species are widespread in Baden-Württemberg, Germany (Nebel et al., 2001) and are known to colonize a wide range of habitats (Nebel et al., 2001; Atherton et al., 2010). While *Brachythecium rutabulum* is particularly common on wood and stones, growing also

on soil and gravelly ground, the habitat of *Oxyrrhynchium hians* is preferentially restricted to bare base-rich soils (Atherton et al., 2010), which renders both pioneer-friendly mosses (Nebel et al., 2001). Due to its competitive strength and broader

distribution, *Brachythecium rutabulum* was even more frequent in the skid trails than *Oxyrrhynchium hians*. At a more advanced stage of succession, *Plagiomnium undulatum* and *Thuidium tamariscinum* also occurred, both of which grow mainly on forest soils (Nebel et al., 2001; Atherton et al., 2010). Furthermore, a clearly different species composition was found in the undisturbed forest soil compared with the skid trails. There, species composition showed an increased occurrence of more specialized species common in acidic woodlands, such as *Hylocomium splendens, Polytrichastrum formosum,* and *Dicranella*

*heteromalla* (Atherton et al., 2010), which can be attributed to the lower pH in the undisturbed forest soil (mean pH = 4.54 ± 0.07) compared with the skid trails (mean pH = 6.19 ± 0.07). Mercier et al. (2019) also observed a different species composition in skid trails of different forest types in northern Bavaria compared with the forest interior during their vegetation surveys of vascular plants and bryophytes, indicating that skid trails can contribute to higher species diversity in managed forests.

### 3.1.2 Succession of bryophyte species composition in different skid trails

The vegetation succession developed differently in the four skid trails (see Figure 1 and Figure 2) in terms of species composition (p = 0.001). At the beginning of vegetation succession after the disturbance due to skidding, we observed the development of protonemata in AS and PT. Whereas protonemata occurred in AS from April 2019 to July 2019 in 50% of the ROPs, it was less common in PT but reached 50% coverage in two ROPs in June 2019. These protonemata and their early successional stages of *Pohlia lutescens*, *Dicranella schreberiana*, and *Trichodon cylindricus* are classified as biocrusts, which

appeared in both PT and AS in April 2019 after the disturbance occurred and persisted in both skid trails until July 2019. The most abundant pioneer species were *Brachythecium rutabulum* and *Oxyrrhynchium hians* in all skid trails, but *Oxyrrhynchium hians* was absent in TS. TS was clearly dominated by *Brachythecium rutabulum*, which occurred in almost every ROP, with the coverage being up to 50% in centre tracks, increasing constantly during the vegetation survey. *Brachythecium rutabulum* was present in all other skid trails, but with less than 5% coverage. Furthermore, *Thuidium tamariscinum* occurred in TS in

almost every ROP and in centre track plots, also with a considerably high coverage of up to 25% in October 2019 and February 2020; it did not colonize PT or AS, but it was also abundant in LS, with cover up to 5%. Liverwort species developed most notably in October 2019 in PT, LS, and TS, with *Apopellia endiviifolia* occurring in PT and LS, and *Lophocolea bidentata*

found only in TS. While *Plagiomnium undulatum* did not occur in AS, it was very common in all other skid trails, with mostly low coverage (around 5%). Generally, *Plagiomnium undulatum* development started early in summer (June or July 2019) in

PT and LS, and exclusively in autumn in TS. Especially in July and October 2019, *Dicranella schreberiana* was abundant in PT and in some ROPs, up to a coverage of 50%, while it did not grow in all other skid trails. Furthermore, *Oxyrrhynchium hians* achieved high coverage rates of up to 25% in PT.

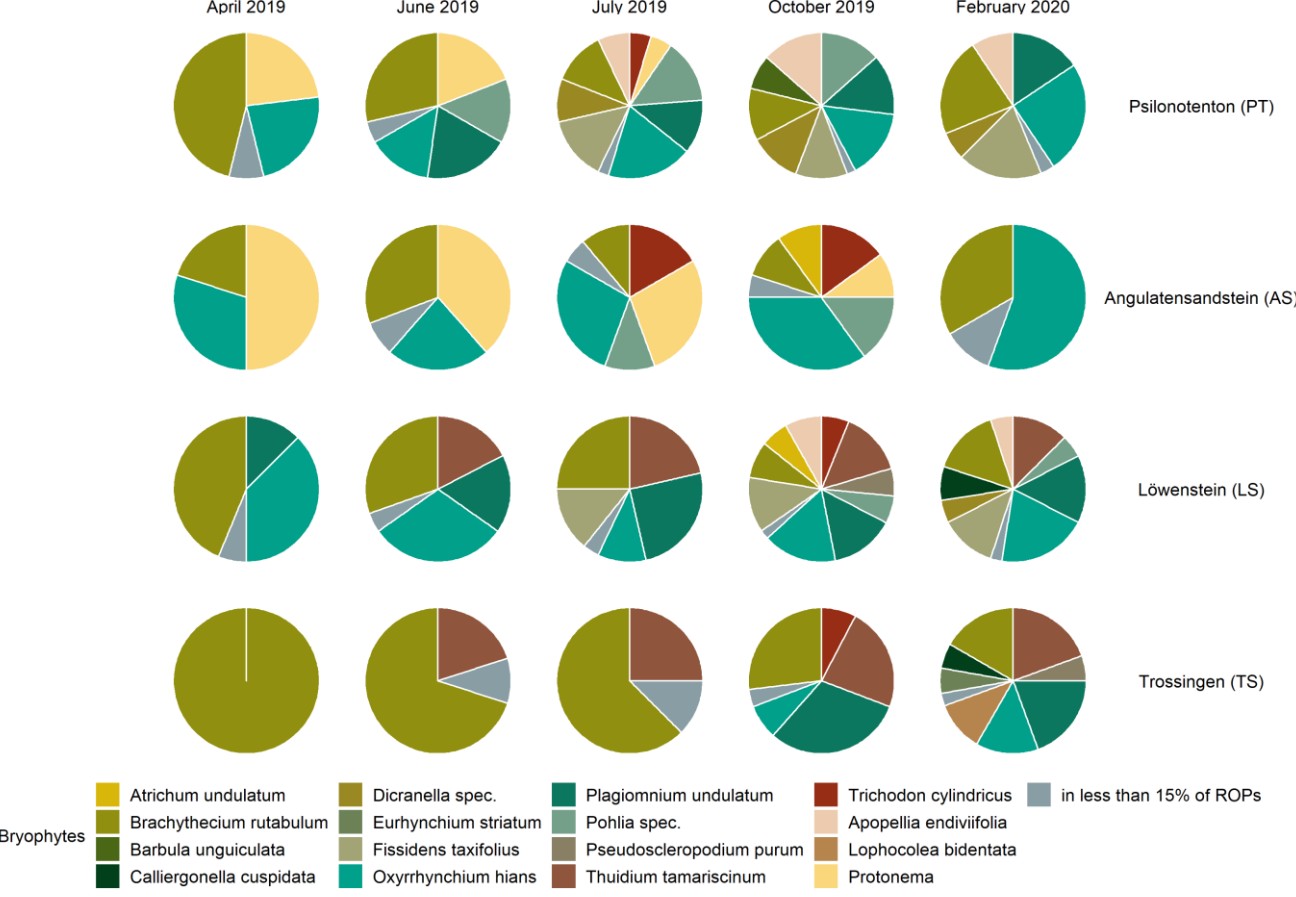

**Figure 1: Bryophyte species composition in the different skid trails for each time of vegetation survey. Species from same genera are**
**grouped together and species, which occur in less than 15% of the runoff plots, are listed in one group.**

Pioneer biocrust species were found in the three skid trails in AS, PT, and LS. It was particularly interesting that the related

moss species *Dicranella schreberiana* and *Pohlia lutescens* were more volatile than expected, spreading only during the

summer and disappearing again at the beginning of autumn. Temporally, the liverwort biocrust species *Apopellia endiviifolia*

appeared just when the moss biocrusts disappeared. As noted by Düll (1991), *Apopellia endiviifolia* is exclusively distributed

at sites with neutral-to-alkaline pH, which is why it occurred in PT and LS in our study area but not in the other two skid trails. *Brachythecium rutabulum* occurred in all skid trails as a pioneer species; however, while in PT, AS, and LS it was associated with other moss species as succession progressed, in TS it was dominant in terms of coverage. Since *Brachythecium rutabulum* is known to be stimulated in growth by eutrophication (Nebel et al., 2001), high $N_t$ in TS could be a possible explanation for its dominant occurrence there. In addition, TS was the only skid trail in which *Oxyrrhynchium hians* did not occur. On the one

hand, this can be attributed to the fact that *Brachythecium rutabulum* is very competitive, especially on eutrophic sites, and suppresses other species (Nebel et al., 2001). On the other hand, TS had a low pH of 5.4 ± 0.11, and since *Oxyrrhynchium hians* grows on base-rich soils, TS is not the preferred growing location. The absence of *Plagiomnium undulatum* in AS can be attributed to the fact that AS was clearly drier than the other sites, and according to Nebel et al. (2001), *Plagiomnium undulatum* is a permanent moisture indicator. This moisture requirement is also shown by the fact that *Plagiomnium undulatum*

occurred comparatively late in the year in TS. We assume that only the formation of a closed vegetation cover of vascular plants at this site developed a sufficiently shady and humid microclimate for *Plagiomnium undulatum* to establish itself there. In this context, Sedia and Ehrenfeld (2003) and Ingerpuu et al. (2005) demonstrated that vascular plants can promote a microhabitat that is more hospitable for moss growth. *Thuidium tamariscinum* occurred exclusively in skid trails surrounded by coniferous forests, which corresponds to its preferential distribution area (Nebel et al., 2001).

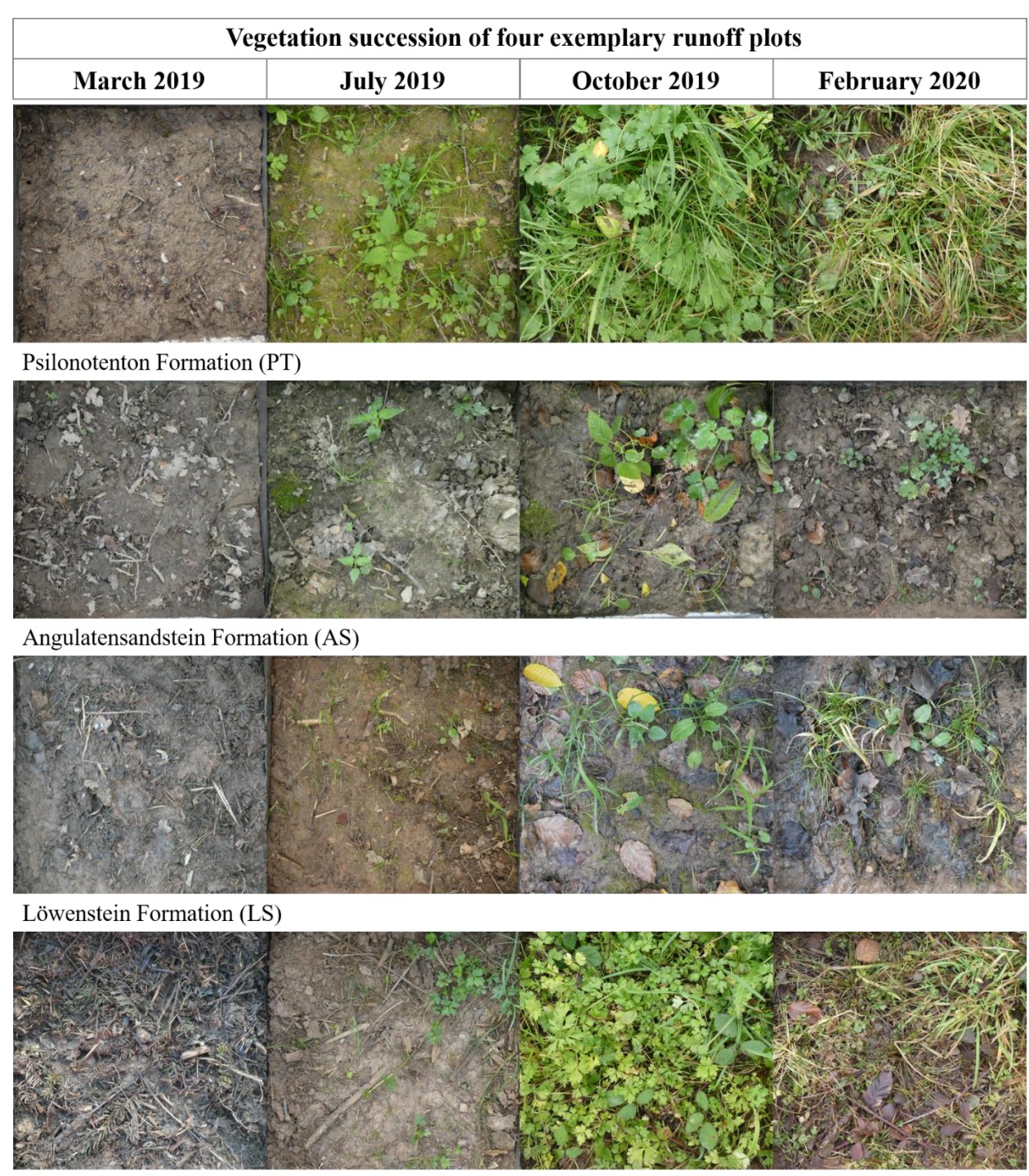

| Vegetation succession of four exemplary runoff plots | | | |
| --- | --- | --- | --- |
| **March 2019** | **July 2019** | **October 2019** | **February 2020** |

Psilonotenton Formation (PT)

Angulatensandstein Formation (AS)

Löwenstein Formation (LS)

Trossingen Formation (TS)


**Figure 2: Vegetation succession of four examplary runoff plots in wheel tracks of the skid trails in Schönbuch Nature Park**

### 3.2. Coverage and species richness

### 3.2.1 Bryophyte and total vegetation coverage

For all skid trails and vegetation surveys, bryophyte coverage was, on average, higher in centre tracks (12.01% ± 1.95) than in
wheel tracks (7.15% ± 1.45; $p < 0.001$), which was also true for total vegetation coverage (centre track: 60.49% ± 3.78; wheel
track: 24.00% ± 3.73; $p < 0.001$). With respect to the individual skid trails, the extent of bryophyte cover varied widely (Figure
3). In AS and LS, bryophyte coverage averaged no more than 12.00%, while in PT it peaked at 33.33% ± 6.67 in July 2019,
and TS achieved 34.64% ± 11.95 in February 2020, with considerable variation in cover between wheel and centre tracks in
the last two skid trails. PT showed a more pronounced development of bryophyte cover in wheel tracks (up to 40% from June
to October 2019), opposite to the preferential colonization of centre tracks in TS (up to 60% in February 2020). While
bryophyte cover in PT decreased between October 2019 and February 2020, this effect did not occur in TS. Calculated in a
GAM that explained 80.3% of the deviation of bryophyte cover, pH ($p < 0.001$), SOC ($p < 0.001$), sand content ($p < 0.001$),
total vegetation coverage ($p < 0.001$), and $N_t$ ($p < 0.05$) were significant.

Generally, total vegetation and bryophyte cover developed with a higher coverage rate in centre tracks, indicating inferior soil
conditions in wheel tracks compared with centre tracks. In this context, we found higher pH values in wheel tracks than in
centre tracks, with the difference being significant for AS (wheel track: 5.8 ± 0.08; centre track: 5.3 ± 0.13; $p < 0.05$), TS
(wheel track: 5.6 ± 0.06; centre track: 5.1 ± 0.12; $p < 0.05$), and LS (wheel track: 7.0 ± 0.04; centre track: 6.8 ± 0.05; $p < 0.05$).
The importance of soil pH on the growth of vascular plants and bryophytes, as well as their composition and diversity, has
also been highlighted in several studies (Löbel et al., 2006; Hydbom et al., 2012; Oldén et al., 2016). For example, Rola et al.
(2021) showed that soils with a more acidic pH promoted larger bryophyte coverage, which could explain, among other things,
the generally higher bryophyte cover in centre tracks in our study.

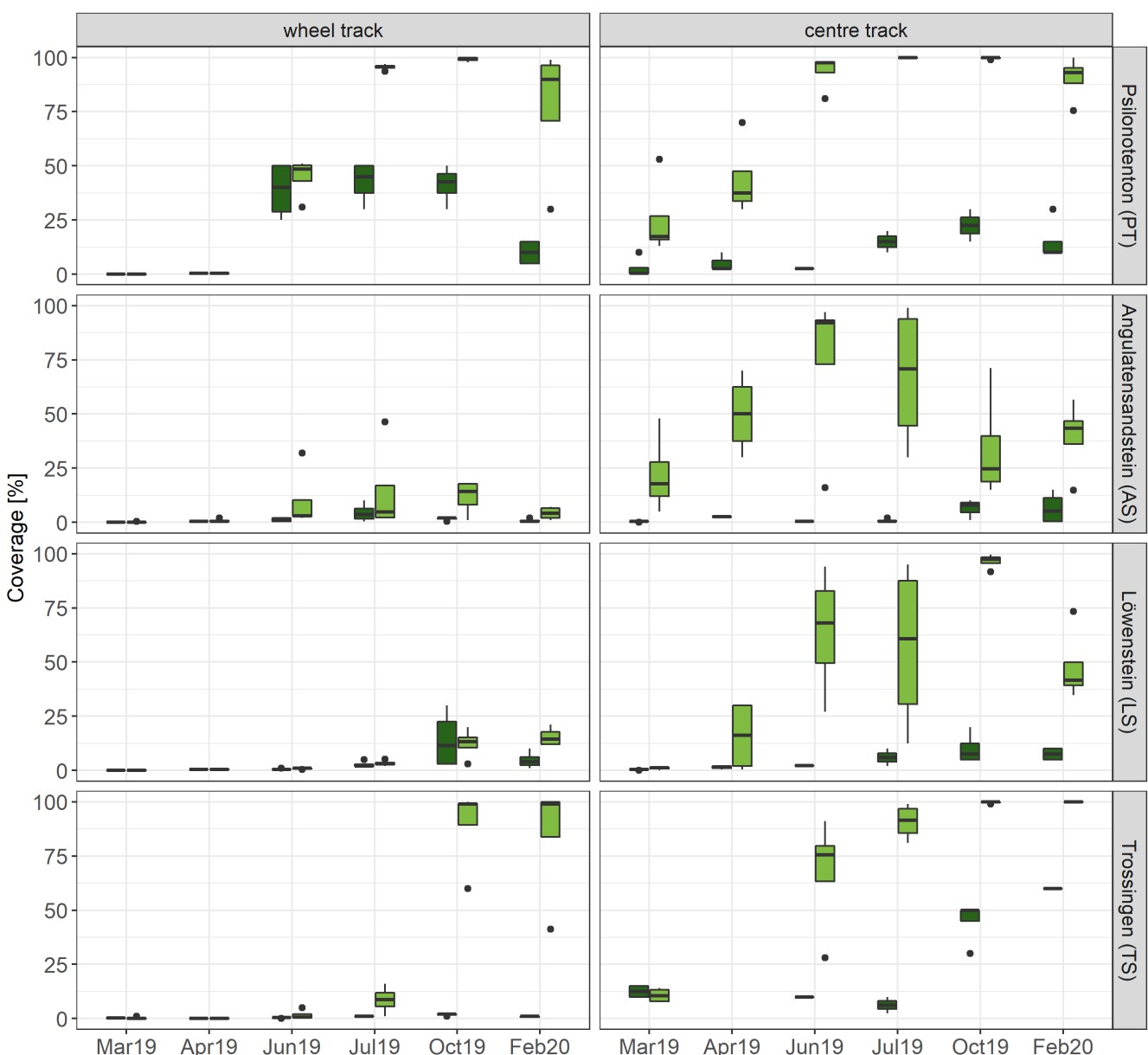

**Figure 3: Development of bryophyte (n = 4) and total vegetation coverage (n = 4) per runoff plot at the individual skid trails. The bottom and top of the box represent the first and third quartiles, and whiskers extend up to 1.5 times the interquartile range (IQR) of the data. Outliers are defined as more than 1.5 times the IQR and are displayed as dots.**


In AS and LS, total vegetation coverage was lower than in PT (p < 0.001), which was also the case for bryophyte cover (for

AS and PT: $p < 0.001$; for LS and PT: $p < 0.01$). In comparison, PT and TS were rapidly overgrown by vascular plants; however, they did not displace bryophytes (see Figure 3). This coexistence of vascular plants and bryophytes was also displayed in a positive correlation between their coverage rates (Spearman's correlation rho = 0.38, $p < 0.001$). Nevertheless,

the overgrowth of bryophytes by vascular plants also marks the transition from biocrust to an evolved successional stage of bryophyte cover, characterized by a large proportion of the biomass being above the soil surface (Belnap et al., 2003). While closed vegetation cover developed in PT and TS until autumn in both centre and wheel tracks, no continuous pattern of growth emerged in AS and LS, with clear differences between centre and wheel tracks. AS and LS developed a very sparse total vegetation cover in wheel tracks (about 5%), and revealed considerably higher coverage in centre tracks.

Biocrusts reached a more developed successional stage as bryophyte cover when they were overgrown by vascular plants. This bryophyte cover could be established even with high total vegetation cover, which contradicts observations that vascular plants limit bryophyte growth in different ecosystems (Bergamini et al., 2001; Fojcik et al., 2019; Corbin and Thiet, 2020). For instance, Fojcik et al. (2019) found a negative relationship between bryophyte cover and the coverage of vascular plants in a temperate forest ecosystem, which they attributed to competition between bryophytes and vascular plants. Bergamini et al.

(2001) also discovered such a negative relationship and explained it primarily in terms of light availability, with a combination of optimal radiation and moisture conditions depending on the extent of vascular plant cover. In contrast, Ingerpuu et al. (2005) verified in a grassland experiment that vascular plants could actually facilitate bryophyte growth, explaining this by the fact that vascular plants create a more favourable microclimate under their canopy. Likewise, positive correlations between vascular plants and bryophyte cover have been reported for temperate forests, which are comparable to our results (Márialigeti

et al., 2009; Rola et al., 2021). According to Rola et al. (2021), this relationship can be explained by the species composition, e.g. expansive grasses and sedges could easily eliminate bryophytes (Chmura and Sierka, 2007), and a relatively low vascular plant cover. A decline in bryophyte cover was observed for the first time in autumn on deciduous forest sites, but not on coniferous sites. For this reason, we assume that bryophyte growth in our study area was limited by leaf litter fall rather than suppression by vascular plants. A negative effect of leaf litter was also reported in several other studies (Márialigeti et al.,

2009; Fojcik et al., 2019; Mercier et al., 2019; Alatalo et al., 2020; Wu et al., 2020).

### 3.2.2 Bryophyte and vascular plant species richness

Regarding bryophyte and vascular plant species richness, we observed that a greater number of vascular plant species occurred in centre tracks (9.85 ± 0.59) than in wheel tracks (4.85 ± 0.53; $p < 0.001$), while no significant difference between tracks was found for bryophyte species richness (Figure 4). Furthermore, species richness varied in the skid trails: PT and LS showed, on average, considerably higher numbers of bryophyte species compared with AS and TS ($p < 0.01$). Concerning vascular plants, the highest species richness was achieved in PT, which was significantly higher than in AS and TS but not much higher than in LS. In comparison, AS, TS, and LS exhibited no differences in vascular plant species richness. While bryophyte species richness was positively correlated with pH (Spearman's correlation rho = 0.40, $p < 0.001$) and negatively correlated with silt content (Spearman's correlation rho = −0.35, $p < 0.001$), we could not find any clear associations between the soil parameters surveyed and vascular plant species richness. A GAM was used to explain 70.9% of the deviation of bryophyte species richness, with pH ($p < 0.001$), bryophyte cover ($p < 0.001$), SOC ($p < 0.01$), and $N_t$ ($p < 0.01$) being significant.

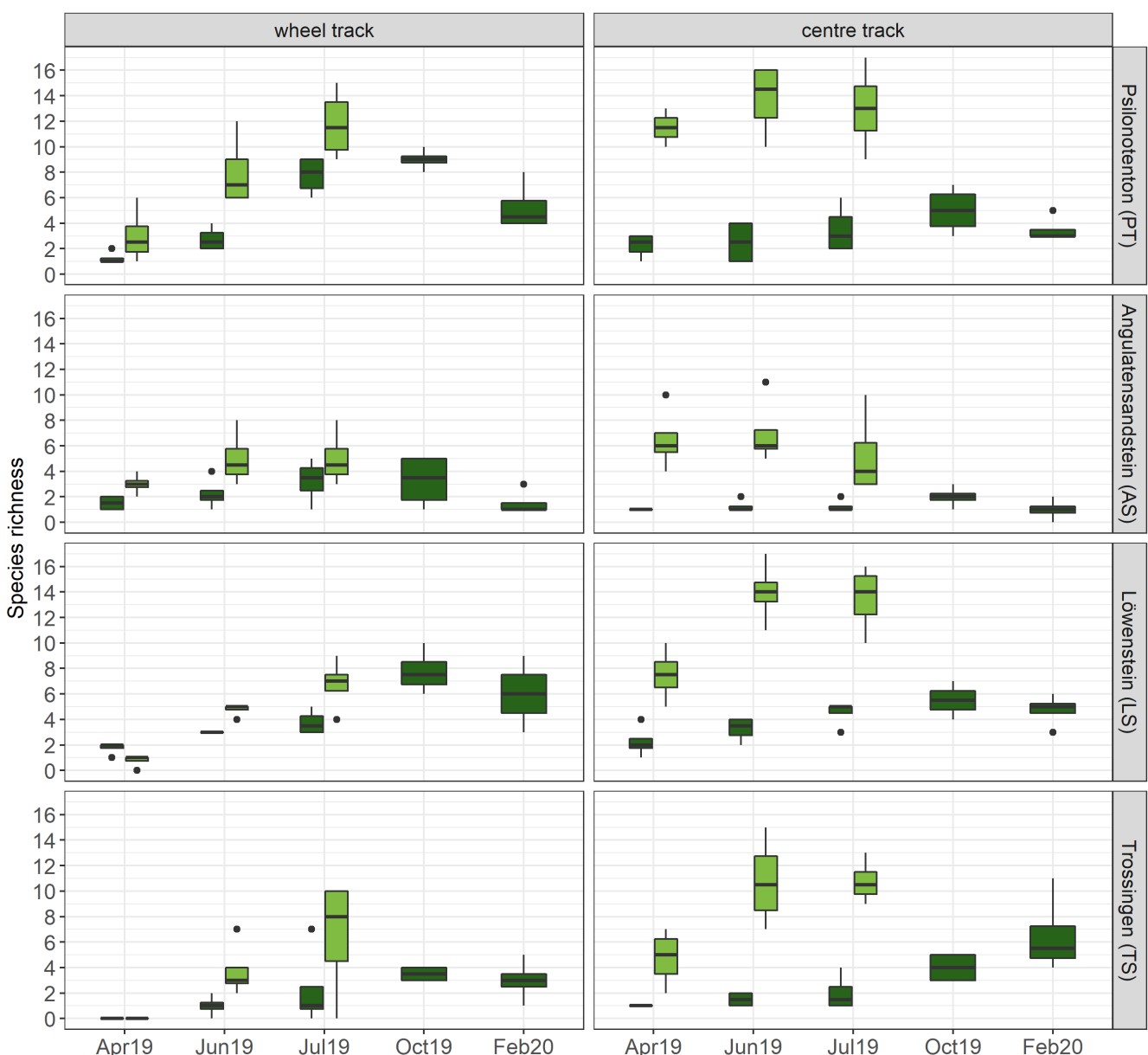

**Figure 4: Species richness of bryophytes (n = 4) and vascular plants (n = 4) per runoff plot at the individual skid trails. The bottom and top of the box represent the first and third quartiles, and whiskers extend up to 1.5 times the interquartile range (IQR) of the data. Outliers are defined as more than 1.5 times the IQR and are displayed as dots.**

Our results revealed that development of total vegetation cover was not only slower and less pronounced in wheel tracks, but

also that fewer vascular plant species could colonize there. Contrary to our expectations, bryophyte species richness was not affected by track position. In this context, Müller et al. (2013) found that experimentally induced disturbances had no impact on bryophyte species richness, whereas the diversity of annual plants benefited from disturbances. Minor disturbances, not exceeding 12% bare ground, could still promote bryophyte species richness, while further disturbance was detrimental. Additionally, Mercier et al. (2019) discovered that soil compaction in skid trails had a positive effect on the species richness of vascular plants, while bryophyte species richness was not affected. AS and LS, which showed particularly low levels of coverage and species richness, exhibited a different underlying substrate (sandstone) from the other two skid trails (claystone), which was also why we found different soil conditions there. Regional variations in species richness of vascular plants and bryophytes due to different soil conditions have also been confirmed in a variety of studies (Löbel et al., 2006; Klaus et al., 2013; Müller et al., 2013; Filibeck et al., 2019), with pH in particular proving to be an important positive control variable for bryophyte species richness (Hydbom et al., 2012; Oldén et al., 2016; Tyler et al., 2018). Additionally, Tyler et al. (2018) discovered a significant influence of substrate type, soil depth, and grazing intensity on overall bryophyte species richness, with pH remaining the most important factor in this study also. Further factors influencing bryophyte species richness, such as light availability, carbon-to-nitrogen ratio, and bark water capacity, were identified by Jagodziński et al. (2018) for 30-year-old reforested areas on lignite mining spoil heaps.

### 3.3 Soil erosion depending on site, track position, and vegetation cover

In total, mean sediment discharge in the wheel tracks reached 206.76 g m$^{-2}$ ± 24.53 and 15.68 g m$^{-2}$ ± 3.84 in the undisturbed forest soil ($p < 0.001$), while centre tracks caused a sediment loss of 63.09 g m$^{-2}$ ± 10.28, which was four times higher than the undisturbed forest soil ($p < 0.05$). Considering ROPs with bare soil separately, an average soil erosion of 341.53 g m$^{-2}$ ± 68.20 was achieved, which corresponds to a 22-fold enhancement compared with undisturbed forest soil. Additionally, sediment discharge in wheel tracks was increased by a factor of 3.3 compared with centre tracks. The main driver of sediment discharge was surface runoff (Spearman's correlation rho = 0.80, $p < 0.001$), and other important influencing soil characteristics were soil bulk density (Spearman's correlation rho = 0.50, $p < 0.001$), SOC and $N_t$ (both with Spearman's correlation rho = −0.46, $p < 0.001$), and MWD (Spearman's correlation rho = −0.46, $p < 0.001$). Additionally, a negative correlation between soil erosion and clay content was identified (Spearman's correlation rho = −0.42, $p < 0.001$), and

antecedent soil moisture and slope played a minor role in soil erosion. A GAM could explain 71.9% of the deviation of sediment discharge, with runoff ($p < 0.001$) and total vegetation cover ($p < 0.001$) being significant.

These results show that skid trails are a major contributor to soil erosion in forest ecosystems, and that compacted wheel tracks in particular significantly increased sediment discharge, which has also been demonstrated in previous studies (Safari et al., 2016; Zemke, 2016). In line with our results, Safari et al. (2016) highlighted soil texture, soil bulk density, SOC, and aggregate stability as the main soil parameters affecting runoff generation and soil erosion in skid trails. Based on these relationships, the significantly higher sediment discharge in skid trails is explained by the fact that the soil was disturbed and compacted by timber harvesting machines, especially in wheel tracks, such that infiltration is reduced, which in turn leads to higher surface runoff and sediment transport (Zemke et al., 2019).

For all skid trails, sediment discharge was, on average, highest in March 2019 with a mean value of 201.80 g m⁻² ± 39.82 and was considerably decreased in July 2019 to 74.13 g m⁻² ± 16.16 ($p < 0.01$). Subsequently, sediment discharge increased significantly in October 2019 (97.77 g m⁻² ± 21.16; $p < 0.05$) and rose again to 165.03 g m⁻² ± 29.75 in February 2020 ($p < 0.001$). Considering the time progression of soil erosion individually in the skid trails, different erosion mechanisms and sediment loads were evident (Figure 5). Average sediment discharge was highest in AS with 243.63 g m⁻² ± 37.30 and lowest in TS with 42.83 g m⁻² ± 10.34, which represented a difference of a factor of 5.7 ($p < 0.001$). While all skid trails differed from each other in terms of sediment discharge, no significant difference was detected between PT (151.62 g m⁻² ± 32.57) and LS (99.26 g m⁻² ± 15.76). With respect to the time progression of soil erosion in the skid trails, we found a difference between the measurement times for PT and LS, but not for AS and TS. In both cases, sediment discharge was significantly reduced from the bare soil condition in March 2019 to an early successional stage of biocrust and vascular plant vegetation in July 2019: PT showed a decrease of 89% and LS a reduction of 59%. The same pattern of soil erosion over the year was also observed in AS but could not be statistically demonstrated. While the correlation between surface runoff and sediment discharge was particularly high on average for the first rainfall simulation (Spearman's correlation rho = 0.89, $p < 0.001$), the influence was distinctly reduced in the other simulations and especially in October 2019 (Spearman's correlation rho = 0.51, $p < 0.01$). In the subsequent rainfall simulations, vegetation cover was an additional factor influencing soil erosion: The negative relationship between total vegetation cover and sediment discharge increased considerably from the first to the third simulation

in October 2019 (first simulation in March: Spearman's correlation rho = −0.45, p < 0.01; third simulation in October: Spearman's correlation rho = -0.86, p < 0.001), and the highest reduction of sediment discharge occurred in July 2019.

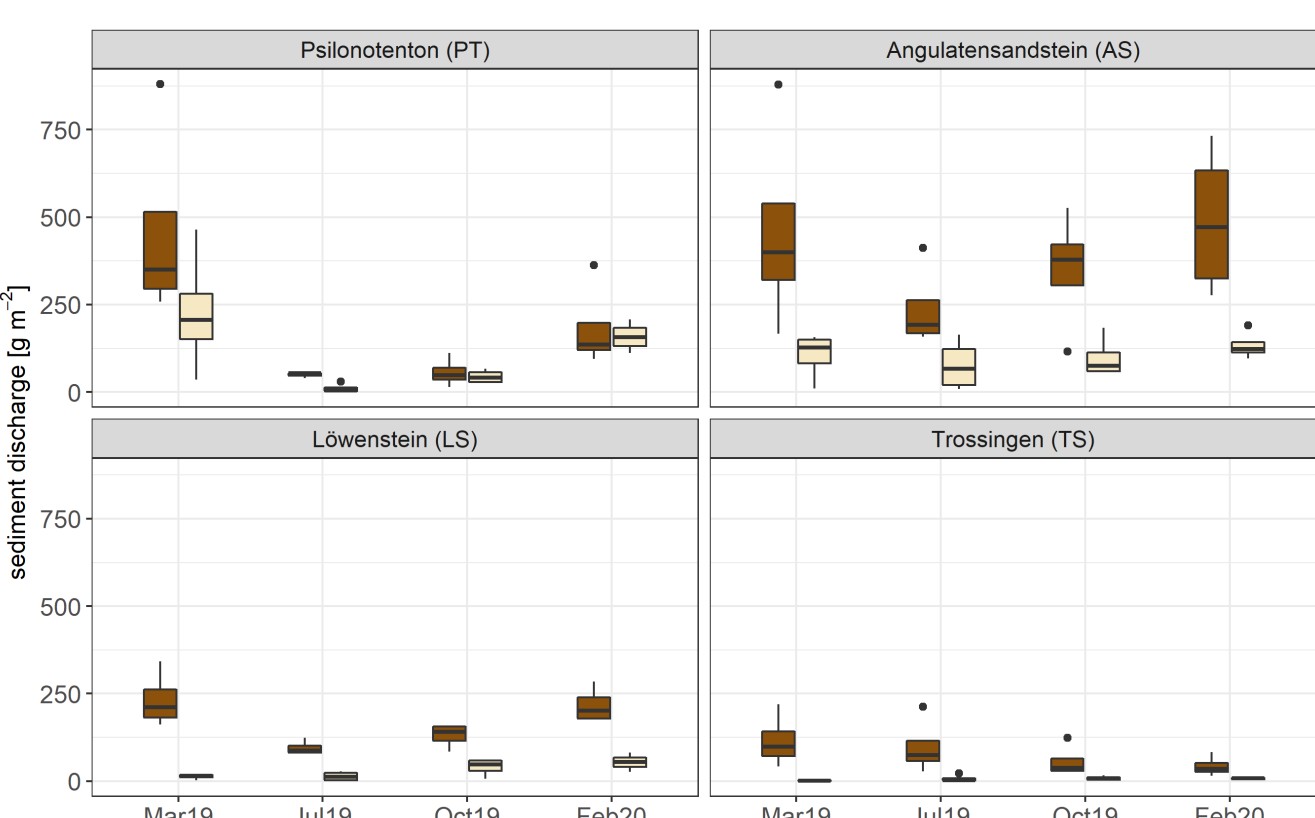

Figure 5: Sediment discharge during simulated rainfall in the wheel track (n = 4) and centre track (n = 4) of the four skid trails for every rainfall simulation time. The bottom and top of the box represent the first and third quartiles, and whiskers extend up to 1.5 times the interquartile range (IQR) of the data. Outliers are defined as more than 1.5 times the IQR and are displayed as dots.

Overall, the amount of discharged sediment clearly depended on the particular site, likely indicating an important effect of parent material on soil properties and adjunct vegetation development, and, thus, on soil erosion. A high influence of parent material on soil erosion was confirmed by Rodrigo-Comino et al. (2018). Regardless of the amount of sediment discharge, three skid trails showed comparable trends in soil erosion over time: In general, soil erosion was highest on bare soil; was reduced during the vegetation period, most with pioneer vegetation in July 2019, where biocrusts predominated; and then increased again in winter. This general trend was not observed in TS, which is probably related to the ecological structure of

TS, since it was the only skid trail located in a clearing and was therefore clearly distinguished from the other skid trails in

terms of vegetation succession. In addition, forest residues, such as bark, small branches, and needles were added to the topsoil

in TS as a result of forestry use, which also had a stabilizing effect and certainly contributed to the low sediment discharge in

this skid trail. The erosion-reducing effect of these types of mulching with forest residues has already been demonstrated in

various studies (Prats et al., 2016; Prosdocimi et al., 2016), and Vinson et al. (2017) recently demonstrated that mulching

strategies could also significantly reduce erosion rates in skid trails.

Several erosion studies on skid trails have already emphasized vegetation cover as one of the key control variables of soil

erosion (Zemke, 2016; Malvar et al., 2017; McEachran et al., 2018). Soil erosion was often observed to be highest in the first

year after skidding and decreased thereafter with increasing vegetation cover (Baharuddin et al., 1995; Jourgholami et al.,

2017; Malvar et al., 2017). Martínez-Zavala et al. (2008) also reported a seasonality in their erosion measurements on forest

road backslopes in southern Spain, with higher soil loss rates in winter despite vegetation cover, primarily attributed to higher

soil moisture. However, they further found that this seasonal effect did not occur above a vegetation cover of 30%. Thus, we

hypothesize that, among other factors, higher soil moisture may have influenced increased winter soil erosion in our case as

well, although we have not found significant correlations to support this theory.

### 3.4 Influence of bryophyte cover and early successional bryophyte-dominated biocrusts on soil erosion

Sediment discharge was distinctly negatively affected by total vegetation cover (Spearman's correlation rho = −0.61, p <

0.001). Furthermore, we discovered a stronger negative correlation between bryophyte cover and sediment discharge

(Spearman's correlation rho = −0.54, p < 0.001) than between vascular plant cover and sediment discharge (Spearman's

correlation rho = −0.36, p < 0.001). For these correlations, all undisturbed forest soil ROPs that were covered with leaf litter

were extracted because we assume that litter-covered soils have a different protective mechanism than soils with bryophytes

or vascular plants (Silva et al., 2019; Wang et al., 2020). All cover classes differed significantly from each other in terms of

sediment discharge, with a reduction of 77% being observed between bare ROPs and bryophyte ROPs (p < 0.001) and a

reduction of 59% being observed between bare ROPs and vascular plant ROPs (p < 0.005). Bryophyte ROPs showed 44% less

sediment discharge than vascular plant ROPs (p < 0.05). When ROPs were categorized into different cover classes, there was

a nonsignificant trend for bryophytes to result in less sediment discharge compared with vascular plants (Figure 6). Especially

with a cover of more than 50%, the erosion-reducing effect of bryophytes was more pronounced compared with vascular

plants; for example, the mean sediment discharge of bryophyte ROPs was 3.27 g m$^{-2}$ ± 1.50, while vascular plant ROPs still

reached an average of 57.82 g m$^{-2}$ ± 12.47, an 18-fold difference.

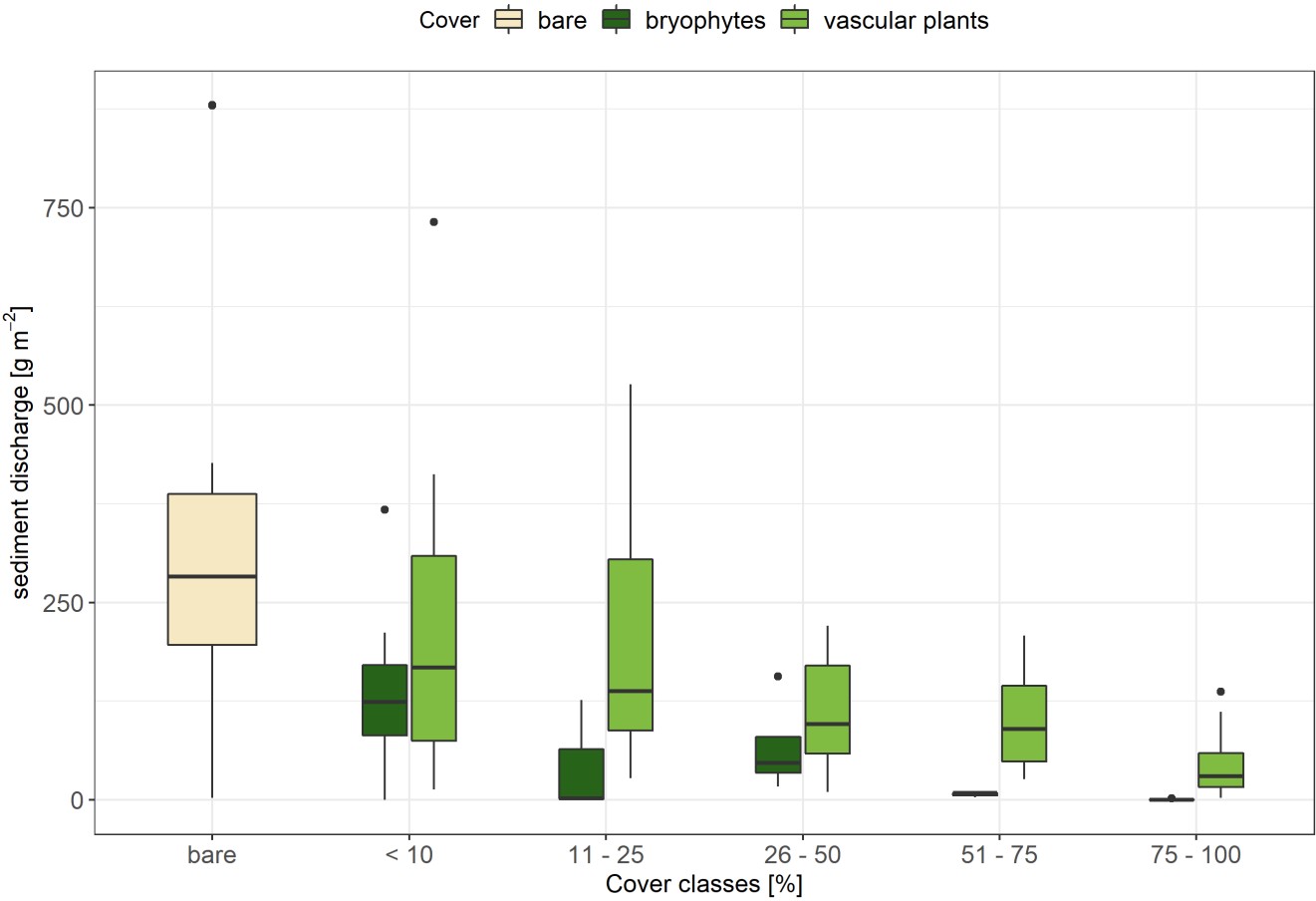

**Figure 6: Sediment discharge during simulated rainfall for bare (n = 14), bryophyte (n = 27) and vascular plant (n = 58) runoff plots (ROPs) categorized into cover classes. The bottom and top of the box represent the first and third quartiles, and whiskers extend up**
**to 1.5 times the interquartile range (IQR) of the data. Outliers are defined as more than 1.5 times the IQR and are displayed as dots.**

Bryophyte covers in temperate forest are known to stabilize soil surfaces and, thus, act as a protective agent against soil erosion

(Mägdefrau and Wutz, 1951; Belnap and Büdel, 2016; Seitz et al., 2017). The same applies to covers of vascular plants (Zuazo

Durán and Rodríguez Pleguezuelo, 2009); however, it is assumed that bryophyte communities have a stronger erosion-reducing

effect than vascular plants (Casermeiro et al., 2004; Bu et al., 2015) due to their large water absorption capacity (Thielen et

al., 2021) and the soil stabilizing effect of their rhizoids (Mitchell et al., 2016). In this context, the biocrust characteristics

demonstrated in this study at the initial successional stage, with communities of bryophytes, their protonemata, cyanobacteria and algae, for example, seeming to further enhance the erosion-reducing effect. Thus, the erosion-reducing effect appears to be stronger than that of communities dominated by vascular plants in the later stages (Figure 5); this might be due to a combination of different complementary plant traits. Likewise, Seitz et al. (2017) attributed a positive effect to bryophyte

protonemata in erosion control in mesic ecosystems. On the Loess Plateau in China, Bu et al. (2015) found that bryophyte-dominated biocrusts achieved a reduction in soil erosion of 81% compared with bare soil, while a mixture of vascular plants and bryophytes contributed significantly less to erosion control (a 0.7%−0.3% reduction depending on plant species). Furthermore, Casermeiro et al. (2004) discovered during rainfall simulations in Spain that scrubs are more effective at mitigating soil erosion when they are underlain with a cover of bryophytes. However, contrasting results were reported for a

very specific setup by Parsakhoo et al. (2012), who found that bryophyte-covered ROPs produced more sediment than ROPs with *Rubus hyrcanus*. Thus, there are still a number of unresolved questions regarding bryophyte−soil interactions on aspects such as water absorption, storage, and therefore erosion processes (Thielen et al., 2021), as well as on the development of biocrusts in mesic and forested areas, which need to be tackled in future research.

**4 Conclusions**

This study examined the initial development of pioneer nonvascular and vascular plant cover, composition, and species richness in temperate forest disturbance zones and their influence on soil erosion. Regarding our hypotheses, the following conclusions were drawn:

(1)    The succession of bryophytes and their composition varied at every site. Generally, *Brachythecium rutabulum* and *Oxyrrhynchium hians* were the most important and persistent pioneer bryophyte species, while *Dicranella schreberiana*

490          and *Pohlia lutescens* formed covers that quickly disappeared after spreading in summer. Biocrust communities occurred immediately after disturbance from April to July 2019, consisting primarily of bryophyte protonemata, cyanobacteria as well as coccoid and filamentous algae.

(2)    Skid trails on clayey substrates showed considerably higher total vegetation cover and species richness, which applied to bryophytes and vascular plants. While vascular plants were more abundant in centre tracks than wheel tracks in terms of

495          both cover and species richness, there was no clear difference in bryophyte species richness in this regard. Although

bryophytes were quickly overtopped by vascular plants during vegetation succession, they managed to coexist until the end of the vegetation period and were then limited, most likely due to leaf litter fall.

(3) The total amount of sediment discharge and the general mechanisms of soil erosion were clearly site dependent. Soil erosion was reduced, especially with the occurrence of pioneer biocrust vegetation in summer, and again increased in winter, when vascular vegetation became dominant. Sediment discharge was 13.2 times higher in wheel tracks than in undisturbed forest soil, and bare soil ROPs produced a 22-fold greater sediment discharge than undisturbed forest soil.

(4) Bryophytes made a major contribution to erosion control after disturbances in this temperate forest ecosystem. They contributed more to mitigating soil erosion than vascular plants. Since soil erosion was especially low when bryophytes occurred within biocrusts, we assume that bryophyte-dominated biocrusts, in particular, are of utmost importance for preventing soil degradation, even in mesic environments.

Based on these results, artificial inoculation of bryophytes as erosion control on bare forest soils is assumed to be of particular interest for future research. In this context, Varela et al. (2021) recently published an approach to establish moss cultures from the laboratory, which could be applied for environmental studies. Moreover, the question arises whether bryophytes reduce soil erosion primarily through their protective-layer effect on splash and runoff, or whether they also improve soil properties, such as aggregate stability, which further enhance erosion control (Riveras-Muñoz et al., 2022). Within this framework, it continues to be of special interest whether there are different mechanisms of erosion control depending on particular bryophyte species and which of their structural traits affect soil erosion patterns the most.

**Appendix**

**Table A1**: Characteristics of studied skid trails.

| | AS | PT | LS | TS |
|---|---|---|---|---|
| Series | Lower Jurassic | Lower Jurassic | Upper Triassic | Upper Triassic |
| Formation | Angulatensandstein (AS) | Psilonotenton (PT) | Löwenstein (LS) | Trossingen (TS) |
| Parent material | sandstone | shale clay | sandstone | claystone |
| Soil type (Ad-hoc-AG Boden, 2005) | Braunerde-Pseudogley | Pseudogley | Braunerde-Pelosol | Braunerde-Pelosol |
| Soil type (IUSS Working Group WRB, 2015) | Dystric Leptosol (Ochric, Siltic, Stagnic) | Calcaric Albic Planosol (Clayic, Ochric, Raptic) | Calcaric Cambisol (Humic, Loamic, Protovertic) | Eutric Cambisol (Geoabruptic, Clayic, Ochric, Protovertic) |
| Soil texture | silt loam<br>• sand: 6.89%<br>• silt: 67.99%<br>• clay: 25.33% | silty clay loam<br>• sand: 6.67%<br>• silt: 56.49%<br>• clay: 36.86% | clay loam<br>• sand: 25.91%<br>• silt: 40.78%<br>• clay: 33.20% | silty clay loam<br>• sand: 11.46%<br>• silt: 50.70%<br>• clay: 37.81% |
| SOC | 4.08% | 5.22% | 5.52% | 7.95% |
| $N_t$ | 0.24% | 0.31% | 0.27% | 0.40% |
| C/N | 17 | 17 | 21 | 19 |
| $pH_{Ca}$ | 5.6 | 6.9 | 6.9 | 5.4 |
| Slope | 4.6° | 7.2° | 10° | 11.3° |
| Aspect | Southwest | South | West | Northwest |
| Sample site coordinates | Tübingen<br>48.553054 N<br>9.119053 E | Tübingen<br>48.557425 N<br>9.114462 E | Tübingen<br>48.557527 N<br>9.088098 E | Tübingen<br>48.556036 N<br>9.089313 E |

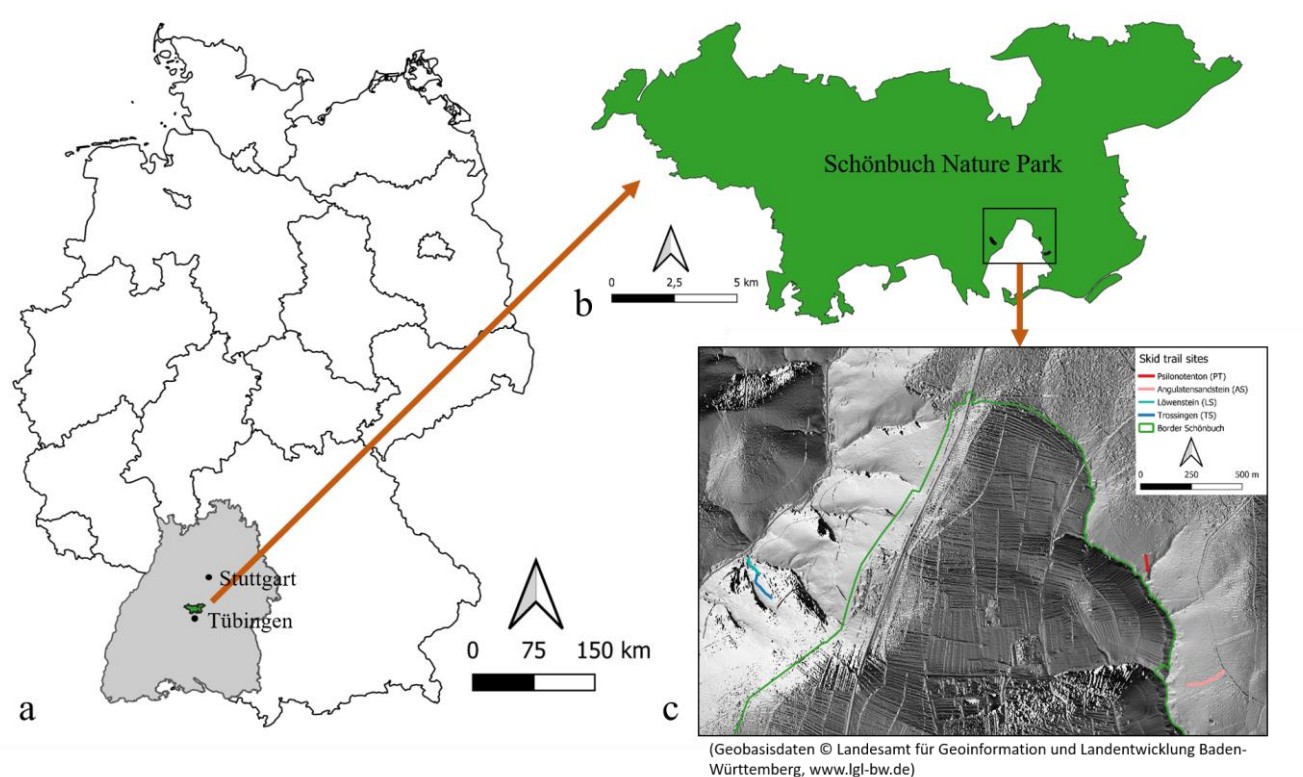

**Figure A1:** Overview of the study area: a) Location of Schönbuch Nature Park in Germany, b) Location of the selected skid trails inside Schönbuch Nature Park, c) Location of the four skid trails on a hillshade raster (Geobasisdaten © Landesamt für Geoinformation und Landentwicklung Baden-Württemberg, www.lgl-bw.de)

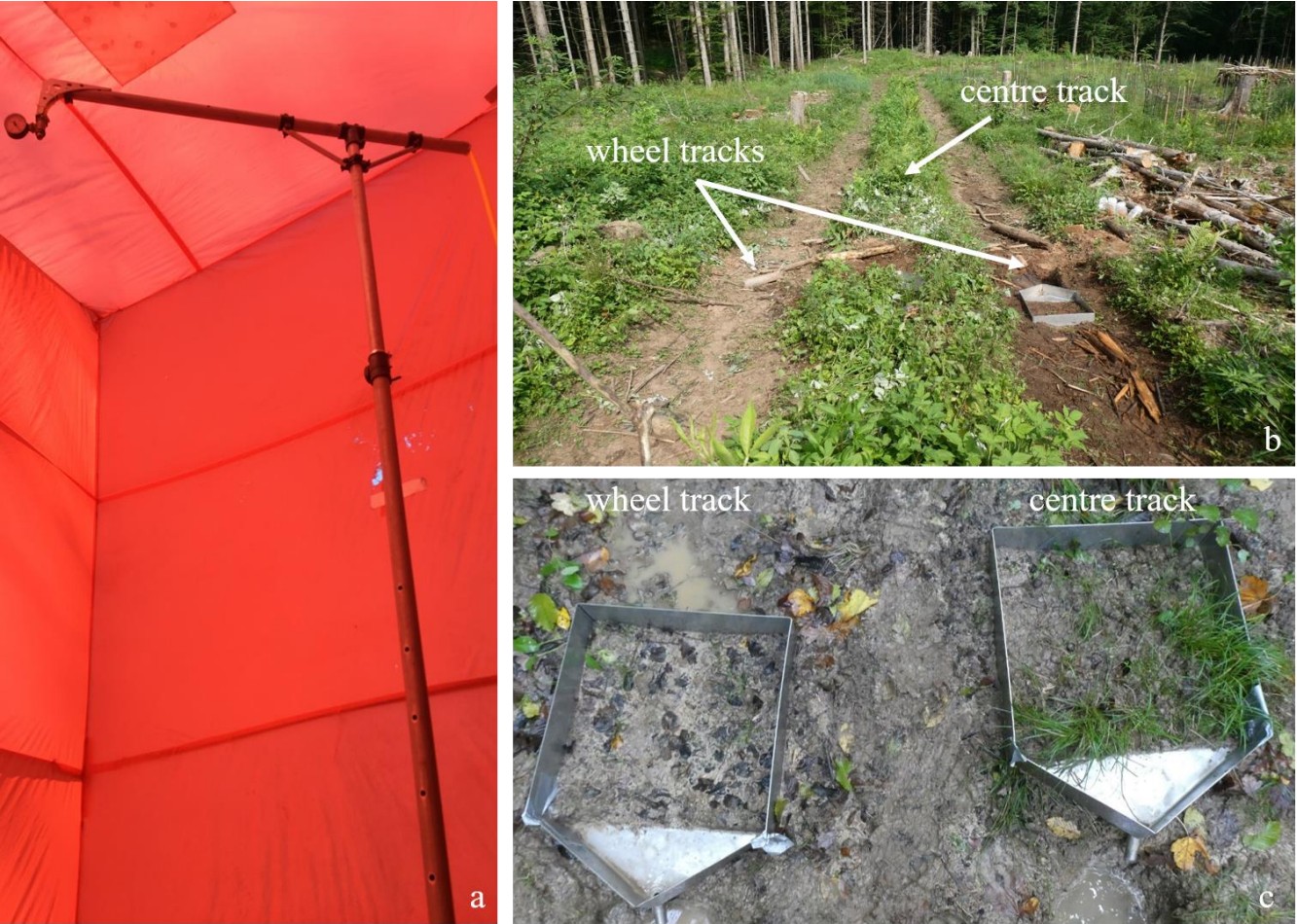

**Figure A2:** Experimental setup: a) Tübingen rainfall simulator inside the protective tent, b) Skid trail in the Trossingen
Formation (TS) in July 2019, c) Runoff plots in the wheel track and the centre track in the Angulatensandstein Formation
(AS) in October 2019

## Code availability

The codes used in this study are available upon request.

## Data availability

The dataset compiled and analysed in this study is available on figshare at https://doi.org/10.6084/m9.figshare.17206835.v3
(Gall et al., 2021).

**Author contribution**

StS, TS, DQ and MN designed the experiment. CG and StS carried out field measurements and CG was responsible for laboratory and data analyses. MN and CG conducted the vegetation surveys. CG and StS prepared the manuscript with contributions from all other co-authors.

**Competing interests**

The authors declare that they have no conflict of interest.

**Acknowledgements**

This research would not have been possible without the exceptional support of Michael Sauer during vegetation surveys. His enthusiasm and knowledge of moss identification was an inspiration and we thank him for his great contribution to this work. We would also like to express our sincere gratitude to Laura Bindereif, Lena Grabherr, Helena Obermeier, Philipp Gries, Delia Maas, Stefanie Gotterbarm, Giulia König, Nicolás Riveras-Muñoz, Sascha Scherer, Matthew A. Bowker, Daniel Schwindt and all students of GEO51 in winter semester 2019/20 for their help with field and lab work. Many thanks also to Sabine Flaiz, Rita Mögenburg and Peter Kühn for always being available to help us with questions regarding lab work. Furthermore, we thank the state forestry service of Baden-Württemberg (ForstBW) for supporting the MesiCrust project. We are grateful to Bettina Weber, further two anonymous reviewers and the students and tutor of the course "Critical Thinking in Ecological and Environmental Sciences" at the University of Edinburgh for a very constructive and helpful review.

**Financial Support**

This study was funded by the German Research Foundation (DFG SE 2767/2-1 "MesiCrust").

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
