# Peer review of "Pioneer biocrust communities prevent soil erosion in temperate forests after disturbances"

_Biogeosciences, 2021_

## Community Comment (CC1)

Comment on the under-review manuscript "**Pioneer biocrust communities prevent soil erosion in temperate forests after disturbances**" by:

Attapun Anivat*, Bella O'Hara*, Emily Tanner*, Ilja Belovolovs*, Laura Barraclough*, Georgios Kazanidis**

*Undergraduate student in the course "Critical Thinking in Ecological and Environmental Sciences" at the University of Edinburgh

**Tutor in in the course "Critical Thinking in Ecological and Environmental Sciences" at the University of Edinburgh

**Dear authors,**

as part of the undergraduate course "Critical Thinking in Ecological and Environmental Sciences" at the University of Edinburgh we have read carefully the mentioned above manuscript and we would like to express here our thoughts. We have found this piece of work is, in overall, a timely and interesting manuscript and we hope that our thoughts will help the authors to improve the status of their under-review paper.

**Overall Summary**
This is a very interesting topic. The study is, in overall, well organized and executed. The manuscript is also in good order and there is clear presentation of results and appropriate use of English language. The 'Introduction' could be shorter with less detail; it should focus to attract more the interest of readers why this research must be done, and what are they key knowledge gaps that it addresses. Occasionally there is too technical terminology used which makes understanding challenging for the non-experts (see detailed comments below). It seems that some major gaps exist in terms of statistical analysis and how the authors have examined the role of environmental parameters in shaping the floral communities. There should be better use of references in 'Materials and Methods' and better elaboration in parts of the 'Discussion (e.g., in parts of Section 3.1 - see detailed comments below). A final major point is that the authors really need to highlight the key findings of their study – currently it seems that their major finding about the superior role of biocrust communities in preventing erosion is already mentioned in the literature. The innovative aspects of the study need to be made clear early in the manuscript and in the 'Abstract'.

**Abstract**

In overall it is well written. We suggest the authors make clearer why is it important to study what they have studied e.g. why is the succession so important? In addition, the final 2-3 sentences of the 'Abstract' need some modifications – making them simpler and easier to understand will increase their impact. It might be preferable to avoid use of "we" in the abstract and perhaps the detail of results could be reduced; authors might also be more clear in highlighting one main conclusion to express.

**Introduction**
Line 35: Please provide some examples why soil erosion will increase through climate change. Also are there any (numerical) projections about how much erosion will increase in years and decades to come?

Line 41-42: Is there a reason behind these relatively large shifts in erosion of forestlands?

Line 44: Please use more plain language in "showed that unsealed forest roads at the catchment scale" so that the reader can get a clearer understanding.

Lines 46-53: This numerical information provided is useful, but we feel it would be better to be used in the "Discussion". Here in the "Introduction" make sure you present the bigger picture and why it is important for this research to be carried out. Lots of numerical information can distract the readers from the major messages.

The sentence on line 55 could be modified to summarise the point of referencing all of these studies and then group them together in the citation for reference

Line 61: Please explain where the term "cryptogamic" refers to. Also, what do you mean by "understory"?

Line 63: Perhaps replace "edaphic" by "floor"?

Line 68: We feel this should be "bryophyte-dominated"?

Lines 68-70: Please provide briefly some information on the direction of these effects by bryophytes e.g. increase/decrease in runoff etc.

Line 81: Please improve wording.

Line 86: The authors need to make clearer which is the research gap and especially to link it better with previous lines/sections.

Lines 92-94. It is welcome that authors make clear the objectives of their study. We feel though that it would be even better if they make some null hypotheses related to their points e.g., how do they expect that the underlying substrate, vegetation cover and track position will affect soil erosion?

Line 96: Please explain what you mean by "interrill".

In the "Introduction" and especially towards the end of it the authors should make some clearer references on how their findings can be used in good practices for management. They can elaborate on that aspect in the Discussion.

**Materials and Methods**

Line 121 and further: Could abbreviate genus name in species scientific names for conciseness purposes (e.g. *P. sylvestris*)

Lines 140-146: Please provide references about the use of similar experimental set up in previous studies.

Lines 148-149: The authors need to provide more information about the particular selection of this rainfall intensity e.g., is similar intensities observed often in the studied area? Provide also relevant references.

Line 149- 153: The authors should provide more details about technical aspects mention in there e.g., measurements on surface run off. Please also provide references.

Line 154-155: For how long were the samples left to dry?

Line 156: Please mention what is exactly the aggregate size and which are the measurement units for this parameter.

Line 159-162: It is interesting that measurements on elements (C, N) were made. Please make sure that there are the relevant references made in the "Introduction" so the sections of the manuscript align better.

Line 173. Please improve the wording about nomenclature in Tables.

Lines 183-187. It seems that post-hoc tests were not carried out. Also, it seems that the role of environmental parameters in the flora structure / development has not been accounted/examined for. If this is the case, then it is regarded as a major gap and needs to be addressed.

More information on the number of replicates is needed.

A map showing where the research was carried out would be welcome.

Overall, we believe that the "Materials and Methods" section could have been written more succinctly to make it easier to read.

**Results and Discussion**

Line 191: 'Section 3.1.1 – Biocrust species composition'. It seems that this title is not fully adequate as in the section 3.1.1 there are also results about temporal trends. This should be reflected in the Section 3.1.1 title.

Line 193: Please avoid using where possible abbreviations (e.g., 'UF') as it is difficult for the reader to follow them.

Line 196: Please clarify what is 'protonema'.

Line 205 / Table 1: Could table 1 provide more information on composition, cover and richness? Do we need Author column?

Lines 222-223: This is just an assumption on the role of pH; there should be appropriate statistical analysis to explore the role of abiotic environmental parameters in shaping the communities.

Tables 1 and 2: The information shown here is interesting; however it seems that these Tables are a bit long – how about moving them to Supplementary Material?

Lines 227-230: These are major findings and should be moved earlier/up in the Results and Discussion section.

Lines 232: Please clarify the categories that the species belong to e.g. do they belong to 'protonema' or another category?

Line 234: "little importance": Please provide numbers rather than terms like "little importance".

Lines 227-242: This is a big chunk of results but discussion on them is absent.

Line 243: It would be better to start the section with the key result; discussion on it should follow.

Line 246: Please see comments above about stats regarding the role of environmental parameters.

Figure 1: Could be useful to have included a longer caption describing what photographs demonstrate to make the article more accessible for the readers that do preliminary paper skimming. A map of the area would have been highly beneficial for the readers to better visualise the studied site spatial distribution.

Line 271: It is not clear what the authors try to say here e.g. that there are similar trends between biocrust and total coverage trends? Or something else? Please clarify.

Figure 2 caption: Perhaps it would read better as "mean values and standard error are given". Please also remind to the readers the number of replicates.

Line 282: The values of pH should be mentioned.

Lines 288-289: The authors should elaborate on their statements about contradictions between their findings and those from (Corbin and Thiet, 2020; Bergamini et al., 2001; Fojcik et al., 2019).

Lines 289-292: The authors should elaborate on the mechanisms driving positive correlations between vascular plants and moss growth.

Line 292: The statements/discussion on biocrust should be on a separate paragraph.

Lines 327 – 338: Please make sure that you provide p-values where needed. Also, it is not necessary to use extensively phrases such as "A was X times higher than B". Providing the average values, standard error and the p-values would suffice.

Lines 339-341: See our comments above about examining the role of environmental parameters in shaping discharge / run off. For example, how much of the variability in discharge is explained by differences in the soil features?

Sections 3.2.1 and 3.2.2 should be merged. The independent and response variables should be subject to appropriate statistical analysis e.g. distance-based linear modelling (Clarke and Gorley 2015)

Clarke KR, Gorley RN (2015) PRIMER v7: User Manual/Tutorial PRIMER-E: Plymouth

Lines 398-401: Some of the lines mentioned here should had been included in the Materials and Methods. Also it is not clear where the term 'reduction' refers to – please clarify.

Figure 5: The box plots for biocrusts and vascular plants are very close (this is not necessarily bad) and some of the outliers for biocrusts may be regarded as outliers for vascular plants (and vice versa). It would be helpful to see the outliers for each of them with different colours. We feel that a sudden change in the colour scheme on this graph could confuse the readers that got used to seeing dark green as 'wheel track' and light green as 'central track' in previous 3 figures.

Line 425 Conclusions

Line 426 : it seems that null hypotheses were not made; it is suggested to adjust accordingly the text at the end of the "Introduction".

The conclusions section looks too lengthy; it should appear more succinct and with higher impact. Focus on your key findings and how they fill gaps in the literature. Avoid repeating results and numerical values.

Could include more discussion of direction and opportunities for future studies

Lines 450- 456: Would it be also of interest to study the factors that support higher growth rates for the biocrust communities?

Appendix

Figure A1. Please clarify in the image (using arrows) the wheel track and center track.

It seems that there is some inconsistency in editing/coloring of symbols across the figures e.g., see color code used Figures 3 and 4.

---

## Author Comment (AC3)

**Response to Community Comment 1 (CC1) on preprint bg-2021-343: "Pioneer biocrust communities prevent soil erosion in temperate forests after disturbances"**

Thank you again for selecting our study for discussion in your undergraduate course. We were very pleased that you have dealt with our manuscript in such detail and your feedback has contributed significantly to the improvement of our manuscript.

| Comments | Authors responses |
|---|---|
| **Abstract**
*"In overall it is well written. We suggest the authors make clearer why is it important to study what they have studied e.g. why is the succession so important?"* | To date, very little is known about when soil-protective vegetation begins to develop in a forest disturbance area, so that it is important to monitor the process of succession. Furthermore, we wanted to determine the timing of biocrust occurrence and its impact on soil erosion, which is generally poorly studied in temperate climates. We included a sentence in the abstract that highlights the importance of vegetation succession to our study. |
| *"In addition, the final 2-3 sentences of the 'Abstract' need some modifications – making them simpler and easier to understand will increase their impact. It might be preferable to avoid use of "we" in the abstract and perhaps the detail of results could be reduced; authors might also be more clear in highlighting one main conclusion to express."* | Thank you very much for this comment, as it helped to improve the presentation of results in the abstract. We reduced the results in the abstract to the most important points of the study, which increases comprehensibility for the reader.
Additionally, we now use the passive form exclusively in the abstract. |
| **Introduction**
*"Line 35: Please provide some examples why soil erosion will increase through climate change. Also are there any (numerical) projections about how much erosion will increase in years and decades to come?"* | In the context of climate change, increasing rainfall intensities are the key driver of soil erosion, as this enhances the erosive power of precipitation and thus the probability of soil losses. We added this example in the introduction.
In our opinion, further examples and explanations would lead too far at this point. Projections of soil erosion rates are clearly influenced by local conditions and the percentage increase varies widely. For example, Li & Fang 2016 concluded that 136 studies predict an increase in soil erosion rates in the future, with relative increases ranging from 1.2% to 1614%. (*Li, Z. & Fang, H. (2016): Impacts of climate change on water erosion: A review, Earth-Science Reviews, 163, 94-117, https://doi.org/10.1016/j.earscirev.2016.10.004*) |
| *"Line 41-42: Is there a reason behind these relatively large shifts in erosion of forestlands?"* | There are a variety of factors influencing soil erosion in forests, and it would be too much of a stretch to discuss them all at this point. In the references we highlighted here, the shifts in erosion rate referred mainly due to forest management intensity and tree species composition. We added these factors in in the mentioned lines of the introduction. |
| *"Line 44: Please use more plain language in "showed that unsealed forest roads at the catchment scale" so that the reader can get a clearer understanding."* | We simplified the sentence in the mentioned line. |

| | |
|---|---|
| *"Lines 46-53: This numerical information provided is useful, but we feel it would be better to be used in the "Discussion". Here in the "Introduction" make sure you present the bigger picture and why it is important for this research to be carried out. Lots of numerical information can distract the readers from the major messages."* | According to your recommendation, we reduced the number of numerical information in the introduction to avoid distractions from the overall context. |
| *"The sentence on line 55 could be modified to summarise the point of referencing all of these studies and then group them together in the citation for reference"* | We followed your comment and changed the structure of this sentence. |
| *"Line 61: Please explain where the term "cryptogamic" refers to. Also, what do you mean by "understory"?"* | We explained what we meant by "cryptogamic" in the abstract and in the introduction. This term includes all non-flowering plants and plant-like organisms that reproduce by spores, such as bryophytes, lichens, ferns, algae and fungi. The term "understory" was also specified in the introduction. By this we refer to the vegetation growing on the forest soil. |
| *"Line 63: Perhaps replace "edaphic" by "floor"?"* | We replaced "edaphic" by "soil" in this line. |
| *"Line 68: We feel this should be "bryophyte-dominated"?"* | Thank you for bringing this to our attention. We corrected this according to your comment. |
| *„Lines 68-70: Please provide briefly some information on the direction of these effects by bryophytes e.g. increase/decrease in runoff etc."* | We provided the direction of these effects in the mentioned lines. |
| *"Line 81: Please improve wording."* | We changed the order of the sentence to improve wording. |
| *"Line 86: The authors need to make clearer which is the research gap and especially to link it better with previous lines/sections."* | We followed your suggestion and clarified the research gaps in the introduction. |
| *"Lines 92-94. It is welcome that authors make clear the objectives of their study. We feel though that it would be even better if they make some null hypotheses related to their points e.g., how do they expect that the underlying substrate, vegetation cover and track position will affect soil erosion?"* | Thank you for noting this. We agree that null hypotheses improve the comprehensibility of the manuscript and implemented this wherever possible. |
| *"Line 96: Please explain what you mean by "interrill"."* | Soil erosion processes by water can be divided into three different stages: splash erosion, interrill and rill erosion. Interrill erosion is known as the discharge of sediment in thin sheets between rills by shallow surface runoff after raindrop impact. We added an explanation on that term in the introduction. |
| *"In the "Introduction" and especially towards the end of it the authors should make some clearer references on how their findings can be used in good practices for management. They can elaborate on that aspect in the discussion."* | According to your suggestion, we added a short outlook of good practices for forestry at the end of the introduction. |
| **Materials and Methods** *"Line 121 and further: Could abbreviate genus name in species scientific names for conciseness purposes (e.g. P. sylvestris)"* | For the second use of the scientific name of each species, we used an abbreviated genus name. |

| | |
|---|---|
| *"Lines 140-146: Please provide references about the use of similar experimental set up in previous studies."* | We provided more references about the use of rainfall simulators in combination with small-scale runoff plots. |
| *"Lines 148-149: The authors need to provide more information about the particular selection of this rainfall intensity e.g., is similar intensities observed often in the studied area? Provide also relevant references."* | We inserted more background information to the selected rainfall intensity with a reference. The rainfall intensity refers to a heavy rainfall event for this region that occurs less frequently than once every 100 years. |
| *"Line 149- 153: The authors should provide more details about technical aspects mention in there e.g., measurements on surface run off. Please also provide references."* | In our opinion, it is not necessary to add further details or references here, since the common procedure in soil erosion measurements is to collect surface runoff and the sediment discharged with it in sample bottles. References to this are already available in the previous section. |
| *"Line 154-155: For how long were the samples left to dry?"* | It usually took about three to four weeks until all samples were dry. But this depended strongly on the amount of water, which was different in each measurement. |
| *"Line 156: Please mention what is exactly the aggregate size and which are the measurement units for this parameter."* | Soil aggregate size is a basic parameter in soil science what we assume as basic knowledge and would not explain it in detail in the manuscript. Depending on a variety of biotic and abiotic factors, soil forms aggregates consisting of agglutinated soil particles. |
| *"Line 159-162: It is interesting that measurements on elements (C, N) were made. Please make sure that there are the relevant references made in the "Introduction" so the sections of the manuscript align better."* | According to your comment, we mentioned carbon and nitrogen levels in the introduction to point out the gap of knowledge of the factors that affect bryophyte species richness and cover. |
| *"Line 173. Please improve the wording about nomenclature in Tables."* | We improved the wording of this sentence. |
| *"Lines 183-187. It seems that post-hoc tests were not carried out. Also, it seems that the role of environmental parameters in the flora structure / development has not been accounted/examined for. If this is the case, then it is regarded as a major gap and needs to be addressed."* | We performed post-hoc Wilcoxon Rank-Sum tests and Wilcoxon Signed-rank tests for sediment discharge, surface runoff, coverage and species richness averaged for all skid trail sites and averaged for wheel and center tracks in each skid trail. On the level of individual skid trails, there are four replicates per track position, which is insufficient for performing post-hoc statistics. To assess the effect of environmental parameters on soil erosion, bryophyte coverage and species richness, we performed generalized additive models (GAM) with restricted maximum likelihood and smoothing parameters selected by an unbiased risk estimator (UBRE) criterion. |
| *"More information on the number of replicates is needed."* | We revised the information on the replicates in the method section so that the sample design is now more comprehensible. There are four replicates for each wheel track, four replicates for each center track and two replicates for each undisturbed forest soil. |
| *"A map showing where the research was carried out would be welcome."* | We have not provided a map to locate the research area because our study already contains a large number of figures and tables. |

| | |
|---|---|
| *"Overall, we believe that the "Materials and Methods" section could have been written more succinctly to make it easier to read."* | We will try to make the methods section more concise, for example by transferring more information of the study site description into table A1. |
| **Results and Discussion**
*"Line 191: 'Section 3.1.1 – Biocrust species composition'. It seems that this title is not fully adequate as in the section 3.1.1 there are also results about temporal trends. This should be reflected in the Section 3.1.1 title."* | We changed the title of Section 3.1.1. in "Succession of bryophyte species composition" to reflect the temporal trends included. |
| *"Line 193: Please avoid using where possible abbreviations (e.g., 'UF') as it is difficult for the reader to follow them."* | According to your comment, we decided to reduce the use of abbreviations in the text in order to improve readability. Therefore, we spelled out CT, WT, and UF in the revised manuscript. |
| *"Line 196: Please clarify what is 'protonema'."* | We introduced the term "protonema" in the abstract and later in the results and discussion section. Protonema is the earliest stage of bryophyte development consisting of green cell filaments. |
| *"Line 205 / Table 1: Could table 1 provide more information on composition, cover and richness? Do we need Author column?"* | We implemented your idea by providing additional information on the percentage occurrence of species in the runoff plots in total and for each vegetation survey time step. Furthermore, we added a diagram that illustrates the occurrence of bryophyte species in the ROPs for each vegetation survey time step in every skid trail site. This also includes more information about taxonomic composition and species richness at the different skid trail sites and considerably increased the comprehensibility of the results.
In the botanical nomenclature a reference to the person who first gave a name to the botanical entity is required and we followed these rules. Instead we discarded the family names. |
| *"Lines 222-223: This is just an assumption on the role of pH; there should be appropriate statistical analysis to explore the role of abiotic environmental parameters in shaping the communities."* | The effect of environmental parameters on species composition was not a focus of our study, so we made an assumption at this point that we did not support with statistical analysis. |
| *"Tables 1 and 2: The information shown here is interesting; however it seems that these Tables are a bit long – how about moving them to Supplementary Material?"* | With the additional information on the percentage occurrence of species in the runoff plots, we believe that table 1 and 2 should remain in the text. |
| *"Lines 227-230: These are major findings and should be moved earlier/up in the Results and Discussion section."* | In this section, we first wanted to give a general overview of the occurrence of bryophyte species in the research area (section 1) and discuss this species composition (section 2). Afterwards, section 3 deals with the different species compositions of the four skid trail sites, which is why more detailed results are listed there for the first time. |
| *"Lines 232: Please clarify the categories that the species belong to e.g. do they belong to 'protonema' or another category?"* | For protonema, we did not determine the species, so either the moss occurred as protonema or the species was mentioned. |

| | |
|---|---|
| *"Line 234: "little importance": Please provide numbers rather than terms like "little importance"."* | As recommended, we changed this wording and inserted numbers instead. |
| *"Lines 227-242: This is a big chunk of results but discussion on them is absent."* | The discussion of these results can be found in the following section. |
| *"Line 243: It would be better to start the section with the key result; discussion on it should follow."* | Based on your comment, we have restructured this section so that it is now more comprehensible and exciting for the reader. |
| *"Line 246: Please see comments above about stats regarding the role of environmental parameters."* | To assess the effect of environmental parameters on soil erosion, bryophyte coverage and species richness, we performed generalized additive models (GAM) with restricted maximum likelihood and smoothing parameters selected by an unbiased risk estimator (UBRE) criterion. |
| *"Figure 1: Could be useful to have included a longer caption describing what photographs demonstrate to make the article more accessible for the readers that do preliminary paper skimming. A map of the area would have been highly beneficial for the readers to better visualise the studied site spatial distribution."* | We added a title to Figure 1. We will not include a map of the research area in the text because our study already contains a large number of figures and tables. |
| *"Line 271: It is not clear what the authors try to say here e.g. that there are similar trends between biocrust and total coverage trends? Or something else? Please clarify."* | We clarified this sentence in the revised manuscript. |
| *"Figure 2 caption: Perhaps it would read better as "mean values and standard error are given". Please also remind to the readers the number of replicates."* | The connected scatterplot diagrams in figures 2 and 3 were replaced with boxplot diagrams, so that the figure description is different now. We added the number of replicates in all figure descriptions. |
| *"Line 282: The values of pH should be mentioned."* | The pH values were included in the text. |
| *"Lines 288-289: The authors should elaborate on their statements about contradictions between their findings and those from (Corbin and Thiet, 2020; Bergamini et al., 2001; Fojcik et al., 2019)."* | We added more information in this section to clarify the contradictory results. |
| *"Lines 289-292: The authors should elaborate on the mechanisms driving positive correlations between vascular plants and moss growth."* | We inserted additional information on this in the mentioned lines. |
| *"Line 292: The statements/discussion on biocrust should be on a separate paragraph."* | In order to better distinguish between biocrust cover and bryophyte cover, we have revised the entire manuscript so that we now refer to bryophyte covers in this line. |
| *"Lines 327 – 338: Please make sure that you provide p-values where needed. Also, it is not necessary to use extensively phrases such as "A was X times higher than B". Providing the average values, standard error and the p-values would suffice."* | Thank you for this comment. We included p-values, means, and standard deviations wherever appropriate. |
| *"Lines 339-341: See our comments above about examining the role of environmental parameters in shaping discharge / run off. For example, how much of the variability in discharge is explained by differences in the soil features?"* | To assess the effect of environmental parameters on soil erosion, bryophyte coverage and species richness, we performed generalized additive models (GAM) with restricted maximum |

| | likelihood and smoothing parameters selected by an unbiased risk estimator (UBRE) criterion. |
|---|---|
| *"Sections 3.2.1 and 3.2.2 should be merged. The independent and response variables should be subject to appropriate statistical analysis e.g. distance-based linear modelling (Clarke and Gorley 2015) Clarke KR, Gorley RN (2015) PRIMER v7: User Manual/Tutorial PRIMER-E: Plymouth"* | We merged the sections 3.2.1 and 3.2.2 and added the results of our GAMs. |
| *"Lines 398-401: Some of the lines mentioned here should had been included in the Materials and Methods. Also it is not clear where the term 'reduction' refers to – please clarify."* | As suggested, we moved the mentioned lines to the methods section.
Further we clarified that the term "reduction" in this section refers to sediment discharge. |
| *"Figure 5: The box plots for biocrusts and vascular plants are very close (this is not necessarily bad) and some of the outliers for biocrusts may be regarded as outliers for vascular plants (and vice versa). It would be helpful to see the outliers for each of them with different colours. We feel that a sudden change in the colour scheme on this graph could confuse the readers that got used to seeing dark green as 'wheel track' and light green as 'central track' in previous 3 figures."* | Thank you for bringing this to our attention. We adjusted the colour code in all figures so that dark green is used for "bryophytes" and light green for "vascular plants". Additionally, we reduced the size of the jitter points in figure 5, as this was the most appropriate measure to make the outliers of each displayed group more visible. |
| **Conclusions**
*"Line 426 : it seems that null hypotheses were not made; it is suggested to adjust accordingly the text at the end of the "Introduction"."* | According to your comment, we inserted null hypotheses in the introduction section wherever possible and adjusted the conclusion section as well. |
| *"The conclusions section looks too lengthy; it should appear more succinct and with higher impact. Focus on your key findings and how they fill gaps in the literature. Avoid repeating results and numerical values."* | We shortened the conclusion to the most important outcomes of our study. |
| *"Could include more discussion of direction and opportunities for future studies"* | At the end of the conclusion section, we mentioned a variety of future research topics related to our study. We think that this outlook is sufficient. |
| *"Lines 450- 456: Would it be also of interest to study the factors that support higher growth rates for the biocrust communities?"* | Yes, this is very interesting to study, especially in temperate climates where the evidence of biocrust communities is scarce. |
| *"Figure A1. Please clarify in the image (using arrows) the wheel track and center track."* | We inserted arrows in figure A1 to mark the location of wheel and center tracks. |
| *"It seems that there is some inconsistency in editing/coloring of symbols across the figures e.g., see color code used Figures 3 and 4."* | Thank you for bringing this to our attention. We adjusted the colour code in all figures so that dark green is used for "bryophytes" and light green for "vascular plants". |

---

## Author Comment (AC4)

**Response to Reviewer Comment 1 (RC1) to**

**preprint bg-2021-343: "Pioneer biocrust communities prevent soil erosion in temperate forests after disturbances"**

We thank the reviewer very much for this in depth and positive evaluation of our work. The comments provide a strong basis for substantial improvements, which are included in the revised manuscript.

| Comments                                                                                        | Authors responses                                   |
|-------------------------------------------------------------------------------------------------|-----------------------------------------------------|
| "First, I doubt that many of the bryophytes                                                     | We would like to thank you for this significant     |
| reported in this study fully meet the                                                           | comment, which hits a most interesting point        |
| characteristics of biological soil crusts                                                       | that has been discussed intensively.                |
| (biocrusts). The biocrust definition, as it was                                                 | It is agreed that the moss genera mentioned         |
| first brought forward by Belnap, Büdel and                                                      | grow with the bulk of their biomass above the       |
| Lange (2003) in the first Ecological Studies                                                    | ground and do not meet the basic definition of a    |
| volume on biocrusts, referred to communities of                                                 | biocrust. At the same time, however, they make      |
| organisms that live within or only few                                                          | up a smaller part of the biomass at the beginning   |
| centimeters on top of soil. A key characteristic is                                             | of succession. Along with many other moss           |
| that the major part of the biomass is located                                                   | species, single lichens, algae, and cyanobacteria,  |
| within the soil and that it creates a hardened                                                  | larger amounts of moss protonema can be             |
| soil surface (an encrustation). I think both of                                                 | observed on the soil surface immediately after      |
| these factors are not fully met by the                                                          | disturbance. Together, they can show crustal        |
| communities reported here. In genera like                                                       | characteristics at the beginning, which fulfill the |
| Atrichum, Rhytidiadelphus and Plagiomnium the                                                   | definition of Belnap et al. (2003). In this mesic   |
| major part of the biomass grows above the soil                                                  | forest ecosystem, however, biocrusts occurred as    |
| surface and I also have not experienced a soil                                                  | visually recognizable green cover, which was        |
| hardening effect in the vicinity of them. Thus, I                                               | also reported in recent biocrust studies of         |
| think the term "biological soil crust" is                                                       | comparable forest sites (Kurth et al., 2021;        |
| irritating in this context, as the reader expects                                               | Glaser et al., 2022). In contrast to these studies, |
| somewhat alfferent properties. I think that                                                     | the green cover of our sites is primarily due to    |
| blocrusts indeed could occur at the slopes next                                                 | moss protonema, which is found, as you are          |
| io a joresi pain with species like Folymenum
pilifarum and it might ha that in some parts of | continues to develop quickly, with the crustel      |
| the investigated sites biocrust fragments could                                                 | characteristic disappearing more and more           |
| occur. But for the complete community I doubt                                                   | Furthermore, we accounted thallose liverworts       |
| the correctness of this term                                                                    | among the biocrust species. Nevertheless, this      |
| However I do not see that as a deficit of this                                                  | observation has been made more often in mesic       |
| study at all. The authors could describe the                                                    | ecosystems and very clearly e.g. in highly          |
| studied communities as bryophyte or cryptogam                                                   | disturbed subtropical forest plantations where      |
| communities and they could discuss the                                                          | larger crustal patches were still detectable after  |
| similarities and differences between biocrusts                                                  | 2-3 years (Seitz et al., 2017).                     |
| and their study objects. I think it also is relevant                                            | In this context, this early soil cover after timber |
| that not only biocrusts, but cryptogam                                                          | harvest fulfills an essential (biocrust) function,  |
| communities in general are highly relevant for a                                                | namely, the protection against erosion at a         |
| variety of functional ecosystem processes and                                                   | moment when the soil is highly susceptible. This    |
| the present study shows this clearly once more."                                                | protective function then passes smoothly into       |
| 1 7 7                                                                                           | further vegetation development and, according       |
|                                                                                                 | to our observations, is even more enhanced by       |
|                                                                                                 | fully developed mosses. However, the                |
|                                                                                                 | distinction between biocrust and cryptogamic or     |
|                                                                                                 | just non-vascular vegetation is not always easy     |
|                                                                                                 | to make.                                            |
|                                                                                                 | In summary, we agree that the prominent use of      |
|                                                                                                 | the term biocrusts may lead the reader down the     |
|                                                                                                 | wrong track. This will be adjusted accordingly,     |
|                                                                                                 | and more reference to cryptogamic and/or non-       |
|                                                                                                 | vascular vegetation will be made. Nevertheless,     |

|                                                                                                                                                                                                                                                                                                                                                                                                                                                                                                                                                                                                                                                                                                                                                                                                                                                                                                                                                                                                         | we think that plant communities under the                                                                                                                                                                                                                                                                                                                                                                                                                                                                                                                                                                                                                                                                                                                                                                                                                                                                                                                                                                                                                                                  |
|---------------------------------------------------------------------------------------------------------------------------------------------------------------------------------------------------------------------------------------------------------------------------------------------------------------------------------------------------------------------------------------------------------------------------------------------------------------------------------------------------------------------------------------------------------------------------------------------------------------------------------------------------------------------------------------------------------------------------------------------------------------------------------------------------------------------------------------------------------------------------------------------------------------------------------------------------------------------------------------------------------|--------------------------------------------------------------------------------------------------------------------------------------------------------------------------------------------------------------------------------------------------------------------------------------------------------------------------------------------------------------------------------------------------------------------------------------------------------------------------------------------------------------------------------------------------------------------------------------------------------------------------------------------------------------------------------------------------------------------------------------------------------------------------------------------------------------------------------------------------------------------------------------------------------------------------------------------------------------------------------------------------------------------------------------------------------------------------------------------|
|                                                                                                                                                                                                                                                                                                                                                                                                                                                                                                                                                                                                                                                                                                                                                                                                                                                                                                                                                                                                         | biocrust definition are not yet adequately                                                                                                                                                                                                                                                                                                                                                                                                                                                                                                                                                                                                                                                                                                                                                                                                                                                                                                                                                                                                                                                 |
|                                                                                                                                                                                                                                                                                                                                                                                                                                                                                                                                                                                                                                                                                                                                                                                                                                                                                                                                                                                                         | described in these mesic (and thus rather                                                                                                                                                                                                                                                                                                                                                                                                                                                                                                                                                                                                                                                                                                                                                                                                                                                                                                                                                                                                                                                  |
|                                                                                                                                                                                                                                                                                                                                                                                                                                                                                                                                                                                                                                                                                                                                                                                                                                                                                                                                                                                                         | atypical) ecosystems. We therefore strongly                                                                                                                                                                                                                                                                                                                                                                                                                                                                                                                                                                                                                                                                                                                                                                                                                                                                                                                                                                                                                                                |
|                                                                                                                                                                                                                                                                                                                                                                                                                                                                                                                                                                                                                                                                                                                                                                                                                                                                                                                                                                                                         | welcome your suggestion to compare and                                                                                                                                                                                                                                                                                                                                                                                                                                                                                                                                                                                                                                                                                                                                                                                                                                                                                                                                                                                                                                                     |
|                                                                                                                                                                                                                                                                                                                                                                                                                                                                                                                                                                                                                                                                                                                                                                                                                                                                                                                                                                                                         | discuss similarities and differences between the                                                                                                                                                                                                                                                                                                                                                                                                                                                                                                                                                                                                                                                                                                                                                                                                                                                                                                                                                                                                                                           |
|                                                                                                                                                                                                                                                                                                                                                                                                                                                                                                                                                                                                                                                                                                                                                                                                                                                                                                                                                                                                         | communities.                                                                                                                                                                                                                                                                                                                                                                                                                                                                                                                                                                                                                                                                                                                                                                                                                                                                                                                                                                                                                                                                               |
| "Second, I think the illustrations in this                                                                                                                                                                                                                                                                                                                                                                                                                                                                                                                                                                                                                                                                                                                                                                                                                                                                                                                                                              | Thank you for your recommendation to display                                                                                                                                                                                                                                                                                                                                                                                                                                                                                                                                                                                                                                                                                                                                                                                                                                                                                                                                                                                                                                               |
| manuscript could be improved. In section 3.1.1                                                                                                                                                                                                                                                                                                                                                                                                                                                                                                                                                                                                                                                                                                                                                                                                                                                                                                                                                          | the taxonomic composition graphically which                                                                                                                                                                                                                                                                                                                                                                                                                                                                                                                                                                                                                                                                                                                                                                                                                                                                                                                                                                                                                                                |
| the composition of bryophytes is explained, but                                                                                                                                                                                                                                                                                                                                                                                                                                                                                                                                                                                                                                                                                                                                                                                                                                                                                                                                                         | has considerably increased the                                                                                                                                                                                                                                                                                                                                                                                                                                                                                                                                                                                                                                                                                                                                                                                                                                                                                                                                                                                                                                                             |
| the taxa are only listed in a table and the                                                                                                                                                                                                                                                                                                                                                                                                                                                                                                                                                                                                                                                                                                                                                                                                                                                                                                                                                             | comprehensibility of the results. We added a pie                                                                                                                                                                                                                                                                                                                                                                                                                                                                                                                                                                                                                                                                                                                                                                                                                                                                                                                                                                                                                                           |
| taxonomic composition is not graphically                                                                                                                                                                                                                                                                                                                                                                                                                                                                                                                                                                                                                                                                                                                                                                                                                                                                                                                                                                | diagram that illustrates the occurrence of                                                                                                                                                                                                                                                                                                                                                                                                                                                                                                                                                                                                                                                                                                                                                                                                                                                                                                                                                                                                                                                 |
| displayed. I think this is urgently needed and                                                                                                                                                                                                                                                                                                                                                                                                                                                                                                                                                                                                                                                                                                                                                                                                                                                                                                                                                          | bryophyte species in the ROPs for each                                                                                                                                                                                                                                                                                                                                                                                                                                                                                                                                                                                                                                                                                                                                                                                                                                                                                                                                                                                                                                                     |
| would clearly improve the comprehensibility of                                                                                                                                                                                                                                                                                                                                                                                                                                                                                                                                                                                                                                                                                                                                                                                                                                                                                                                                                          | vegetation survey time step in every skid trail                                                                                                                                                                                                                                                                                                                                                                                                                                                                                                                                                                                                                                                                                                                                                                                                                                                                                                                                                                                                                                            |
| the results. In figures 2 and 3 the line diagram is                                                                                                                                                                                                                                                                                                                                                                                                                                                                                                                                                                                                                                                                                                                                                                                                                                                                                                                                                     | site (see Figure 1).                                                                                                                                                                                                                                                                                                                                                                                                                                                                                                                                                                                                                                                                                                                                                                                                                                                                                                                                                                                                                                                                       |
| not the correct way to illustrate the results, as                                                                                                                                                                                                                                                                                                                                                                                                                                                                                                                                                                                                                                                                                                                                                                                                                                                                                                                                                       | The connected scatterplot diagrams in Figures 2                                                                                                                                                                                                                                                                                                                                                                                                                                                                                                                                                                                                                                                                                                                                                                                                                                                                                                                                                                                                                                            |
| there are no data available for the times                                                                                                                                                                                                                                                                                                                                                                                                                                                                                                                                                                                                                                                                                                                                                                                                                                                                                                                                                               | and 3 were replaced with boxplot diagrams. (see                                                                                                                                                                                                                                                                                                                                                                                                                                                                                                                                                                                                                                                                                                                                                                                                                                                                                                                                                                                                                                            |
| between the measurements. For this type of                                                                                                                                                                                                                                                                                                                                                                                                                                                                                                                                                                                                                                                                                                                                                                                                                                                                                                                                                              | Figure 3 and Figure 4). Furthermore, the                                                                                                                                                                                                                                                                                                                                                                                                                                                                                                                                                                                                                                                                                                                                                                                                                                                                                                                                                                                                                                                   |
| alta, box-whisker plots are correct, as they have                                                                                                                                                                                                                                                                                                                                                                                                                                                                                                                                                                                                                                                                                                                                                                                                                                                                                                                                                       | visualization of the results in Figure 5 was                                                                                                                                                                                                                                                                                                                                                                                                                                                                                                                                                                                                                                                                                                                                                                                                                                                                                                                                                                                                                                               |
| diso been used in the subsequent figures. In                                                                                                                                                                                                                                                                                                                                                                                                                                                                                                                                                                                                                                                                                                                                                                                                                                                                                                                                                            | distinguished (see Figure 4)                                                                                                                                                                                                                                                                                                                                                                                                                                                                                                                                                                                                                                                                                                                                                                                                                                                                                                                                                                                                                                                               |
| superated from each other: I think this could be                                                                                                                                                                                                                                                                                                                                                                                                                                                                                                                                                                                                                                                                                                                                                                                                                                                                                                                                                        | On the level of individual skid trails, which is                                                                                                                                                                                                                                                                                                                                                                                                                                                                                                                                                                                                                                                                                                                                                                                                                                                                                                                                                                                                                                           |
| improved regarding form and color. In all plots                                                                                                                                                                                                                                                                                                                                                                                                                                                                                                                                                                                                                                                                                                                                                                                                                                                                                                                                                         | displayed in the figures, there are four replicates                                                                                                                                                                                                                                                                                                                                                                                                                                                                                                                                                                                                                                                                                                                                                                                                                                                                                                                                                                                                                                        |
| where sampling was conducted at different                                                                                                                                                                                                                                                                                                                                                                                                                                                                                                                                                                                                                                                                                                                                                                                                                                                                                                                                                               | per track position which is insufficient for                                                                                                                                                                                                                                                                                                                                                                                                                                                                                                                                                                                                                                                                                                                                                                                                                                                                                                                                                                                                                                               |
| times the statistics should be added in order to                                                                                                                                                                                                                                                                                                                                                                                                                                                                                                                                                                                                                                                                                                                                                                                                                                                                                                                                                        | performing post-hoc statistics. Furthermore, the                                                                                                                                                                                                                                                                                                                                                                                                                                                                                                                                                                                                                                                                                                                                                                                                                                                                                                                                                                                                                                           |
| illustrate which changes were statistically                                                                                                                                                                                                                                                                                                                                                                                                                                                                                                                                                                                                                                                                                                                                                                                                                                                                                                                                                             | figures are already quite detailed which is why                                                                                                                                                                                                                                                                                                                                                                                                                                                                                                                                                                                                                                                                                                                                                                                                                                                                                                                                                                                                                                            |
| significant "                                                                                                                                                                                                                                                                                                                                                                                                                                                                                                                                                                                                                                                                                                                                                                                                                                                                                                                                                                                           | we did not consider it helpful to include                                                                                                                                                                                                                                                                                                                                                                                                                                                                                                                                                                                                                                                                                                                                                                                                                                                                                                                                                                                                                                                  |
|                                                                                                                                                                                                                                                                                                                                                                                                                                                                                                                                                                                                                                                                                                                                                                                                                                                                                                                                                                                                         | additional information.                                                                                                                                                                                                                                                                                                                                                                                                                                                                                                                                                                                                                                                                                                                                                                                                                                                                                                                                                                                                                                                                    |
|                                                                                                                                                                                                                                                                                                                                                                                                                                                                                                                                                                                                                                                                                                                                                                                                                                                                                                                                                                                                         |                                                                                                                                                                                                                                                                                                                                                                                                                                                                                                                                                                                                                                                                                                                                                                                                                                                                                                                                                                                                                                                                                            |
| "Third, the naming of the plots could be                                                                                                                                                                                                                                                                                                                                                                                                                                                                                                                                                                                                                                                                                                                                                                                                                                                                                                                                                                | Thank you for this suggestion. The sites are                                                                                                                                                                                                                                                                                                                                                                                                                                                                                                                                                                                                                                                                                                                                                                                                                                                                                                                                                                                                                                               |
| "Third, the naming of the plots could be improved. The names of the different forests do                                                                                                                                                                                                                                                                                                                                                                                                                                                                                                                                                                                                                                                                                                                                                                                                                                                                                                         | Thank you for this suggestion. The sites are named according to the geologic formation of                                                                                                                                                                                                                                                                                                                                                                                                                                                                                                                                                                                                                                                                                                                                                                                                                                                                                                                                                                                                  |
| "Third, the naming of the plots could be
improved. The names of the different forests do
not mean anything to the reader. I think it would                                                                                                                                                                                                                                                                                                                                                                                                                                                                                                                                                                                                                                                                                                                                                                                                                                                        | Thank you for this suggestion. The sites are
named according to the geologic formation of
the parent material, and the associated soil types                                                                                                                                                                                                                                                                                                                                                                                                                                                                                                                                                                                                                                                                                                                                                                                                                                                                                                                                         |
| "Third, the naming of the plots could be
improved. The names of the different forests do
not mean anything to the reader. I think it would
be better to name the plots e.g. according to the                                                                                                                                                                                                                                                                                                                                                                                                                                                                                                                                                                                                                                                                                                                                                                                                   | Thank you for this suggestion. The sites are
named according to the geologic formation of
the parent material, and the associated soil types
and textures are outlined in Table A1 in the                                                                                                                                                                                                                                                                                                                                                                                                                                                                                                                                                                                                                                                                                                                                                                                                                                                                                         |
| "Third, the naming of the plots could be
improved. The names of the different forests do
not mean anything to the reader. I think it would
be better to name the plots e.g. according to the
parent material, soil type and/or texture or to                                                                                                                                                                                                                                                                                                                                                                                                                                                                                                                                                                                                                                                                                                                                                | Thank you for this suggestion. The sites are
named according to the geologic formation of
the parent material, and the associated soil types
and textures are outlined in Table A1 in the
Appendix. Since these designations are already                                                                                                                                                                                                                                                                                                                                                                                                                                                                                                                                                                                                                                                                                                                                                                                                                                       |
| "Third, the naming of the plots could be
improved. The names of the different forests do
not mean anything to the reader. I think it would
be better to name the plots e.g. according to the
parent material, soil type and/or texture or to
just give them numbers. This would be                                                                                                                                                                                                                                                                                                                                                                                                                                                                                                                                                                                                                                                                                                       | Thank you for this suggestion. The sites are
named according to the geologic formation of
the parent material, and the associated soil types
and textures are outlined in Table A1 in the
Appendix. Since these designations are already
used in another publication related to this                                                                                                                                                                                                                                                                                                                                                                                                                                                                                                                                                                                                                                                                                                                                                                                        |
| "Third, the naming of the plots could be
improved. The names of the different forests do
not mean anything to the reader. I think it would
be better to name the plots e.g. according to the
parent material, soil type and/or texture or to
just give them numbers. This would be
particularly helpful, as you explain later that the                                                                                                                                                                                                                                                                                                                                                                                                                                                                                                                                                                                                                                                | Thank you for this suggestion. The sites are
named according to the geologic formation of
the parent material, and the associated soil types
and textures are outlined in Table A1 in the
Appendix. Since these designations are already
used in another publication related to this
manuscript, we suggest to retain them (Thielen                                                                                                                                                                                                                                                                                                                                                                                                                                                                                                                                                                                                                                                                                                                                      |
| "Third, the naming of the plots could be
improved. The names of the different forests do
not mean anything to the reader. I think it would
be better to name the plots e.g. according to the
parent material, soil type and/or texture or to
just give them numbers. This would be
particularly helpful, as you explain later that the
substrate indeed had an effect on the observed                                                                                                                                                                                                                                                                                                                                                                                                                                                                                                                                                                                              | Thank you for this suggestion. The sites are
named according to the geologic formation of
the parent material, and the associated soil types
and textures are outlined in Table A1 in the
Appendix. Since these designations are already
used in another publication related to this
manuscript, we suggest to retain them (Thielen
et al., 2021).                                                                                                                                                                                                                                                                                                                                                                                                                                                                                                                                                                                                                                                                                                                    |
| "Third, the naming of the plots could be
improved. The names of the different forests do
not mean anything to the reader. I think it would
be better to name the plots e.g. according to the
parent material, soil type and/or texture or to
just give them numbers. This would be
particularly helpful, as you explain later that the
substrate indeed had an effect on the observed
vegetation."                                                                                                                                                                                                                                                                                                                                                                                                                                                                                                                                                                              | Thank you for this suggestion. The sites are
named according to the geologic formation of
the parent material, and the associated soil types
and textures are outlined in Table A1 in the
Appendix. Since these designations are already
used in another publication related to this
manuscript, we suggest to retain them (Thielen
et al., 2021).
However, with regard to your comment and as                                                                                                                                                                                                                                                                                                                                                                                                                                                                                                                                                                                                                                                                     |
| "Third, the naming of the plots could be
improved. The names of the different forests do
not mean anything to the reader. I think it would
be better to name the plots e.g. according to the
parent material, soil type and/or texture or to
just give them numbers. This would be
particularly helpful, as you explain later that the
substrate indeed had an effect on the observed
vegetation."                                                                                                                                                                                                                                                                                                                                                                                                                                                                                                                                                                              | Thank you for this suggestion. The sites are
named according to the geologic formation of
the parent material, and the associated soil types
and textures are outlined in Table A1 in the
Appendix. Since these designations are already
used in another publication related to this
manuscript, we suggest to retain them (Thielen
et al., 2021).
However, with regard to your comment and as
we agree that readability can generally be                                                                                                                                                                                                                                                                                                                                                                                                                                                                                                                                                                                                                       |
| "Third, the naming of the plots could be
improved. The names of the different forests do
not mean anything to the reader. I think it would
be better to name the plots e.g. according to the
parent material, soil type and/or texture or to
just give them numbers. This would be
particularly helpful, as you explain later that the
substrate indeed had an effect on the observed
vegetation."                                                                                                                                                                                                                                                                                                                                                                                                                                                                                                                                                                              | Thank you for this suggestion. The sites are
named according to the geologic formation of
the parent material, and the associated soil types
and textures are outlined in Table A1 in the
Appendix. Since these designations are already
used in another publication related to this
manuscript, we suggest to retain them (Thielen
et al., 2021).
However, with regard to your comment and as
we agree that readability can generally be
improved, we decided to reduce the use of                                                                                                                                                                                                                                                                                                                                                                                                                                                                                                                                                                          |
| "Third, the naming of the plots could be
improved. The names of the different forests do
not mean anything to the reader. I think it would
be better to name the plots e.g. according to the
parent material, soil type and/or texture or to
just give them numbers. This would be
particularly helpful, as you explain later that the
substrate indeed had an effect on the observed
vegetation."                                                                                                                                                                                                                                                                                                                                                                                                                                                                                                                                                                              | Thank you for this suggestion. The sites are
named according to the geologic formation of
the parent material, and the associated soil types
and textures are outlined in Table A1 in the
Appendix. Since these designations are already
used in another publication related to this
manuscript, we suggest to retain them (Thielen
et al., 2021).
However, with regard to your comment and as
we agree that readability can generally be
improved, we decided to reduce the use of
abbreviations in the text. Therefore, we have                                                                                                                                                                                                                                                                                                                                                                                                                                                                                                                         |
| "Third, the naming of the plots could be
improved. The names of the different forests do
not mean anything to the reader. I think it would
be better to name the plots e.g. according to the
parent material, soil type and/or texture or to
just give them numbers. This would be
particularly helpful, as you explain later that the
substrate indeed had an effect on the observed
vegetation."                                                                                                                                                                                                                                                                                                                                                                                                                                                                                                                                                                              | Thank you for this suggestion. The sites are
named according to the geologic formation of
the parent material, and the associated soil types
and textures are outlined in Table A1 in the
Appendix. Since these designations are already
used in another publication related to this
manuscript, we suggest to retain them (Thielen
et al., 2021).
However, with regard to your comment and as
we agree that readability can generally be
improved, we decided to reduce the use of
abbreviations in the text. Therefore, we have
spelled out CT (center track), WT (wheel track),                                                                                                                                                                                                                                                                                                                                                                                                                                                                     |
| "Third, the naming of the plots could be
improved. The names of the different forests do
not mean anything to the reader. I think it would
be better to name the plots e.g. according to the
parent material, soil type and/or texture or to
just give them numbers. This would be
particularly helpful, as you explain later that the
substrate indeed had an effect on the observed
vegetation."                                                                                                                                                                                                                                                                                                                                                                                                                                                                                                                                                                              | Thank you for this suggestion. The sites are
named according to the geologic formation of
the parent material, and the associated soil types
and textures are outlined in Table A1 in the
Appendix. Since these designations are already
used in another publication related to this
manuscript, we suggest to retain them (Thielen
et al., 2021).
However, with regard to your comment and as
we agree that readability can generally be
improved, we decided to reduce the use of
abbreviations in the text. Therefore, we have
spelled out CT (center track), WT (wheel track),
and UF (undisturbed forest soil) in the revised
manuscript                                                                                                                                                                                                                                                                                                                                                                                                    |
| "Third, the naming of the plots could be
improved. The names of the different forests do
not mean anything to the reader. I think it would
be better to name the plots e.g. according to the
parent material, soil type and/or texture or to
just give them numbers. This would be
particularly helpful, as you explain later that the
substrate indeed had an effect on the observed
vegetation."                                                                                                                                                                                                                                                                                                                                                                                                                                                                                                                                                                              | Thank you for this suggestion. The sites are
named according to the geologic formation of
the parent material, and the associated soil types
and textures are outlined in Table A1 in the
Appendix. Since these designations are already
used in another publication related to this
manuscript, we suggest to retain them (Thielen
et al., 2021).
However, with regard to your comment and as
we agree that readability can generally be
improved, we decided to reduce the use of
abbreviations in the text. Therefore, we have
spelled out CT (center track), WT (wheel track),
and UF (undisturbed forest soil) in the revised
manuscript.                                                                                                                                                                                                                                                                                                                                                                                                   |
| "Third, the naming of the plots could be
improved. The names of the different forests do
not mean anything to the reader. I think it would
be better to name the plots e.g. according to the
parent material, soil type and/or texture or to
just give them numbers. This would be
particularly helpful, as you explain later that the
substrate indeed had an effect on the observed
vegetation."                                                                                                                                                                                                                                                                                                                                                                                                                                                                                                                                                                              | Thank you for this suggestion. The sites are
named according to the geologic formation of
the parent material, and the associated soil types
and textures are outlined in Table A1 in the
Appendix. Since these designations are already
used in another publication related to this
manuscript, we suggest to retain them (Thielen
et al., 2021).
However, with regard to your comment and as
we agree that readability can generally be
improved, we decided to reduce the use of
abbreviations in the text. Therefore, we have
spelled out CT (center track), WT (wheel track),
and UF (undisturbed forest soil) in the revised
manuscript.
For clarification, we have added the years to the
months in all figures (see Figure 2 Figure 3                                                                                                                                                                                                                                                                                              |
| "Third, the naming of the plots could be
improved. The names of the different forests do
not mean anything to the reader. I think it would
be better to name the plots e.g. according to the
parent material, soil type and/or texture or to
just give them numbers. This would be
particularly helpful, as you explain later that the
substrate indeed had an effect on the observed
vegetation."
"Fourth, I think it might be irritating to name
only the month of sampling. It would be clearer
if you name them e.g. as Mar19 Jul19 Oct19                                                                                                                                                                                                                                                                                                                                                                                                                          | Thank you for this suggestion. The sites are
named according to the geologic formation of
the parent material, and the associated soil types
and textures are outlined in Table A1 in the
Appendix. Since these designations are already
used in another publication related to this
manuscript, we suggest to retain them (Thielen
et al., 2021).
However, with regard to your comment and as
we agree that readability can generally be
improved, we decided to reduce the use of
abbreviations in the text. Therefore, we have
spelled out CT (center track), WT (wheel track),
and UF (undisturbed forest soil) in the revised
manuscript.
For clarification, we have added the years to the
months in all figures (see Figure 2, Figure 3,
Figure 4, Figure 5) and in the text, as suggested                                                                                                                                                                                                                                       |
| "Third, the naming of the plots could be
improved. The names of the different forests do
not mean anything to the reader. I think it would
be better to name the plots e.g. according to the
parent material, soil type and/or texture or to
just give them numbers. This would be
particularly helpful, as you explain later that the
substrate indeed had an effect on the observed
vegetation."
"Fourth, I think it might be irritating to name
only the month of sampling. It would be clearer
if you name them e.g. as Mar19, Jul19, Oct19,
Feb20"                                                                                                                                                                                                                                                                                                                                                                                                             | Thank you for this suggestion. The sites are
named according to the geologic formation of
the parent material, and the associated soil types
and textures are outlined in Table A1 in the
Appendix. Since these designations are already
used in another publication related to this
manuscript, we suggest to retain them (Thielen
et al., 2021).
However, with regard to your comment and as
we agree that readability can generally be
improved, we decided to reduce the use of
abbreviations in the text. Therefore, we have
spelled out CT (center track), WT (wheel track),
and UF (undisturbed forest soil) in the revised
manuscript.
For clarification, we have added the years to the
months in all figures (see Figure 2, Figure 3,
Figure 4, Figure 5) and in the text, as suggested.                                                                                                                                                                                                                                      |
| "Third, the naming of the plots could be
improved. The names of the different forests do
not mean anything to the reader. I think it would
be better to name the plots e.g. according to the
parent material, soil type and/or texture or to
just give them numbers. This would be
particularly helpful, as you explain later that the
substrate indeed had an effect on the observed
vegetation."
"Fourth, I think it might be irritating to name
only the month of sampling. It would be clearer
if you name them e.g. as Mar19, Jul19, Oct19,
Feb20"
"Fifth and finally, the language needs to be                                                                                                                                                                                                                                                                                                                                                             | Thank you for this suggestion. The sites are
named according to the geologic formation of
the parent material, and the associated soil types
and textures are outlined in Table A1 in the
Appendix. Since these designations are already
used in another publication related to this
manuscript, we suggest to retain them (Thielen
et al., 2021).
However, with regard to your comment and as
we agree that readability can generally be
improved, we decided to reduce the use of
abbreviations in the text. Therefore, we have
spelled out CT (center track), WT (wheel track),
and UF (undisturbed forest soil) in the revised
manuscript.
For clarification, we have added the years to the
months in all figures (see Figure 2, Figure 3,
Figure 4, Figure 5) and in the text, as suggested.
We regret that there were problems with our                                                                                                                                                                                       |
| "Third, the naming of the plots could be
improved. The names of the different forests do
not mean anything to the reader. I think it would
be better to name the plots e.g. according to the
parent material, soil type and/or texture or to
just give them numbers. This would be
particularly helpful, as you explain later that the
substrate indeed had an effect on the observed
vegetation."
"Fourth, I think it might be irritating to name
only the month of sampling. It would be clearer
if you name them e.g. as Mar19, Jul19, Oct19,
Feb20"
"Fifth and finally, the language needs to be
carefully and thoroughly checked throughout the                                                                                                                                                                                                                                                                                                          | Thank you for this suggestion. The sites are
named according to the geologic formation of
the parent material, and the associated soil types
and textures are outlined in Table A1 in the
Appendix. Since these designations are already
used in another publication related to this
manuscript, we suggest to retain them (Thielen
et al., 2021).
However, with regard to your comment and as
we agree that readability can generally be
improved, we decided to reduce the use of
abbreviations in the text. Therefore, we have
spelled out CT (center track), WT (wheel track),
and UF (undisturbed forest soil) in the revised
manuscript.
For clarification, we have added the years to the
months in all figures (see Figure 2, Figure 3,
Figure 4, Figure 5) and in the text, as suggested.
We regret that there were problems with our
uses of English. According to your                                                                                                                                                 |
| "Third, the naming of the plots could be
improved. The names of the different forests do
not mean anything to the reader. I think it would
be better to name the plots e.g. according to the
parent material, soil type and/or texture or to
just give them numbers. This would be
particularly helpful, as you explain later that the
substrate indeed had an effect on the observed
vegetation."
"Fourth, I think it might be irritating to name
only the month of sampling. It would be clearer
if you name them e.g. as Mar19, Jul19, Oct19,
Feb20"
"Fifth and finally, the language needs to be
carefully and thoroughly checked throughout the
manuscript. Beyond minor mistakes, which are                                                                                                                                                                                                                                                          | Thank you for this suggestion. The sites are
named according to the geologic formation of
the parent material, and the associated soil types
and textures are outlined in Table A1 in the
Appendix. Since these designations are already
used in another publication related to this
manuscript, we suggest to retain them (Thielen
et al., 2021).
However, with regard to your comment and as
we agree that readability can generally be
improved, we decided to reduce the use of
abbreviations in the text. Therefore, we have
spelled out CT (center track), WT (wheel track),
and UF (undisturbed forest soil) in the revised
manuscript.
For clarification, we have added the years to the
months in all figures (see Figure 2, Figure 3,
Figure 4, Figure 5) and in the text, as suggested.
We regret that there were problems with our
uses of English. According to your
recommendation the revised manuscript has                                                                                                    |
| "Third, the naming of the plots could be
improved. The names of the different forests do
not mean anything to the reader. I think it would
be better to name the plots e.g. according to the
parent material, soil type and/or texture or to
just give them numbers. This would be
particularly helpful, as you explain later that the
substrate indeed had an effect on the observed
vegetation."
"Fourth, I think it might be irritating to name
only the month of sampling. It would be clearer
if you name them e.g. as Mar19, Jul19, Oct19,
Feb20"
"Fifth and finally, the language needs to be
carefully and thoroughly checked throughout the
manuscript. Beyond minor mistakes, which are
not a big issue, there are also sentences where                                                                                                                                                                                                       | Thank you for this suggestion. The sites are
named according to the geologic formation of
the parent material, and the associated soil types
and textures are outlined in Table A1 in the
Appendix. Since these designations are already
used in another publication related to this
manuscript, we suggest to retain them (Thielen
et al., 2021).
However, with regard to your comment and as
we agree that readability can generally be
improved, we decided to reduce the use of
abbreviations in the text. Therefore, we have
spelled out CT (center track), WT (wheel track),
and UF (undisturbed forest soil) in the revised
manuscript.
For clarification, we have added the years to the
months in all figures (see Figure 2, Figure 3,
Figure 4, Figure 5) and in the text, as suggested.
We regret that there were problems with our
uses of English. According to your
recommendation the revised manuscript has
been carefully proofread by a native English                                                    |
| "Third, the naming of the plots could be
improved. The names of the different forests do
not mean anything to the reader. I think it would
be better to name the plots e.g. according to the
parent material, soil type and/or texture or to
just give them numbers. This would be
particularly helpful, as you explain later that the
substrate indeed had an effect on the observed
vegetation."
"Fourth, I think it might be irritating to name
only the month of sampling. It would be clearer
if you name them e.g. as Mar19, Jul19, Oct19,
Feb20"
"Fifth and finally, the language needs to be
carefully and thoroughly checked throughout the
manuscript. Beyond minor mistakes, which are
not a big issue, there are also sentences where
the meaning remains unclear. Thus, careful and                                                                                                                                                     | Thank you for this suggestion. The sites are
named according to the geologic formation of
the parent material, and the associated soil types
and textures are outlined in Table A1 in the
Appendix. Since these designations are already
used in another publication related to this
manuscript, we suggest to retain them (Thielen
et al., 2021).
However, with regard to your comment and as
we agree that readability can generally be
improved, we decided to reduce the use of
abbreviations in the text. Therefore, we have
spelled out CT (center track), WT (wheel track),
and UF (undisturbed forest soil) in the revised
manuscript.
For clarification, we have added the years to the
months in all figures (see Figure 2, Figure 3,
Figure 4, Figure 5) and in the text, as suggested.
We regret that there were problems with our
uses of English. According to your
recommendation the revised manuscript has
been carefully proofread by a native English
speaker to improve the grammar and readability. |
| "Third, the naming of the plots could be
improved. The names of the different forests do
not mean anything to the reader. I think it would
be better to name the plots e.g. according to the
parent material, soil type and/or texture or to
just give them numbers. This would be
particularly helpful, as you explain later that the
substrate indeed had an effect on the observed
vegetation."
"Fourth, I think it might be irritating to name
only the month of sampling. It would be clearer
if you name them e.g. as Mar19, Jul19, Oct19,
Feb20"
"Fifth and finally, the language needs to be
carefully and thoroughly checked throughout the
manuscript. Beyond minor mistakes, which are
not a big issue, there are also sentences where
the meaning remains unclear. Thus, careful and
thorough language editing is urgently needed                                                                                                     | Thank you for this suggestion. The sites are
named according to the geologic formation of
the parent material, and the associated soil types
and textures are outlined in Table A1 in the
Appendix. Since these designations are already
used in another publication related to this
manuscript, we suggest to retain them (Thielen
et al., 2021).
However, with regard to your comment and as
we agree that readability can generally be
improved, we decided to reduce the use of
abbreviations in the text. Therefore, we have
spelled out CT (center track), WT (wheel track),
and UF (undisturbed forest soil) in the revised
manuscript.
For clarification, we have added the years to the
months in all figures (see Figure 2, Figure 3,
Figure 4, Figure 5) and in the text, as suggested.
We regret that there were problems with our
uses of English. According to your
recommendation the revised manuscript has
been carefully proofread by a native English
speaker to improve the grammar and readability. |
| "Third, the naming of the plots could be
improved. The names of the different forests do
not mean anything to the reader. I think it would
be better to name the plots e.g. according to the
parent material, soil type and/or texture or to
just give them numbers. This would be
particularly helpful, as you explain later that the
substrate indeed had an effect on the observed
vegetation."
"Fourth, I think it might be irritating to name
only the month of sampling. It would be clearer
if you name them e.g. as Mar19, Jul19, Oct19,
Feb20"
"Fifth and finally, the language needs to be
carefully and thoroughly checked throughout the
manuscript. Beyond minor mistakes, which are
not a big issue, there are also sentences where
the meaning remains unclear. Thus, careful and
thorough language editing is urgently needed
before final publication could be considered."                                                   | Thank you for this suggestion. The sites are
named according to the geologic formation of
the parent material, and the associated soil types
and textures are outlined in Table A1 in the
Appendix. Since these designations are already
used in another publication related to this
manuscript, we suggest to retain them (Thielen
et al., 2021).
However, with regard to your comment and as
we agree that readability can generally be
improved, we decided to reduce the use of
abbreviations in the text. Therefore, we have
spelled out CT (center track), WT (wheel track),
and UF (undisturbed forest soil) in the revised
manuscript.
For clarification, we have added the years to the
months in all figures (see Figure 2, Figure 3,
Figure 4, Figure 5) and in the text, as suggested.
We regret that there were problems with our
uses of English. According to your
recommendation the revised manuscript has
been carefully proofread by a native English
speaker to improve the grammar and readability. |
| "Third, the naming of the plots could be
improved. The names of the different forests do
not mean anything to the reader. I think it would
be better to name the plots e.g. according to the
parent material, soil type and/or texture or to
just give them numbers. This would be
particularly helpful, as you explain later that the
substrate indeed had an effect on the observed
vegetation."
"Fourth, I think it might be irritating to name
only the month of sampling. It would be clearer
if you name them e.g. as Mar19, Jul19, Oct19,
Feb20"
"Fifth and finally, the language needs to be
carefully and thoroughly checked throughout the
manuscript. Beyond minor mistakes, which are
not a big issue, there are also sentences where
the meaning remains unclear. Thus, careful and
thorough language editing is urgently needed
before final publication could be considered."
"In line 143-145 it is written that "Four ROPs | Thank you for this suggestion. The sites are
named according to the geologic formation of
the parent material, and the associated soil types
and textures are outlined in Table A1 in the
Appendix. Since these designations are already
used in another publication related to this
manuscript, we suggest to retain them (Thielen
et al., 2021).
However, with regard to your comment and as
we agree that readability can generally be
improved, we decided to reduce the use of
abbreviations in the text. Therefore, we have
spelled out CT (center track), WT (wheel track),
and UF (undisturbed forest soil) in the revised
manuscript.
For clarification, we have added the years to the
months in all figures (see Figure 2, Figure 3,
Figure 4, Figure 5) and in the text, as suggested.
We regret that there were problems with our
uses of English. According to your
recommendation the revised manuscript has
been carefully proofread by a native English
speaker to improve the grammar and readability. |

| trail ( $n = 32$ ), and two ROPs in the undisturbed
forest soil (UF) next to every skid trial site ( $n = 8$ )." This is not clear. Does it mean that on every
skid trail four ROPs were installed? This would
mean that there were 4 skid trails in total? Does
it mean 4 skid trails each at WT and CT? This
needs to be clarified. Also the rainfall simulation
numbers given in the following sentence are not
clear. I think a thorough language check will | In total, we had four skid trails and installed four
ROPs in each wheel track and center track (n =
32), and two ROPs in the undisturbed forest soil
adjacent to every skid trail (n = 8). The rainfall
simulations in the skid trails were repeated four
times a year (March 2019, July 2019, October
2019, February 2020), while the rainfall
simulations in the undisturbed forest soil were
rapeated twice in October 2019 and February |
|---------------------------------------------------------------------------------------------------------------------------------------------------------------------------------------------------------------------------------------------------------------------------------------------------------------------------------------------------------------------------------------------------------------------------------------------------------------------------------------|---------------------------------------------------------------------------------------------------------------------------------------------------------------------------------------------------------------------------------------------------------------------------------------------------------------------------------------------------------------------------------------------------------------------------------------------------------------------|
| help to also clarify these issues."                                                                                                                                                                                                                                                                                                                                                                                                                                                   | 2020. In summary, this brings us to 144                                                                                                                                                                                                                                                                                                                                                                                                                             |
| "I inc 35-37. In this sentence there are several                                                                                                                                                                                                                                                                                                                                                                                                                                      | measurements.                                                                                                                                                                                                                                                                                                                                                                                                                                                       |
| language style problems. I would suggest to                                                                                                                                                                                                                                                                                                                                                                                                                                           | changed the sentence accordingly.                                                                                                                                                                                                                                                                                                                                                                                                                                   |
| reformulate it in the following way: The most                                                                                                                                                                                                                                                                                                                                                                                                                                         |                                                                                                                                                                                                                                                                                                                                                                                                                                                                     |
| prominent soil loss occurs in agricultural                                                                                                                                                                                                                                                                                                                                                                                                                                            |                                                                                                                                                                                                                                                                                                                                                                                                                                                                     |
| environments, and thus a considerable part of                                                                                                                                                                                                                                                                                                                                                                                                                                         |                                                                                                                                                                                                                                                                                                                                                                                                                                                                     |
| relevant research is conducted in these                                                                                                                                                                                                                                                                                                                                                                                                                                               |                                                                                                                                                                                                                                                                                                                                                                                                                                                                     |
| habitats.                                                                                                                                                                                                                                                                                                                                                                                                                                                                             | Thank you for bringing this to our attention. We                                                                                                                                                                                                                                                                                                                                                                                                                    |
| Line 40-47. nere 1 linink you want to suy The most important reason for this is soil                                                                                                                                                                                                                                                                                                                                                                             | inserted the word "caused" which clearly                                                                                                                                                                                                                                                                                                                                                                                                                            |
| compaction and reduced infiltration rates                                                                                                                                                                                                                                                                                                                                                                                                                                             | improves the sentence.                                                                                                                                                                                                                                                                                                                                                                                                                                              |
| caused by heavy machines used for timber                                                                                                                                                                                                                                                                                                                                                                                                                                              |                                                                                                                                                                                                                                                                                                                                                                                                                                                                     |
| harvesting""                                                                                                                                                                                                                                                                                                                                                                                                                                                                          |                                                                                                                                                                                                                                                                                                                                                                                                                                                                     |
| "Line 48: significantly"                                                                                                                                                                                                                                                                                                                                                                                                                                                              | We inserted "significantly".                                                                                                                                                                                                                                                                                                                                                                                                                                        |
| "Line 55: exchange "which" by "that""                                                                                                                                                                                                                                                                                                                                                                                                                                                 | We replaced "which" by "that".                                                                                                                                                                                                                                                                                                                                                                                                                                      |
| "Line 60: "These" instead of "those""                                                                                                                                                                                                                                                                                                                                                                                                                                                 | We exchanged "Those" by "These".                                                                                                                                                                                                                                                                                                                                                                                                                                    |
| "Line 75: As most studies investigating the impact"                                                                                                                                                                                                                                                                                                                                                                                                                            | We adjusted the sentence accordingly.                                                                                                                                                                                                                                                                                                                                                                                                                               |
| "Line 80-81: This sentence is upside down.                                                                                                                                                                                                                                                                                                                                                                                                                                     | We changed the order of the sentence as                                                                                                                                                                                                                                                                                                                                                                                                                             |
| 'Pioneer biocrust communities could provide                                                                                                                                                                                                                                                                                                                                                                                                                                           | suggested.                                                                                                                                                                                                                                                                                                                                                                                                                                                          |
| benefits' or 'the soil benefits from biocrusts'"                                                                                                                                                                                                                                                                                                                                                                                                                                      |                                                                                                                                                                                                                                                                                                                                                                                                                                                                     |
| "Line 114: The skid trails show no geological                                                                                                                                                                                                                                                                                                                                                                                                                                         | We have made clarifying rephrasings for this                                                                                                                                                                                                                                                                                                                                                                                                                        |
| formation, but the underlying rocks and soil do.                                                                                                                                                                                                                                                                                                                                                                                                                                      | purpose                                                                                                                                                                                                                                                                                                                                                                                                                                                             |
| Please adapt wording"                                                                                                                                                                                                                                                                                                                                                                                                                                                                 |                                                                                                                                                                                                                                                                                                                                                                                                                                                                     |
| Line 119: formea by extensive perigiaciai                                                                                                                                                                                                                                                                                                                                                                                                                                             | we have reformulated the sentence accordingly.                                                                                                                                                                                                                                                                                                                                                                                                                      |
| "Line 125-127. There are several abbreviations                                                                                                                                                                                                                                                                                                                                                                                                                                        | The explanations for the abbreviations were                                                                                                                                                                                                                                                                                                                                                                                                                         |
| that need to be explained: Ad-hoc-Ag Boden.                                                                                                                                                                                                                                                                                                                                                                                                                                           | inserted in the revised manuscript.                                                                                                                                                                                                                                                                                                                                                                                                                                 |
| Iuss Working Group Wrb, WRB Tool"                                                                                                                                                                                                                                                                                                                                                                                                                                                     | ······································                                                                                                                                                                                                                                                                                                                                                                                                                              |
| "Line 148: A rainfall intensity of 45 mm does                                                                                                                                                                                                                                                                                                                                                                                                                                         | Thank you for clarifying this. We have corrected                                                                                                                                                                                                                                                                                                                                                                                                                    |
| not make sense. I think you speak of a rainfall                                                                                                                                                                                                                                                                                                                                                                                                                                       | the sentence accordingly.                                                                                                                                                                                                                                                                                                                                                                                                                                           |
| intensity of 90 mm h-1, applied over a duration                                                                                                                                                                                                                                                                                                                                                                                                                                       |                                                                                                                                                                                                                                                                                                                                                                                                                                                                     |
| of 30 minutes"                                                                                                                                                                                                                                                                                                                                                                                                                                                                        |                                                                                                                                                                                                                                                                                                                                                                                                                                                                     |
| "Line 200-201: meaning of sentence unclear"                                                                                                                                                                                                                                                                                                                                                                                                                                    | We removed this sentence.                                                                                                                                                                                                                                                                                                                                                                                                                                           |

**References**

Belnap, J., Büdel, B., and Lange, O. L.: Biological Soil Crusts: Characteristics and Distribution, in:Biological Soil Crusts: Structure, Function, and Management, edited by: Belnap, J., and Lange, O. L.,Springer Berlin Heidelberg, Berlin, Heidelberg, 3-30, 10.1007/978-3-642-56475-8\_1, 2003.

Glaser, K., Albrecht, M., Baumann, K., Overmann, J., and Sikorski, J.: Biological Soil Crust From Mesic Forests Promote a Specific Bacteria Community, Frontiers in Microbiology, 13, https://doi.org/10.3389/fmicb.2022.769767, 2022.

Kurth, J. K., Albrecht, M., Karsten, U., Glaser, K., Schloter, M., and Schulz, S.: Correlation of the abundance of bacteria catalyzing phosphorus and nitrogen turnover in biological soil crusts of temperate forests of Germany, Biology and Fertility of Soils, 57, 179-192, https://doi.org/10.1007/s00374-020-01515-3, 2021.

Seitz, S., Nebel, M., Goebes, P., Käppeler, K., Schmidt, K., Shi, X., Song, Z., Webber, C. L., Weber, B., and Scholten, T.: Bryophyte-dominated biological soil crusts mitigate soil erosion in an early successional Chinese subtropical forest, Biogeosciences, 14, 5775-5788, https://doi.org/10.5194/bg-14-5775-2017, 2017.

Thielen, S. M., Gall, C., Ebner, M., Nebel, M., Scholten, T., and Seitz, S.: Water's path from moss to soil: A multi-methodological study on water absorption and evaporation of soil-moss combinations, Journal of Hydrology and Hydromechanics, 69, https://doi.org/10.2478/johh-2021-0021, 2021.

---

## Author Comment (AC5)

**Response to Reviewer Comment 2 (RC2) to**

**preprint bg-2021-343: "Pioneer biocrust communities prevent soil erosion in temperate forests after disturbances"**

Thank you very much for your review, the positive evaluation of our work and the very valuable suggestions to improve the manuscript.

| Comments | Authors responses |
|---|---|
| **Figure 2 and 3**
*"For Figure 2 and 3 I would recommend not using line charts but possibly box plots. Since these are specific monitoring times and not continuous monitoring it gives the wrong suggestion to the reader, especially since the slope of the lines is very different (because the x-axis distances are all the same, although timewise they are not, June-July is not the same time as July-October)."* | Thank you for bringing this to our attention. We replaced the connected scatterplot diagrams in Figures 2 and 3 with boxplot diagrams (see Figure 3 and Figure 4). |
| **Figure 2**
*"Perhaps you could consider, for Figure 2, putting the difference between wheel track and center track in one panel (bryophytes) and the difference between wheel track and center track for total vegetation in another panel. With an adjusted y-axis for bryophytes it would be much easier to see differences between the two track types. This is just a suggestion."* | Thank you for this recommendation. We tried the suggested display for Figure 2, but discarded it after closer examination because we would have to distinguish colour between wheel and center tracks for this representation, and we believe that it is more comprehensible to the reader at this point to stick with the selected uniform colour code to distinguish between "bryophytes" and "vascular plants". Please see also comments given by public review #1. |
| **Figure 2 and 3**
*"To distinguish the information in Figure 2 from Figure 3 it might be better to use different colours. In Fig. 2 bryophytes are presented in dark green while total vegetation is yellowish, in Fig. 3 these colours are used to distinguish the track types which makes it more difficult to grasp the information from the figure directly. Consider using larger symbols for bryophytes etc. so it is more easily readable."* | We decided to adjust the colour code in all figures so that dark green is used for "bryophytes" and light green for "vascular plants", which makes the figures more comprehensible for the reader. |
| **Figure 5**
*"The distribution of sample dots in Figure 5 just seems random and does not improve the quality of the figure. The information about the number of sampling points could also be added into the figure caption."* | Thank you for this comment. We removed the jitter points in Figure 5, which clearly improved the visualization (see Figure 6). Furthermore, we added the number of sample points for each cover in the figure caption. |
| *"Line 148 rainfall intensity should be given as mm h-1. Do you mean 45 mm in 30 minutes meaning 90 mm h-1. This would be an extremely heavy precipitation event and one not typically found in the region, I presume."* | We inserted more background information to the selected rainfall intensity and corrected the given unit to mm h$^{-1}$. |
| *"Chapter 3.2.1 I understand that you want to distinguish the skid trails from the undisturbed forest, yet the results seem to show that wheel tracks and center tracks are very different in their soil erosion characteristics, maybe separate them when speaking about the total* | As suggested, we removed the mean values for the entire skid trails in this chapter and instead only dealt with the mean values per wheel track and center track. |

| | |
|---|---|
| *values for sediment discharge and surface runoff."* | |
| *"**Lines 358-364** You speak of rainfall events, but you mean rainfall simulations? As I understand it, these ROPs can also be used to measure sediment loss and surface runoff during natural rainfall events, did you measure these in between your monitoring times?"* | Yes, you are right, we mean rainfall simulations in these cases. We clarified this. Generally, ROPs can be used to measure surface runoff and sediment discharge during simulated rainfall and natural rainfall events. In our study, we just conducted measurements with simulated rainfall. |
| *"**Figure 5** As you write the higher the percentage of vascular plant cover or biocrust cover the lower sediment loss. Why is the sediment discharge for 11-25 % biocrust cover so low in comparison to the sediment discharge with higher biocrust cover (26-50%)? Do you think it is because of only few measurements were performed in this cover class? You should also explain not only the outlier dots but also your „sample" dots in the figure caption."* | In Figure 5, our measurements of sediment discharge at four different skid trails were reclassified and plotted in cover classes to represent the general influence of bryophytes and vascular plants on soil erosion. Except of the cover class "< 10 %" with 13 measurements, we have 3 – 4 measurements for bryophyte ROPs in each cover class, so this difference is not due to sample size. We assume the reason is that different skid trails are grouped together in each cover class. Cover class "11-25 %" includes two measurements of TS and one of LS, while cover class "26-50 %" contains two measurements of PT, one of TS and one of LS. In general, soil erosion was significantly higher in PT than in TS. The jitter points in Figure 5 were removed to increase comprehensibility (see Figure 6). |
| *"**Figure A1** Unfortunately, the rainfall simulator (except for the cannot be seen, consider using a different, more expressive picture."* | We replaced image "a" in Figure A1 (see Figure A2) so that readers can also see the Tübingen rainfall simulator inside the protective tent. |
| *"**Chapter 2.1** Consider adding an extra figure for the study area"* | We added an extra figure (Figure A1) for the study area in the Appendix. |
| *"**Lines 27- 28** the last sentence needs work: ... biocrusts showed an average sediment loss that was 18 times lower than under vascular plants."* | We decided to delete this sentence in the abstract because it was too specific at this point. |
| *"**Line 41** important dimensions?"* | We have rephrased this sentence to make clearer that soil erosion in forests can be locally very severe. |
| *"**Line 68** bryophyte-dominated biocrusts"* | Thank you, we corrected this according to your comment. |
| *"**Line 75** very most? As the most studies"* | According to your comment, we deleted "very" in this sentence. |
| *"**Line 127** „a" Eutric Cambisol"* | We adjusted this. |
| *"**Line 135** „a" Eutric Calcaric"* | We adjusted this. |
| *"**Line 173** Nomenclature see Table 1 and Table 2 à please use full sentences or use brackets"* | As suggested, we have now used brackets instead. |
| *"**Line 202** no italics for citation"* | We removed this sentence. |
| *"**Table 1** no italics for the authors"* | We changed the formatting of the authors for liverwort species in Table 1. |
| *"**Line 313** further disturbance was detrimental"* | We corrected this. |
| *"**Line 349** rose again"* | We corrected this. |
| *"**Line 352** a difference by a factor of 5.7"* | We changed the sentence according to your comment. |

| | |
|---|---|
| *"**Lines 356-357** keep value and unit together, 59 %"* | Thanks for mentioning this, we will insert fixed spaces between values and units to avoid separating them at the end of the line. |
| *"**Line 375** skid trail"* | We corrected this. |
| *"**Line 407** with an 18-fold difference"* | We changed the sentence according to your comment. |
| *"**Line 417** both scouring water? Maybe remove both"* | We removed this sentence. |
| *"**Lines 437-438** The pH was identified as the main influencing..."* | We shortened the conclusion to the most important outcomes of our study, so that this sentence was removed at this point. |

**Figures**

**Figure 3: Development of bryophyte (n = 4) and total vegetation coverage (n = 4) per runoff plot at the individual skid trails. The bottom and top of the box represent the first and third quartiles, and whiskers extend up to 1.5 times the interquartile range (IQR) of the data. Outliers are defined as more than 1.5 times the IQR and are displayed as dots.**

**Figure 4: Species richness of bryophytes (n = 4) and vascular plants (n = 4) per runoff plot at the individual skid trails. The bottom and top of the box represent the first and third quartiles, and whiskers extend up to 1.5 times the interquartile range (IQR) of the data. Outliers are defined as more than 1.5 times the IQR and are displayed as dots.**

**Figure 6: Sediment discharge for bare (n = 14), bryophyte (n = 27) and vascular plant (n = 58) runoff plots (ROPs) categorized into cover classes. The bottom and top of the box represent the first and third quartiles, and whiskers extend up to 1.5 times the interquartile range (IQR) of the data. Outliers are defined as more than 1.5 times the IQR and are displayed as dots.**

**Figure A1: Overview of the study area: a) Location of the Schönbuch Nature Park in Germany, b) Location of the selected skid trails inside the Schönbuch Nature Park, c) Location of the four skid trails on a hillshade raster (Geobasisdaten © Landesamt für Geoinformation und Landentwicklung Baden-Württemberg, www.lgl-bw.de)**

**Figure A2: Experimental setup: a) Tübingen rainfall simulator inside the protective tent, b) Skid trail in the Trossingen-Formation (TS) in July 2019, c) Runoff plots in the wheel track and the center track in the Angulatensandstein-Formation (AS) in October 2019**

---

## Author Comment (AC6)

**Response to Reviewer Comment 3 (RC3) to**

**preprint bg-2021-343: "Pioneer biocrust communities prevent soil erosion in temperate forests after disturbances"**

Thank you very much for taking the time to revise this manuscript and for giving this positive evaluation with constructive comments. We considered your comments and revised the manuscript accordingly.

| Comments | Authors responses |
|---|---|
| *"This is an interesting study examining the importance of biocrust species on soil erosion. The experiments were conducted in an appropriate manner. Unfortunately, it is difficult to understand the contents, especially in the results and discussion section. Detailed information and key messages are mixed. A solution would be that the section is divided into the results section and the discussion section."* | We agree with your concerns about clarity and revised the results and discussion section thoroughly to make the content more comprehensible. We have paid special attention to the clear separation of pure results and interpretation against the background of the relevant literature. However, as other reviews noticed that the manuscript should not gain in length and agreed with the combination of results and discussion, we are afraid that a separation will be contradictory to that. We therefore believe it is more appropriate in this case to keep a combined results-discussion section after our adaptions. |
| *"**L109** "newly-established": When were these skid trails established? Winter 2018/19?"* | All skid trails were established in Winter 2018/19. We clarified this again in the text. |
| *"**L118** "a loess plateau": I cannot catch the meaning."* | For clarity, we replaced "plateau" with "deposition". |
| *"**L347** "bare soil ROPs": The meaning is unclear."* | We deleted this term since it was not necessarily needed at this point. |
| *"**Fig.2**: I have not understood how to obtain the biocrust coverage. Did the authors remove plants except biocrust before taking photographs for biocrust?"* | During our vegetation surveys, we determined total vegetation and bryophyte cover for each ROP, while Braun-Blanquet cover-abundance scale was used to determine coverages at the species level (Braun-Blanquet, 1964). |
| *"**Fig.3**: Why did not data of vascular plants shown in October and February? I guess the difference between the total and biocrust in Fig.2 came from vascular plants; the differences were not zero in October and February."* | Species richness for vascular plants was only surveyed for the main vegetation period in southern Germany, while species richness for bryophytes was assessed throughout the year. |
| *"**Fig.5**: Do the dots with gray color indicate?"* | The jitter points in Figure 5 indicate the single measurements in each cover class. These were removed in the revised manuscript to increase comprehensibility (see Figure 6). |

**References**

Braun-Blanquet, J.: Pflanzensoziologie. Grundzüge der Vegetationskunde, Springer Verlag, Wien 1964.

**Figures**

[Figure]

**Figure 1: Sediment discharge for bare (n = 14), bryophyte (n = 27) and vascular plant (n = 58) runoff plots (ROPs) categorized into cover classes. The bottom and top of the box represent the first and third quartiles, and whiskers extend up to 1.5 times the interquartile range (IQR) of the data. Outliers are defined as more than 1.5 times the IQR and are displayed as dots.**

---

## Author Response (AR1)

**Response to community, referee and editor comments to**

**preprint bg-2021-343: "Pioneer biocrust communities prevent soil erosion in temperate forests after disturbances"**

We would like to thank the community, editors and referees for their helpful comments, which clearly improved our text. We have prepared a revised manuscript where we address all points raised by the reviewers, as described below. Additionally, we conducted changes regarding writing, grammar and comprehensibility. All changes are tracked in the marked-up version of the manuscript.

In this point-by-point reply, reviewer comments are given in grey italic letters in the left column, while our responses are formatted in black as standard text in the right column. Line indications refer to the revised manuscript without marked changes.

**Reviewer Comment 1 (RC1):**

We thank the reviewer very much for this in depth and positive evaluation of our work. The comments provide a strong basis for substantial improvements, which are included in the revised manuscript.

| Comments | Authors responses |
|---|---|
| *"First, I doubt that many of the bryophytes reported in this study fully meet the characteristics of biological soil crusts (biocrusts). The biocrust definition, as it was first brought forward by Belnap, Büdel and Lange (2003) in the first Ecological Studies volume on biocrusts, referred to communities of organisms that live within or only few centimeters on top of soil. A key characteristic is that the major part of the biomass is located within the soil and that it creates a hardened soil surface (an encrustation). I think both of these factors are not fully met by the communities reported here. In genera like Atrichum, Rhytidiadelphus and Plagiomnium the major part of the biomass grows above the soil surface and I also have not experienced a soil hardening effect in the vicinity of them. Thus, I think the term "biological soil crust" is irritating in this context, as the reader expects somewhat different properties. I think that biocrusts indeed could occur at the slopes next to a forest path with species like Polytrichum piliferum and it might be that in some parts of the investigated sites biocrust fragments could occur. But for the complete community I doubt the correctness of this term. However, I do not see that as a deficit of this study at all. The authors could describe the studied communities as bryophyte or cryptogam communities and they could discuss the similarities and differences between biocrusts and their study objects. I think it also is relevant that not only biocrusts, but cryptogam communities in general are highly relevant for a variety of functional ecosystem processes and the present study shows this clearly once more."* | We would like to thank you for this significant comment, which hits a most interesting point that has been discussed intensively. It is agreed that the moss genera mentioned grow with the bulk of their biomass above the ground and do not meet the basic definition of a biocrust. At the same time, however, they make up a smaller part of the biomass at the beginning of succession. Along with many other moss species, single lichens, algae, and cyanobacteria, larger amounts of moss protonema can be observed on the soil surface immediately after disturbance. Together, they can show biocrust characteristics at the beginning, which fulfill the definition of Belnap et al. (2003). In this mesic forest ecosystem, however, biocrusts occurred as visually recognizable green cover, which was also reported in recent biocrust studies of comparable forest sites (Kurth et al., 2021; Glaser et al., 2022). The green cover of our sites was primarily due to bryophyte protonemata and cyanobacteria as well as coccoid and filamentous algae, which were found, as you are correctly assuming, only selectively and continued to develop quickly, with the crustal characteristic disappearing more and more (lines 242-247). Furthermore, we accounted thallose liverworts among the biocrust species (lines 247-248). Nevertheless, this observation has been made more often in mesic ecosystems, and very clearly e.g. in highly disturbed subtropical forest plantations, where larger crustal patches were still detectable after 2-3 years (Seitz et al., 2017). In this context, this early soil cover after timber harvest fulfills an essential (biocrust) function, namely, the protection against erosion at a |

| | moment when the soil is highly susceptible. This protective function then passes smoothly into further vegetation development and, according to our observations, is even more enhanced by fully developed mosses (lines 457-460). However, the distinction between biocrust and cryptogamic or just non-vascular vegetation is not always easy to make. In summary, we agree that the prominent use of the term biocrusts may lead the reader down the wrong track. This will be adjusted accordingly, and more reference to cryptogamic and/or non-vascular vegetation will be made (lines 80-83; lines 465-469; lines 477-481). Nevertheless, we think that plant communities under the biocrust definition are not yet adequately described in these mesic (and thus rather atypical) ecosystems. We therefore strongly welcome your suggestion to compare and discuss similarities and differences between the communities (lines 72-77; lines 242-248). |
|---|---|
| *"Second, I think the illustrations in this manuscript could be improved. In section 3.1.1 the composition of bryophytes is explained, but the taxa are only listed in a table and the taxonomic composition is not graphically displayed. I think this is urgently needed and would clearly improve the comprehensibility of the results. In figures 2 and 3 the line diagram is not the correct way to illustrate the results, as there are no data available for the times between the measurements. For this type of data, box-whisker plots are correct, as they have also been used in the subsequent figures. In figure 3, the signatures are difficult to be separated from each other; I think this could be improved regarding form and color. In all plots where sampling was conducted at different times, the statistics should be added in order to illustrate which changes were statistically significant."* | Thank you for your recommendation to display the taxonomic composition graphically which has considerably increased the comprehensibility of the results. We added a pie diagram that illustrates the occurrence of bryophyte species in the ROPs for each vegetation survey time step in every skid trail site (see Figure 1 in line 282). The connected scatterplot diagrams in Figures 2 and 3 were replaced with boxplot diagrams. (see Figure 3 in line 326 and Figure 4 in line 366). Furthermore, the visualization of the results in Figure 3 was adjusted so that the signatures can be better distinguished (see Figure 4 in line 366). On the level of individual skid trails, which is displayed in the figures, there are four replicates per track position, which is insufficient for performing post-hoc statistics. Furthermore, the figures are already quite detailed, which is why we did not consider it helpful to include additional information. |
| *"Third, the naming of the plots could be improved. The names of the different forests do not mean anything to the reader. I think it would be better to name the plots e.g. according to the parent material, soil type and/or texture or to just give them numbers. This would be particularly helpful, as you explain later that the substrate indeed had an effect on the observed vegetation."* | Thank you for this suggestion. The sites are named according to the geologic formation of the parent material, and the associated soil types and textures are outlined in Table A1 in the Appendix. Since these designations are already used in another publication related to this manuscript, we suggest to retain them (Thielen et al., 2021). However, with regard to your comment and as we agree that readability can generally be improved, we decided to reduce the use of abbreviations in the text. Therefore, we have spelled out CT (centre track), WT (wheel track), |

| | and UF (undisturbed forest soil) in the revised manuscript. |
|---|---|
| *"Fourth, I think it might be irritating to name only the month of sampling. It would be clearer if you name them e.g. as Mar19, Jul19, Oct19, Feb20"* | For clarification, we have added the years to the months in all figures (see Figure 2 in line 304, Figure 3 in line 326, Figure 4 in line 366, Figure 5 in line 423) and in the text, as suggested. |
| *"Fifth and finally, the language needs to be carefully and thoroughly checked throughout the manuscript. Beyond minor mistakes, which are not a big issue, there are also sentences where the meaning remains unclear. Thus, careful and thorough language editing is urgently needed before final publication could be considered."* | We regret that there were problems with our uses of English. According to your recommendation the revised manuscript has been carefully proofread by a native English speaker to improve the grammar and readability. |
| *"In line 143-145 it is written that "Four ROPs were placed in the WT and the CT in every skid trail (n = 32), and two ROPs in the undisturbed forest soil (UF) next to every skid trial site (n = 8)." This is not clear. Does it mean that on every skid trail four ROPs were installed? This would mean that there were 4 skid trails in total? Does it mean 4 skid trails each at WT and CT? This needs to be clarified. Also the rainfall simulation numbers given in the following sentence are not clear. I think a thorough language check will help to also clarify these issues."* | We followed your suggestion and clarified the sampling design in lines 164-168. In total, we had four skid trails and installed four ROPs in each wheel track and centre track (n = 32), and two ROPs in the undisturbed forest soil adjacent to every skid trail (n = 8). The rainfall simulations in the skid trails were repeated four times a year (March 2019, July 2019, October 2019, February 2020), while the rainfall simulations in the undisturbed forest soil were repeated twice in October 2019 and February 2020. In summary, this brings us to 144 measurements. |
| *"Line 35-37: In this sentence there are several language style problems. I would suggest to reformulate it in the following way: The most prominent soil loss occurs in agricultural environments, and thus a considerable part of relevant research is conducted in these habitats."* | Thank you for the wording suggestion. We changed the sentence accordingly (line 40-41). |
| *"Line 46-47: here I think you want to say "The most important reason for this is soil compaction and reduced infiltration rates caused by heavy machines used for timber harvesting""* | Thank you for bringing this to our attention. We inserted the word "caused" in line 50, which clearly improves the sentence. |
| *"Line 48: significantly"* | We inserted "significantly" (line 52). |
| *"Line 55: exchange "which" by "that""* | We replaced "which" by "that" (line 59). |
| *"Line 60: "These" instead of "those""* | We exchanged "Those" by "These" (line 63). |
| *"Line 75: As most studies investigating the impact..."* | We adjusted the sentence accordingly in lines 86-88. |
| *"Line 80-81: This sentence is upside down. 'Pioneer biocrust communities could provide benefits' or 'the soil benefits from biocrusts'"* | We changed the order of the sentence as suggested (line 92-93). |
| *"Line 114: The skid trails show no geological formation, but the underlying rocks and soil do. Please adapt wording"* | We have made clarifying rephrasings for this purpose (lines 132-133). |
| *"Line 119: formed by extensive periglacial processes..."* | We have reformulated the sentence accordingly (line 137-138). |
| *"Line 125-127: There are several abbreviations that need to be explained: Ad-hoc-Ag Boden, Iuss Working Group Wrb, WRB Tool"* | The explanations for the abbreviations were inserted in the revised manuscript in lines 144-146. |

| | |
|---|---|
| *"Line 148: A rainfall intensity of 45 mm does not make sense. I think you speak of a rainfall intensity of 90 mm h-1, applied over a duration of 30 minutes"* | Thank you for clarifying this. We have corrected the sentence accordingly in lines 170-171. |
| *"Line 200-201: meaning of sentence unclear"* | We removed this sentence. |

**Community Comment 1 (CC1)**

Thank you again for selecting our study for discussion in your undergraduate course. We were very pleased that you have dealt with our manuscript in such detail and your feedback has contributed significantly to the improvement of our manuscript.

| Comments | Authors responses |
|---|---|
| **Abstract**
*"In overall it is well written. We suggest the authors make clearer why is it important to study what they have studied e.g. why is the succession so important?"* | To date, very little is known about when soil-protective vegetation begins to develop in a forest disturbance area, so that it is important to monitor the process of succession. Furthermore, we wanted to determine the timing of biocrust occurrence and its impact on soil erosion, which is generally poorly studied in temperate climates. We included a sentence in the abstract that highlights the importance of vegetation succession to our study (lines 14-16). |
| *"In addition, the final 2-3 sentences of the 'Abstract' need some modifications – making them simpler and easier to understand will increase their impact. It might be preferable to avoid use of "we" in the abstract and perhaps the detail of results could be reduced; authors might also be more clear in highlighting one main conclusion to express."* | Thank you very much for this comment, as it helped to improve the presentation of results in the abstract. We reduced the results in the abstract to the most important points of the study, which increases comprehensibility for the reader (lines 27-32).
Additionally, we now use the passive form exclusively in the abstract. |
| **Introduction**
*"Line 35: Please provide some examples why soil erosion will increase through climate change. Also are there any (numerical) projections about how much erosion will increase in years and decades to come?"* | In the context of climate change, increasing rainfall intensities are the key driver of soil erosion, as this enhances the erosive power of precipitation and thus the probability of soil losses. We added this example in the introduction (line 39).
In our opinion, further examples and explanations would lead too far at this point. Projections of soil erosion rates are clearly influenced by local conditions and the percentage increase varies widely. For example, Li and Fang (2016) concluded that 136 studies predict an increase in soil erosion rates in the future, with relative increases ranging from 1.2% to 1614%. |
| *"Line 41-42: Is there a reason behind these relatively large shifts in erosion of forestlands?"* | There are a variety of factors influencing soil erosion in forests, and it would be too much of a stretch to discuss them all at this point. In the references we highlighted here, the shifts in erosion rate referred mainly due to forest management intensity and tree species composition. We added these factors in in the mentioned lines of the introduction (lines 45-46). |

| | |
|---|---|
| *"Line 44: Please use more plain language in "showed that unsealed forest roads at the catchment scale" so that the reader can get a clearer understanding."* | We simplified the sentence in the mentioned lines 48-50. |
| *"Lines 46-53: This numerical information provided is useful, but we feel it would be better to be used in the "Discussion". Here in the "Introduction" make sure you present the bigger picture and why it is important for this research to be carried out. Lots of numerical information can distract the readers from the major messages."* | According to your recommendation, we reduced the number of numerical information in the introduction to avoid distractions from the overall context. |
| *"The sentence on line 55 could be modified to summarise the point of referencing all of these studies and then group them together in the citation for reference"* | We followed your comment and changed the structure of this sentence (lines 58-59). |
| *"Line 61: Please explain where the term "cryptogamic" refers to. Also, what do you mean by "understory"?"* | We explained what we meant by "cryptogamic" in the introduction (lines 64-66). This term includes all non-flowering plants and plant-like organisms that reproduce by spores, such as bryophytes, lichens, ferns, algae and fungi. The term "understory" was also specified in the introduction (line 64). By this we refer to the vegetation growing on the forest soil. |
| *"Line 63: Perhaps replace "edaphic" by "floor"?"* | We replaced "edaphic" by "soil" in this line 67. |
| *"Line 68: We feel this should be "bryophyte-dominated"?"* | Thank you for bringing this to our attention. We corrected this according to your comment (line 78). |
| *„Lines 68-70: Please provide briefly some information on the direction of these effects by bryophytes e.g. increase/decrease in runoff etc."* | We provided the direction of these effects in the mentioned lines 78-80. |
| *"Line 81: Please improve wording."* | We changed the order of the sentence to improve wording (lines 92-93). |
| *"Line 86: The authors need to make clearer which is the research gap and especially to link it better with previous lines/sections."* | We followed your suggestion and clarified the research gaps in the introduction (lines 98 – 106). |
| *"Lines 92-94. It is welcome that authors make clear the objectives of their study. We feel though that it would be even better if they make some null hypotheses related to their points e.g., how do they expect that the underlying substrate, vegetation cover and track position will affect soil erosion?"* | Thank you for noting this. We agree that null hypotheses improve the comprehensibility of the manuscript and implemented this wherever possible (lines 112-117). |
| *"Line 96: Please explain what you mean by "interrill"."* | Soil erosion processes by water can be divided into three different stages: splash erosion, interrill and rill erosion. Interrill erosion is known as the discharge of sediment in thin sheets between rills by shallow surface runoff after raindrop impact (Blanco and Lal, 2008). We added an explanation on that term in the introduction (lines 162-164). |
| *"In the "Introduction" and especially towards the end of it the authors should make some clearer references on how their findings can be* | According to your suggestion, we added a short outlook of good practices for forestry at the end of the introduction (lines 104 – 106). |

| | |
|---|---|
| *used in good practices for management. They can elaborate on that aspect in the discussion.”* | |
| **Materials and Methods** *"Line 121 and further: Could abbreviate genus name in species scientific names for conciseness purposes (e.g. P. sylvestris)”* | We decided to not abbreviate genus names. |
| *"Lines 140-146: Please provide references about the use of similar experimental set up in previous studies.”* | We provided more references about the use of rainfall simulators in combination with small-scale runoff plots (line 163). |
| *"Lines 148-149: The authors need to provide more information about the particular selection of this rainfall intensity e.g., is similar intensities observed often in the studied area? Provide also relevant references.”* | We inserted more background information to the selected rainfall intensity with a reference (lines 170-172). |
| *"Line 149- 153: The authors should provide more details about technical aspects mention in there e.g., measurements on surface run off. Please also provide references.”* | In our opinion, it is not necessary to add further details or references here, since the common procedure in soil erosion measurements is to collect surface runoff and the sediment discharged with it in sample bottles. References to this are already available in the previous section. |
| *"Line 154-155: For how long were the samples left to dry?”* | It usually took about three to four weeks until all samples were dry. But this depended strongly on the amount of water, which was different in each measurement. |
| *"Line 156: Please mention what is exactly the aggregate size and which are the measurement units for this parameter.”* | Soil aggregate size is a basic parameter in soil science what we assume as basic knowledge and would not explain it in detail in the manuscript. Depending on a variety of biotic and abiotic factors, soil forms aggregates consisting of agglutinated soil particles. |
| *"Line 159-162: It is interesting that measurements on elements (C, N) were made. Please make sure that there are the relevant references made in the "Introduction" so the sections of the manuscript align better.”* | According to your comment, we mentioned carbon and nitrogen levels in the introduction to point out the gap of knowledge of the factors that affect bryophyte species richness and cover (lines 98 – 101). |
| *"Line 173. Please improve the wording about nomenclature in Tables.”* | We improved the wording of this sentence (line 195). |
| *"Lines 183-187. It seems that post-hoc tests were not carried out. Also, it seems that the role of environmental parameters in the flora structure / development has not been accounted/examined for. If this is the case, then it is regarded as a major gap and needs to be addressed.”* | We performed post hoc Wilcoxon rank-sum tests and Wilcoxon signed-rank tests for sediment discharge, coverage and species richness averaged for all skid trail sites and averaged for wheel and centre tracks in each skid trail. On the level of individual skid trails, there are four replicates per track position, which is insufficient for performing post hoc statistics. To assess the effect of environmental parameters on soil erosion, bryophyte coverage and species richness, we performed generalized additive models (GAM) with restricted maximum likelihood and smoothing parameters selected by an unbiased risk estimator (UBRE) criterion. |
| *"More information on the number of replicates is needed.”* | We revised the information on the replicates in the method section so that the sample design is now more comprehensible (164-168). |

| | |
|---|---|
| *"A map showing where the research was carried out would be welcome."* | We added an extra figure (new Figure A1) for the location of the study area in the Appendix (line 518). |
| *"Overall, we believe that the "Materials and Methods" section could have been written more succinctly to make it easier to read."* | We will try to make the methods section more concise. |
| **Results and Discussion**
*"Line 191: 'Section 3.1.1 – Biocrust species composition'. It seems that this title is not fully adequate as in the section 3.1.1 there are also results about temporal trends. This should be reflected in the Section 3.1.1 title."* | We have changed the titles as well as the division within section 3.1. The titles also reflect the temporal trends (line 223, line 263). |
| *"Line 193: Please avoid using where possible abbreviations (e.g., 'UF') as it is difficult for the reader to follow them."* | According to your comment, we decided to reduce the use of abbreviations in the text in order to improve readability. Therefore, we spelled out CT, WT, and UF in the revised manuscript. |
| *"Line 196: Please clarify what is 'protonema'."* | We introduced the term "protonema" in the results and discussion section (line 227-228). Protonema is the earliest stage of bryophyte development consisting of green cell filaments. |
| *"Line 205 / Table 1: Could table 1 provide more information on composition, cover and richness? Do we need Author column?"* | We implemented your idea by providing additional information on the percentage occurrence of species in the runoff plots in total and for each vegetation survey time (Table 1, line 234 and Table 2, line 239).
Furthermore, we added a diagram that illustrates the occurrence of bryophyte species in the ROPs for each vegetation survey time in every skid trail. This also includes more information about taxonomic composition and species richness at the different skid trails and considerably increased the comprehensibility of the results (Figure 1, line 282).
In the botanical nomenclature a reference to the person who first gave a name to the botanical entity is required and we followed these rules. Instead we discarded the family names. |
| *"Lines 222-223: This is just an assumption on the role of pH; there should be appropriate statistical analysis to explore the role of abiotic environmental parameters in shaping the communities."* | The effect of environmental parameters on species composition was not a focus of our study, so we made an assumption at this point that we did not support with statistical analysis. |
| *"Tables 1 and 2: The information shown here is interesting; however it seems that these Tables are a bit long – how about moving them to Supplementary Material?"* | With the additional information on the percentage occurrence of species in the runoff plots, we believe that Table 1 and 2 should remain in the text. |
| *"Lines 227-230: These are major findings and should be moved earlier/up in the Results and Discussion section."* | In this section, we first wanted to give a general overview of the occurrence of bryophyte species in the research area (section 1) and discuss this species composition (section 2). Afterwards, section 3 deals with the different species compositions of the four skid trail sites, which is why more detailed results are listed there for the first time. |

| | |
|---|---|
| *"Lines 232: Please clarify the categories that the species belong to e.g. do they belong to 'protonema' or another category?"* | For protonemata, we did not determine the species, so either the moss occurred as protonema or the species was mentioned. |
| *"Line 234: "little importance": Please provide numbers rather than terms like "little importance"."* | As recommended, we changed this wording and inserted numbers instead (line 272-273). |
| *"Lines 227-242: This is a big chunk of results but discussion on them is absent."* | The discussion of these results can be found in the following section. |
| *"Line 243: It would be better to start the section with the key result; discussion on it should follow."* | Based on your comment, we have restructured this section so that it is now more comprehensible and exciting for the reader (lines 285-289). |
| *"Line 246: Please see comments above about stats regarding the role of environmental parameters."* | To assess the effect of environmental parameters on soil erosion (lines 396-397), bryophyte coverage (lines 315-317) and species richness (lines 364-365), we performed generalized additive models (GAM) with restricted maximum likelihood and smoothing parameters selected by an unbiased risk estimator (UBRE) criterion. |
| *"Figure 1: Could be useful to have included a longer caption describing what photographs demonstrate to make the article more accessible for the readers that do preliminary paper skimming. A map of the area would have been highly beneficial for the readers to better visualise the studied site spatial distribution."* | We added a title to Figure 1 (new Figure 2, line 304).
We provided an extra figure (new Figure A1) for the study area in the Appendix (line 518). |
| *"Line 271: It is not clear what the authors try to say here e.g. that there are similar trends between biocrust and total coverage trends? Or something else? Please clarify."* | We clarified this sentence in the revised manuscript (line 330). |
| *"Figure 2 caption: Perhaps it would read better as "mean values and standard error are given". Please also remind to the readers the number of replicates."* | The connected scatterplot diagrams in Figures 2 and 3 were replaced with boxplot diagrams, so that the figure description is different now. We added the number of replicates in all figure descriptions. |
| *"Line 282: The values of pH should be mentioned."* | The pH values were included in the text (319-321). |
| *"Lines 288-289: The authors should elaborate on their statements about contradictions between their findings and those from (Corbin and Thiet, 2020; Bergamini et al., 2001; Fojcik et al., 2019)."* | We added more information in this section to clarify the contradictory results (lines 341-351). |
| *"Lines 289-292: The authors should elaborate on the mechanisms driving positive correlations between vascular plants and moss growth."* | We inserted additional information on this in the mentioned lines (lines 347-354). |
| *"Line 292: The statements/discussion on biocrust should be on a separate paragraph."* | In order to better distinguish between biocrust cover and bryophyte cover, we have revised the entire manuscript so that we now refer to bryophyte covers in this line. |
| *"Lines 327 – 338: Please make sure that you provide p-values where needed. Also, it is not necessary to use extensively phrases such as "A was X times higher than B". Providing the* | Thank you for this comment. We included p-values, means, and standard deviations wherever appropriate. |

| | |
|---|---|
| *average values, standard error and the p-values would suffice."* | |
| *"Lines 339-341: See our comments above about examining the role of environmental parameters in shaping discharge / run off. For example, how much of the variability in discharge is explained by differences in the soil features?"* | To assess the effect of environmental parameters on soil erosion, bryophyte coverage and species richness, we performed generalized additive models (GAM) with restricted maximum likelihood and smoothing parameters selected by an unbiased risk estimator (UBRE) criterion. |
| *"Sections 3.2.1 and 3.2.2 should be merged. The independent and response variables should be subject to appropriate statistical analysis e.g. distance-based linear modelling (Clarke and Gorley 2015) Clarke KR, Gorley RN (2015) PRIMER v7: User Manual/Tutorial PRIMER-E: Plymouth"* | We merged the sections 3.2.1 and 3.2.2 and added the results of our GAMs. |
| *"Lines 398-401: Some of the lines mentioned here should had been included in the Materials and Methods. Also it is not clear where the term 'reduction' refers to – please clarify."* | As suggested, we moved the mentioned lines to the methods section (lines 207-210). Further we clarified that the term "reduction" in this section refers to sediment discharge (lines 453-455). |
| *"Figure 5: The box plots for biocrusts and vascular plants are very close (this is not necessarily bad) and some of the outliers for biocrusts may be regarded as outliers for vascular plants (and vice versa). It would be helpful to see the outliers for each of them with different colours. We feel that a sudden change in the colour scheme on this graph could confuse the readers that got used to seeing dark green as 'wheel track' and light green as 'central track' in previous 3 figures."* | Thank you for bringing this to our attention. We adjusted the colour code in all figures so that dark green is used for "bryophytes" and light green for "vascular plants". Additionally, we removed the jitter points in Figure 5 (new Figure 6, line 461), as this was the most appropriate measure to make the outliers of each displayed group more visible, and this also significantly improved the visualization. |
| **Conclusions** *"Line 426 : it seems that null hypotheses were not made; it is suggested to adjust accordingly the text at the end of the "Introduction"."* | According to your comment, we inserted null hypotheses in the introduction section wherever possible and adjusted the conclusion section as well. |
| *"The conclusions section looks too lengthy; it should appear more succinct and with higher impact. Focus on your key findings and how they fill gaps in the literature. Avoid repeating results and numerical values."* | We shortened the conclusion to the most important outcomes of our study (lines 487-504). |
| *"Could include more discussion of direction and opportunities for future studies"* | At the end of the conclusion section, we mentioned several future research topics related to our study. We think that this outlook is sufficient. |
| *"Lines 450- 456: Would it be also of interest to study the factors that support higher growth rates for the biocrust communities?"* | Yes, this is very interesting to study, especially in temperate climates where the evidence of biocrust communities is scarce. |
| *"Figure A1. Please clarify in the image (using arrows) the wheel track and center track."* | We inserted arrows in Figure A1 (new Figure A2, line 522) to mark the location of wheel and centre tracks. |
| *"It seems that there is some inconsistency in editing/coloring of symbols across the figures e.g., see color code used Figures 3 and 4."* | Thank you for bringing this to our attention. We adjusted the colour code in all figures so that dark green is used for "bryophytes" and light green for "vascular plants". |

**Reviewer Comment 2 (RC2)**

Thank you very much for your review, the positive evaluation of our work and the very valuable suggestions to improve the manuscript.

| Comments | Authors responses |
|---|---|
| **Figure 2 and 3**
*"For Figure 2 and 3 I would recommend not using line charts but possibly box plots. Since these are specific monitoring times and not continuous monitoring it gives the wrong suggestion to the reader, especially since the slope of the lines is very different (because the x-axis distances are all the same, although timewise they are not, June-July is not the same time as July-October)."* | Thank you for bringing this to our attention. We replaced the connected scatterplot diagrams in Figures 2 and 3 with boxplot diagrams (see Figure 3 in line 326, Figure 4 in line 366). |
| **Figure 2**
*"Perhaps you could consider, for Figure 2, putting the difference between wheel track and center track in one panel (bryophytes) and the difference between wheel track and center track for total vegetation in another panel. With an adjusted y-axis for bryophytes it would be much easier to see differences between the two track types. This is just a suggestion."* | Thank you for this recommendation. We tried the suggested display for Figure 2, but discarded it after closer examination because we would have to distinguish colour between wheel and centre tracks for this representation, and we believe that it is more comprehensible to the reader at this point to stick with the selected uniform colour code to distinguish between "bryophytes" and "vascular plants". Please see also community comment #1. |
| **Figure 2 and 3**
*"To distinguish the information in Figure 2 from Figure 3 it might be better to use different colours. In Fig. 2 bryophytes are presented in dark green while total vegetation is yellowish, in Fig. 3 these colours are used to distinguish the track types which makes it more difficult to grasp the information from the figure directly. Consider using larger symbols for bryophytes etc. so it is more easily readable."* | We decided to adjust the colour code in all figures so that dark green is used for "bryophytes" and light green for "vascular plants", which makes the figures more comprehensible for the reader. |
| **Figure 5**
*"The distribution of sample dots in Figure 5 just seems random and does not improve the quality of the figure. The information about the number of sampling points could also be added into the figure caption."* | Thank you for this comment. We removed the jitter points in Figure 5 which clearly improved the visualization (new Figure 6, line 461). Furthermore, we added the number of sample points for each cover in the figure caption (line 462). |
| *"**Line 148** rainfall intensity should be given as mm h-1. Do you mean 45 mm in 30 minutes meaning 90 mm h-1. This would be an extremely heavy precipitation event and one not typically found in the region, I presume."* | We inserted more background information to the selected rainfall intensity and corrected the given unit to mm h$^{-1}$ (lines 170-172). |
| *"**Chapter 3.2.1** I understand that you want to distinguish the skid trails from the undisturbed forest, yet the results seem to show that wheel tracks and center tracks are very different in their soil erosion characteristics, maybe separate them when speaking about the total values for sediment discharge and surface runoff."* | As suggested, we removed the mean values for the entire skid trails in this chapter and instead only dealt with the mean values per wheel track and centre track (lines 387-391). |

| | |
|---|---|
| *"**Lines 358-364** You speak of rainfall events, but you mean rainfall simulations? As I understand it, these ROPs can also be used to measure sediment loss and surface runoff during natural rainfall events, did you measure these in between your monitoring times?"* | Yes, we mean rainfall simulations in these cases. We clarified this in line 416-422. Generally, ROPs can be used to measure surface runoff and sediment discharge during simulated rainfall and natural rainfall events. In our study, we just conducted measurements with simulated rainfall. |
| *"**Figure 5** As you write the higher the percentage of vascular plant cover or biocrust cover the lower sediment loss. Why is the sediment discharge for 11-25 % biocrust cover so low in comparison to the sediment discharge with higher biocrust cover (26-50%)? Do you think it is because of only few measurements were performed in this cover class? You should also explain not only the outlier dots but also your „sample" dots in the figure caption."* | In Figure 5, our measurements of sediment discharge at four different skid trail sites were reclassified and plotted in cover classes to represent the general influence of bryophytes and vascular plants on soil erosion. Except of the cover class "< 10 %" with 13 measurements, we have 3 – 4 measurements for bryophyte ROPs in each cover class, so this difference is not due to sample size. We assume the reason is that different skid trail sites are grouped together in each cover class. Cover class "11-25 %" includes two measurements of TS and one of LS, while cover class "26-50 %" contains two measurements of PT, one of TS and one of LS. Due to skid trail site, soil erosion was significantly higher in PT than in TS. The jitter points in Figure 5 were removed to increase comprehensibility (see Figure 6 in line 461). |
| *"**Figure A1** Unfortunately, the rainfall simulator (except for the cannot be seen, consider using a different, more expressive picture."* | We replaced image "a" in Figure A1 (new Figure A2, line 522) so that you can also see the Tübingen rainfall simulator inside the protective tent. |
| *"**Chapter 2.1** Consider adding an extra figure for the study area"* | We added an extra figure (new Figure A1, line 518) for the study area in the Appendix. |
| *"**Lines 27- 28** the last sentence needs work: ... biocrusts showed an average sediment loss that was 18 times lower than under vascular plants."* | We decided to delete this sentence in the abstract because it was too specific at this point. |
| *"**Line 41** important dimensions?"* | We have rephrased this sentence to make clearer that soil erosion in forests can be locally very severe (lines 45-46). |
| *"**Line 68** bryophyte-dominated biocrusts"* | Thank you, we corrected this according to your comment (line 78). |
| *"**Line 75** very most? As the most studies"* | According to your comment, we deleted "very" in this sentence (line 84). |
| *"**Line 127** „a" Eutric Cambisol"* | We adjusted this (line 146). |
| *"**Line 135** „a" Eutric Calcaric"* | We adjusted this (line 155). |
| *"**Line 173** Nomenclature see Table 1 and Table 2 à please use full sentences or use brackets"* | We improved the wording of this sentence in line 195. |
| *"**Line 202** no italics for citation"* | We removed this sentence. |
| *"**Table 1** no italics for the authors"* | We changed the formatting of the authors for liverwort species in Table 1 in line 236. |
| *"**Line 313** further disturbance was detrimental"* | We corrected this (line 374). |
| *"**Line 349** rose again"* | We corrected this (line 407). |
| *"**Line 352** a difference by a factor of 5.7"* | We changed the sentence (line 410). |
| *"**Lines 356-357** keep value and unit together, 59 %"* | Thanks for mentioning this, we inserted fixed spaces between values and units to avoid separating them at the end of the line. |

| | |
|---|---|
| *"**Line 375** skid trail"* | We corrected this (line 433). |
| *"**Line 407** with an 18-fold difference"* | We changed the sentence according to your comment (line 460). |
| *"**Line 417** both scouring water? Maybe remove both"* | We removed this sentence. |
| *"**Lines 437-438** The pH was identified as the main influencing..."* | We shortened the conclusion to the most important outcomes of our study, so that this sentence was removed at this point. |

**Reviewer Comment 3 (RC3)**

Thank you very much for taking the time to revise this manuscript and for giving this positive evaluation with constructive comments. We considered your comments and revised the manuscript. The detailed responses to your comments can be found in the following table.

| Comments | Authors responses |
|---|---|
| *"This is an interesting study examining the importance of biocrust species on soil erosion. The experiments were conducted in an appropriate manner. Unfortunately, it is difficult to understand the contents, especially in the results and discussion section. Detailed information and key messages are mixed. A solution would be that the section is divided into the results section and the discussion section."* | We agree with your concerns about clarity and revised the results and discussion section thoroughly to make the content more comprehensible. We have paid special attention to the clear separation of pure results and interpretation against the background of the relevant literature. However, as other reviews noticed that the manuscript should not gain in length and agreed with the combination of results and discussion, we are afraid that a separation will be contradictory to that. We therefore believe it is more appropriate in this case to keep a combined results-discussion section after our adaptions. |
| *"**L109** "newly-established": When were these skid trails established? Winter 2018/19?"* | All skid trails were established in winter 2018/19. We clarified this again in line 128. |
| *"**L118** "a loess plateau": I cannot catch the meaning."* | For clarity, we replaced "plateau" with "deposition"(line 137). |
| *"**L347** "bare soil ROPs": The meaning is unclear."* | We deleted this term since it was not necessarily needed at this point (line 405). |
| *"**Fig.2**: I have not understood how to obtain the biocrust coverage. Did the authors remove plants except biocrust before taking photographs for biocrust?"* | During our vegetation surveys, we determined total vegetation and bryophyte cover for each ROP, while Braun-Blanquet cover-abundance scale was used to determine coverages at the species level (Braun-Blanquet, 1964). |
| *"**Fig.3**: Why did not data of vascular plants shown in October and February? I guess the difference between the total and biocrust in Fig.2 came from vascular plants; the differences were not zero in October and February."* | Species richness for vascular plants was only surveyed for the main vegetation period in southern Germany, while species richness for bryophytes was assessed throughout the year. |
| *"**Fig.5**: Do the dots with gray color indicate?"* | The jitter points in Figure 5 indicate the single measurements in each cover class. These were removed in the revised manuscript to increase comprehensibility (see Figure 6 in line 461). |

**References**

Belnap, J., Büdel, B., and Lange, O. L.: Biological Soil Crusts: Characteristics and Distribution, in: Biological Soil Crusts: Structure, Function, and Management, edited by: Belnap, J., and Lange, O. L.,

Springer Berlin Heidelberg, Berlin, Heidelberg, pp. 3-30, https://doi.org/10.1007/978-3-642-56475-8_1, 2003.

Blanco, H. and Lal, R.: Principles of Soil Conservation and Management, Springer New York 2008.

Braun-Blanquet, J.: Pflanzensoziologie. Grundzüge der Vegetationskunde, Springer Verlag, Wien 1964.

Glaser, K., Albrecht, M., Baumann, K., Overmann, J., and Sikorski, J.: Biological Soil Crust From Mesic Forests Promote a Specific Bacteria Community, Frontiers in Microbiology, 13, https://doi.org/10.3389/fmicb.2022.769767, 2022.

Kurth, J. K., Albrecht, M., Karsten, U., Glaser, K., Schloter, M., and Schulz, S.: Correlation of the abundance of bacteria catalyzing phosphorus and nitrogen turnover in biological soil crusts of temperate forests of Germany, Biology and Fertility of Soils, 57, 179-192, https://doi.org/10.1007/s00374-020-01515-3, 2021.

Li, Z. and Fang, H.: Impacts of climate change on water erosion: A review, Earth-Science Reviews, 163, 94-117, https://doi.org/10.1016/j.earscirev.2016.10.004, 2016.

Seitz, S., Nebel, M., Goebes, P., Käppeler, K., Schmidt, K., Shi, X., Song, Z., Webber, C. L., Weber, B., and Scholten, T.: Bryophyte-dominated biological soil crusts mitigate soil erosion in an early successional Chinese subtropical forest, Biogeosciences, 14, 5775-5788, https://doi.org/10.5194/bg-14-5775-2017, 2017.

Thielen, S. M., Gall, C., Ebner, M., Nebel, M., Scholten, T., and Seitz, S.: Water's path from moss to soil: A multi-methodological study on water absorption and evaporation of soil-moss combinations, Journal of Hydrology and Hydromechanics, 69, 421-435, https://doi.org/10.2478/johh-2021-0021, 2021.